# TUNING-FREE BILEVEL OPTIMIZATION: NEW ALGORITHMS AND CONVERGENCE ANALYSIS

**Yifan Yang**[1]**, Hao Ban**[1]**, Minhui Huang**[2]**, Shiqian Ma**[3]**, Kaiyi Ji**[1]
[1]University at Buffalo, [2]Meta, [3] Rice University
[1]{yyang99, haoban, kaiyiji}@buffalo.edu,
[2]mhhuang@meta.com,[3]sqma@rice.edu

## ABSTRACT

Bilevel optimization has recently attracted considerable attention due to its abundant applications in machine learning problems. However, existing methods rely on prior knowledge of problem parameters to determine stepsizes, resulting in significant effort in tuning stepsizes when these parameters are unknown. In this paper, we propose two novel tuning-free algorithms, D-TFBO and S-TFBO. D-TFBO employs a double-loop structure with stepsizes adaptively adjusted by the "inverse of cumulative gradient norms" strategy. S-TFBO features a simpler fully single-loop structure that updates three variables simultaneously with a theory-motivated joint design of adaptive stepsizes for all variables. We provide a comprehensive convergence analysis for both algorithms and show that D-TFBO and S-TFBO respectively require $\mathcal{O}(\frac{1}{\epsilon})$ and $\mathcal{O}(\frac{1}{\epsilon}\log^4(\frac{1}{\epsilon}))$ iterations to find an $\epsilon$-accurate stationary point, (nearly) matching their well-tuned counterparts using the information of problem parameters. Experiments on various problems show that our methods achieve performance comparable to existing well-tuned approaches, while being more robust to the selection of initial stepsizes. To the best of our knowledge, our methods are the first to completely eliminate the need for stepsize tuning, while achieving theoretical guarantees.

## 1 INTRODUCTION

Bilevel optimization has gained considerable attention recently due to its widespread use in various machine learning applications, such as meta-learning (Franceschi et al., 2018; Bertinetto et al., 2018; Rajeswaran et al., 2019), hyperparameter optimization (Shaban et al., 2019; Feurer & Hutter, 2019), reinforcement learning (Konda & Tsitsiklis, 2000; Hong et al., 2023a), robotics Wang et al. (2024), communication (Ji & Ying, 2022) and federated learning Tarzanagh et al. (2022). In this paper, we study a standard bilevel optimization problem that takes the following mathematical formulation:

$$\min_{x \in \mathbb{R}^{d_x}} \Phi(x) := f\big(x, y^*(x)\big)$$
$$\text{s.t. } y^*(x) = \arg\min_{y \in \mathbb{R}^{d_y}} g(x, y), \tag{1}$$

where $f$ and $g$ are jointly continuously differentiable outer (upper-level) and inner (lower-level) functions. In this paper, we focus on the nonconvex-strongly-convex setting, where the lower-level function $g$ is strongly convex w.r.t. $y$ and the outer function $\Phi(x)$ is possibly nonconvex.

Recent years have witnessed the rapid development of bilevel optimization algorithms, which can be categorized into approximate implicit differentiation (AID) (Ji et al., 2021; Dagréou et al., 2022) based, iterative differentiation (ITD) (Ji et al., 2022; Grazzi et al., 2020) based, and value-function based (Kwon et al., 2023; Liu et al., 2021a) approaches. However, these methods often require substantial effort to tune a couple of hyperparameters like stepsizes, which typically depend on **unknown** problem parameters (such as Lipschitzness parameters, strong convexity parameters, and optimal function values). This emphasizes the importance of *adaptive and tuning-free* methods in bilevel optimization. *In this paper, an algorithm is considered tuning-free if it does not need to know the problem parameters in advance but can still achieve almost the same convergence rate guarantee as its well-tuned counterpart using this information.* Despite several recent efforts

to reduce dependence on problem-specific parameters (Fan et al., 2024; Antonakopoulos et al., 2024), developing a fully tuning-free bilevel optimization algorithm remains an open challenge. For instance, Fan et al. (2024) utilizes Polyak's stepsizes to automate both inner and outer updates but still requires information such as gradient Lipschitzness parameters and optimal lower-level function values. Similarly, Antonakopoulos et al. (2024) introduces an "on-the-fly" accumulation strategy for (hyper)gradient norms, which removes the reliance on inner and outer gradient Lipschitzness parameters but still depends on the strong convexity parameter for the inner AdaNGD-type updates.

This paper aims to close this gap by introducing two novel fully tuning-free bilevel optimization algorithms named D-TFBO and S-TFBO (where D and S represent double- and single-loop approaches), along with a comprehensive convergence analysis demonstrating their competitive performance compared to existing well-tuned approaches (which tune their hyperparameters like stepsizes based on the problem parameters). Our key contributions are outlined below.

- Our algorithms are inspired by the "inverse of cumulative gradient norms" strategy introduced by Xie et al. (2020); Ward et al. (2020), adapting the stepsizes based on accumulated (hyper)gradient norms. D-TFBO utilizes two optimization sub-loops: one for solving the inner problem and another for addressing a linear system (LS), which approximates the Hessian-inverse-vector product of each hypergradient. Unlike previous approaches, D-TFBO introduces cold-start adaptive stepsizes that accumulate gradients exclusively within the sub-loops. This method establishes a tighter lower bound on stepsizes, improving gradient complexity. In contrast, S-TFBO adopts a single-loop structure, where all variables are updated simultaneously in each iteration. Rather than applying the "inverse of cumulative gradient norms" uniformly to all updates, our error analysis motivates a joint design of adaptive stepsizes for $y$, $v$, and $x$, which correspond to solving the inner problem, LS, and outer problem, respectively. For instance, the stepsize for $v$ is coupled with that for $y$, while the stepsize for $x$ depends on both $y$ and $v$.
- Compared to the well-tuned AID methods in Ji et al. (2022), our D-TFBO method achieves the same $\mathcal{O}(\frac{1}{T})$ convergence rate. Similarly, our S-TFBO method attains an $\widetilde{\mathcal{O}}(\frac{1}{T})$ convergence rate, matching that of well-tuned counterparts, up to polylogarithmic factors. The complexity analysis shows that D-TFBO and S-TFBO require $\mathcal{O}(\frac{1}{\epsilon^2})$ and $\widetilde{\mathcal{O}}(\frac{1}{\epsilon})$ gradient computations, respectively, to reach an $\epsilon$-accurate stationary point. This comparison differs from the observation in well-tuned bilevel optimization, where double-loop approaches generally achieve lower gradient complexity than single-loop methods (Ji et al., 2022). This is because the inner tuning-free solver requires $\mathcal{O}(\frac{1}{\epsilon})$ more iterations than well-tuned methods to achieve $\epsilon$-level accuracy.
- The theoretical analysis is inspired by the two-stage framework in Xie et al. (2020); Ward et al. (2020), where the stages describe the relationship between the stepsizes and certain constants that depend on the problem parameters. However, exploring this technical framework in bilevel problems is far more challenging because the stages for analyzing each stepsize interact with those for other stepsizes, resulting in intertwined multi-stage dynamics across different variables. For instance, the error analysis for the updates on $v$ must account for the accumulated gradient norms from the updates on $y$. This motivates us to couple the stepsize for $v$ with the adaptive stepsize for $y$ to prevent the propagation of accumulated errors. In addition, our analysis requires establishing precise upper and lower bounds for all stepsizes to ensure convergence results that match those achieved under well-tuned stepsizes.
- We validate the effectiveness of our methods through experiments on regularization selection, data hyper-cleaning, and coreset selection for continual learning. The results show that our methods perform comparably to existing well-tuned methods. More importantly, our methods demonstrate greater robustness to different initial stepsizes, due to the tuning-free design.

## 2 RELATED WORK

**Bilevel Optimization.** Bilevel optimization, initially introduced by Bracken & McGill (1973), has been extensively studied for decades. Early works (Hansen et al., 1992; Shi et al., 2005) approached the bilevel problem from a constrained optimization perspective. More recently, gradient-based methods have gained significant attention for their efficiency and effectiveness. Among these, Approximate Implicit Differentiation (AID) methods (Domke, 2012; Liao et al., 2018; Ji et al., 2021; Dagréou et al., 2022) leverage the implicit derivation of the hypergradient by approximating it through the solution of a linear system. In contrast, Iterative Differentiation (ITD) methods (Maclaurin et al., 2015; Franceschi et al., 2017) estimate the hypergradient using automatic differentiation, employing

either forward or reverse mode. Recently, a range of stochastic bilevel methods have been developed and analyzed, using techniques such as Neumann series (Chen et al., 2022; Ji et al., 2021), recursive momentum (Yang et al., 2021; Guo & Yang, 2021), and variance reduction (Yang et al., 2021). Another class of methods formulates the lower-level problem as a value-function-based constraint (Kwon et al., 2023; Wang et al., 2023), enabling the solution of bilevel problems without the need for second-order gradients. A more detailed discussion of related work can be found in the Appendix.

**Adaptive and Tuning-free Algorithms.** Adaptive gradient descent has achieved remarkable success and is widely studied and applied in modern machine learning. Early adaptive algorithms trace back to line search methods, such as backtracking (Goldstein, 1962), and Polyak's stepsize (Polyak, 1969), both of which have inspired numerous recent variants (Armijo, 1966; Bello Cruz & Nghia, 2016; Salzo, 2017; Vaswani et al., 2019; Hazan & Kakade, 2019; Loizou et al., 2021; Orvieto et al., 2022). To reduce the computational cost of line search and avoid the reliance on an unknown optimal function value, the Barzilai-Borwein stepsize (Barzilai & Borwein, 1988; Raydan, 1993; Dai & Liao, 2002) was introduced, drawing inspiration from quasi-Newton schemes. Normalized gradient descent (Cortés, 2006; Nesterov, 2013; Murray et al., 2019) preserves the direction of the gradient while ignoring its magnitude, removing the need for prior knowledge about the function. Duchi et al. (2011) and McMahan & Streeter (2010) pioneered AdaGrad, an adaptive gradient-based method, which proved efficient in solving online convex optimization problems. AdaGrad rapidly evolved for deep learning applications, giving rise to numerous methods, including popular variants like Adam (Diederik, 2014; Reddi et al., 2018; Luo et al., 2019; Xie et al., 2024), RMSprop (Tieleman & Hinton, 2012), and Adadelta (Zeiler, 2012). Specifically, normalized versions of AdaGrad, such as $\text{AdaNGD}_k$ (Levy, 2017), AcceleGrad (Levy et al., 2018), and AdaGrad-Norm (Ward et al., 2020; Xie et al., 2020), introduced adaptive stepsizes that require no problem-specific parameters, making them tuning-free approaches. Recent work by Maladkar et al. (2024) further established lower bounds for minimizing the deterministic gradient $l_1$-norm. Additional methods, such as Lipschitz contact approximation (Malitsky & Mishchenko, 2020) and restart techniques (Marumo & Takeda, 2024), have also been explored. A more comprehensive discussion refers to Khaled & Jin (2024).

**Adaptive and Tuning-free Bilevel Algorithms.** Instead of focusing on single-level problems, Huang & Huang (2021) extended Adam to bilevel optimization algorithms. Fan et al. (2024) introduced adaptive stepsizes for bilevel problems, based on Polyak's stepsize and line search techniques. Most recently, Antonakopoulos et al. (2024) proposed a novel framework that applies adaptive normalized gradient descent to the strongly convex inner problem and AdaGrad-Norm to the nonconvex outer problem, allowing the algorithm to update adaptively with fewer problem-specific parameters.

## 3 ALGORITHM

### 3.1 STANDARD BILEVEL OPTIMIZATION

A key challenge in bilevel optimization is calculating the hypergradient $\nabla\Phi(x)$, which, according to the implicit function theorem, is given by:

$$\nabla\Phi(x) = \nabla_x f\big(x, y^*(x)\big) - \nabla_x \nabla_y g\big(x, y^*(x)\big) \big[\nabla_y \nabla_y g\big(x, y^*(x)\big)\big]^{-1} \nabla_y f\big(x, y^*(x)\big),$$

when $g$ is twice differentiable, $\nabla_y g$ is continuously differentiable and the Hessian $\nabla_y \nabla_y g\big(x, y^*(x)\big)$ is invertible. In practice, $y^*(x)$ is not directly accessible, and one often use an iterative algorithm to obtain an estimate $\hat{y}$ instead. Since computing the Hessian inverse is prohibitively expensive, a more efficient way is to approximate the Hessian-inverse-vector product in the above hypergradient $\nabla\Phi(x)$ by solving the following linear system:

$$\min_v R(x, \hat{y}, v) = \frac{1}{2} v^T \nabla_y \nabla_y g(x, \hat{y}) v - v^T \nabla_y f(x, \hat{y}). \tag{2}$$

Similarly, an iterative algorithm is usually deployed to obtain an approximate solution $\hat{v}$ of the problem in eq. (2). Given the approximates $\hat{y}$ and $\hat{v}$, the variable $x$ is then updated with a hypergradient estimate given by

$$\bar{\nabla} f(x, \hat{y}, \hat{v}) = \nabla_x f(x, \hat{y}) - \nabla_x \nabla_y g(x, \hat{y})\hat{v}. \tag{3}$$

Standard bilevel optimization approaches select the stepsizes for updating $y$, $v$, and $x$ based on problem-specific parameters, such as Lipschitzness and strong convexity parameters (Dagréou et al.,

---

**Algorithm 1** **D**ouble-loop **T**uning-**F**ree **B**ilevel **O**ptimizer (D-TFBO)

---

1: **Input:** initialization $x_0, y_0, v_0, \alpha_0 > 0, \beta_0 > 0, \gamma_0 > 0$, total iteration rounds $T$, and $\epsilon_y = \epsilon_v = \frac{1}{T}$
2: **for** $t = 0, 1, 2, ..., T-1$ **do**
3:      $p = 0, q = 0$, set $y_t^0 = y_{t-1}^{P_{t-1}}, v_t^0 = v_{t-1}^{Q_{t-1}}$ if $t > 0$ and $y_0, v_0$ otherwise
4:      **while** $\|\nabla_y g(x_t, y_t^p)\|^2 > \epsilon_y$ **do**
5:          $\beta_{p+1}^2 = \beta_p^2 + \|\nabla_y g(x_t, y_t^p)\|^2, \ \ y_t^{p+1} = y_t^p - \frac{1}{\beta_{p+1}} \nabla_y g(x_t, y_t^p), \ \ p = p + 1$
6:      **end while**
7:      $P_t = p$
8:      **while** $\|\nabla_v R(x_t, y_t^{P_t}, v_t^q)\|^2 > \epsilon_v$ **do**
9:          $\gamma_{q+1}^2 = \gamma_q^2 + \|\nabla_v R(x_t, y_t^{P_t}, v_t^q)\|^2, \ \ v_t^{q+1} = v_t^q - \frac{1}{\gamma_{q+1}} \nabla_v R(x_t, y_t^{P_t}, v_t^q), \ \ q = q + 1$
10:     **end while**
11:     $Q_t = q$
12:     $\alpha_{t+1}^2 = \alpha_t^2 + \|\bar{\nabla} f(x_t, y_t^{P_t}, v_t^{Q_t})\|^2, \ \ x_{t+1} = x_t - \frac{1}{\alpha_{t+1}} \bar{\nabla} f(x_t, y_t^{P_t}, v_t^{Q_t})$
13: **end for**

---

2022; Ji et al., 2021; 2022). However, these parameters are often difficult to obtain or approximate in practice, leading to significant tuning efforts. This challenge motivates the development of adaptive bilevel optimization algorithms that require less to no tuning.

### 3.2 EXISTING ADAPTIVE BILEVEL OPTIMIZATION METHODS

Among the existing adaptive bilevel methods, the most closely related to this work are Fan et al. (2024) and Antonakopoulos et al. (2024). Fan et al. (2024) utilizes Polyak's stepsizes and a line search to automate the stepsizes for both inner and outer updates. Antonakopoulos et al. (2024) applies AdaNGD (Levy, 2017) to solve the inner problem and updates $x$ using the inverse of cumulative hypergradient norms, where the hypergradient norms are approximated via gradient mapping (Nesterov, 2013) with Fenchel coupling (Mertikopoulos & Sandholm, 2016).

However, these methods are not entirely tuning-free. For instance, the initialization of Polyak's stepsizes in Fan et al. (2024) depends on Lipschitzness parameters, strong convexity parameters, and the optimal lower-level function values. While the line search approach in Fan et al. (2024) bypasses the need for problem-specific parameters, it lacks theoretical convergence guarantees. Similarly, Antonakopoulos et al. (2024) requires the strong convexity parameter for the inner AdaNGD updates.

### 3.3 DOUBLE-LOOP TUNING-FREE BILEVEL OPTIMIZATION- D-TFBO

As shown in Algorithm 1, our D-TFBO method follows a double-loop structure, where two sub-loops of iterations are used to solve the lower-level and linear system problems. In the first sub-loop, we employ the idea of "inverse of cumulative gradient norm" to design the adaptive updates as

$$y_t^{p+1} \leftarrow y_t^p - \frac{1}{\beta_{p+1}} \nabla_y g(x_t, y_t^p), \quad \text{with } \beta_{p+1}^2 = \beta_p^2 + \|\nabla_y g(x_t, y_t^p)\|^2.$$

It can be seen from Algorithm 1 that our D-TFBO algorithm employs a stopping criterion based on the gradient norm: $\|\nabla_y g(x_t, y_t^p)\|^2 \leq \epsilon_y$, where $\epsilon_y$ (**defaulted to** $1/T$ **for convergence analysis**) is independent of problem parameters. The rationale behind this design is that if the stopping criterion is not met (i.e., $\|\nabla_y g(x_t, y_t^p)\|^2 > \epsilon_y$), the accumulation $\beta_p$ of gradient norms continues to increase. This increase causes the stepsize $\frac{1}{\beta_p}$ to decrease to a value at which a descent in the optimality gap is guaranteed. A similar stopping criterion applies to the updates of $v_t^q$ when solving the linear system.

Notably, both sub-loops utilize warm-start variable values but reset the stepsizes at each iteration (cold-start stepsizes). The warm-start variables ensure that the initial point is reasonably close to the optimal solution, while the cold-start scheme guarantees stepsizes to achieve stronger lower bounds. Finally, the update of $x_t$ is based on the accumulation of hypergradient estimates $\bar{\nabla} f(x_t, y_t^{P_t}, v_t^{Q_t})$.

**Remark 1** (Extension to a tunable version with problem-parameter-free tuning coefficients). *Although Algorithm 1 is designed as a tuning-free method, a tunable version with the flexibility to preset hyperparameters can still achieve the same convergence rate and gradient complexity. The stepsizes for $\{x, y, v\}$ can be set as $\{\eta_x/\alpha_t, \eta_y/\beta_p, \eta_v/\gamma_q\}$ and the sub-loops stopping criteria can be set to*

---

**Algorithm 2** **S**ingle-loop **T**uning-**F**ree **B**ilevel **O**ptimizer (S-TFBO)

---
1: **Input:** initialization $x_0, y_0, v_0, \alpha_0 \geq 1, \beta_0 > 0, \gamma_0 > 0$, number of iteration rounds $T$
2: **for** $t = 0, 1, 2, ..., T - 1$ **do**
3:      $\beta_{t+1}^2 = \beta_t^2 + \|\nabla_y g(x_t, y_t)\|^2$
4:      $\gamma_{t+1}^2 = \gamma_t^2 + \|\nabla_v R(x_t, y_t, v_t)\|^2$
5:      $\varphi_{t+1} = \max\{\beta_{t+1}, \gamma_{t+1}\}$
6:      $\alpha_{t+1}^2 = \alpha_t^2 + \|\bar{\nabla} f(x_t, y_t, v_t)\|^2$
7:      $y_{t+1} = y_t - \frac{1}{\beta_{t+1}} \nabla_y g(x_t, y_t)$
8:      $v_{t+1} = v_t - \frac{1}{\varphi_{t+1}} \nabla_v R(x_t, y_t, v_t)$
9:      $x_{t+1} = x_t - \frac{1}{\alpha_{t+1}\varphi_{t+1}} \bar{\nabla} f(x_t, y_t, v_t)$
10: **end for**

---

$\{c_y/T, c_v/T\}$, *where* $\{\eta_x, \eta_y, \eta_v, c_y, c_v\}$ *are configurable hyperparameters that are independent of the problem parameters such as strong-convexity and Lipschitzness parameters.*

### 3.4 SINGLE-LOOP TUNING-FREE BILEVEL OPTIMIZATION- S-TFBO

The two sub-loops in D-TFBO may complicate the implementation, and increase the number of iterations to meet the stopping criterion. In this section, we propose a much simpler fully single-loop tuning-free bilevel optimization method named S-TFBO, as described in Algorithm 2.

The design of stepsizes in Algorithm 2 follows a similar idea in Algorithm 1. In each iteration $t$, we update $\alpha_t, \beta_t, \gamma_t$ as accumulations of gradient norms of $\bar{\nabla} f$, $\nabla_y g$, and $\nabla_v R$ from the previous $t - 1$ iterations. We then update variables $y_t$, $v_t$ and $x_t$ simultaneously with adaptive stepsizes $\left\{\frac{1}{\beta_t}, \frac{1}{\max\{\beta_t, \gamma_t\}}, \frac{1}{\alpha_t \max\{\beta_t, \gamma_t\}}\right\}$. However, the stepsizes for $v$ and $x$ are not straightforward and require careful designs guided by our theoretical analysis, as elaborated below.

**Design of stepsize for $v_t$.** Instead of simply using $\frac{1}{\gamma_t}$, we introduce $\frac{1}{\varphi_t} := \frac{1}{\max\{\beta_t, \gamma_t\}}$ as the stepsize. This adjustment is necessary because $\nabla_v R(x_t, y_t, v_t)$ involves the approximation error $\|y_t - y^*(x_t)\|^2$. Since this error is proportional to $\|\nabla_y g(x_t, y_t)\|^2$, using $\frac{1}{\beta_t}$ helps control this error and prevents it from exploding after accumulation, as validated in our theoretical analysis later.

**Design of stepsize for $x_t$.** Similarly, we use $\frac{1}{\alpha_t \varphi_t}$ as the stepsize for updating $x_t$, where the coupled factor $\frac{1}{\varphi_t}$ is introduced to mitigate the approximation errors from the $y_t$ and $v_t$ updates, leading to a more stable convergence.

**Remark 2** (Extension to a tunable version with problem-parameter-free tuning coefficients.)**.** *Similarly to Remark 1, Algorithm 2 can extend to a tunable version with the same convergence rate and gradient complexity. The stepsizes for $\{x, y, v\}$ can be set as $\{\eta_x/\alpha_t\varphi_t, \eta_y/\beta_t, \eta_v/\varphi_t\}$, where $\{\eta_x, \eta_y, \eta_v\}$ are configurable hyperparameters that are independent of the problem parameters.*

## 4 THEORETICAL ANALYSIS

### 4.1 TECHNICAL CHALLENGES

Compared to existing single-level tuning-free approaches, fully tuning-free bilevel optimization poses unique challenges that have not been addressed well.

- Compared to single-level problems, bilevel problems involve interdependent variable updates, resulting in more complex and interconnected stepsize designs.
- The stages for analyzing each stepsize interact with those of other stepsizes, leading to intertwined multi-stage dynamics across various variables.
- The optimization error of each variable can accumulate (hyper)gradient norms from previous iterations due to the adaptive stepsize designs, complicating the error analysis.

In Section 4.2, we introduce the standard definitions and assumptions. Next, in Section 4.3 and 4.4, we provide a detailed convergence analysis, explaining how we address the above challenges.

### 4.2 Assumptions and Definitions

We make the following definitions and assumptions for outer- and inner-objective functions, as also adopted by Ghadimi & Wang (2018); Chen et al. (2022); Khanduri et al. (2021).

**Definition 1.** *A mapping $f$ is $L$-Lipschitz continuous if $\|f(x_1) - f(x_2)\| \leq L\|x_1 - x_2\|$ for $\forall x_1, x_2$.*

Since the outer objective function $\Phi(x)$ is non-convex, we aim to find an $\epsilon$-accurate stationary point, as defined below.

**Definition 2.** *An output $\bar{x}$ of an algorithm is the $\epsilon$-accurate stationary point of the objective function $\Phi(x)$ if $\|\nabla\Phi(\bar{x})\|^2 \leq \epsilon$, where $\epsilon \in (0, 1)$.*

**Assumption 1.** *Functions $f(x, y)$ and $g(x, y)$ are twice continuously differentiable and $g(x, y)$ is $\mu$ strongly convex w.r.t. $y$, for $x \in \mathbb{R}^{d_x}$, $y \in \mathbb{R}^{d_y}$.*

The following assumption imposes the Lipschitz continuity on the outer and inner functions and their derivatives.

**Assumption 2.** *Function $f(x, y)$ is $L_{f,0}$-Lipschitz continuous; the gradients $\nabla f(x, y)$ and $\nabla g(x, y)$ are $L_{f,1}$ and $L_{g,1}$-Lipschitz continuous, respectively; the second-order gradients $\nabla_x \nabla_y g(x, y)$ and $\nabla_y \nabla_y g(x, y)$ are $L_{g,2}$-Lipschitz continuous.*

Rather than directly using the Lipschitz continuity parameters as bounds on gradients-which can cause dimensional inconsistencies during logarithmic operations-we offer the following remark:

**Remark 3.** *Assumption 2 indicates that there exist parameters $C_{f_x}$, $C_{f_y}$, $C_{g_{xy}}$ and $C_{g_{yy}}$ such that $\|\nabla_x f(x, y)\| \leq C_{f_x}$, $\|\nabla_y f(x, y)\| \leq C_{f_y}$, $\|\nabla_x \nabla_y g(x, y)\| \leq C_{g_{xy}}$ and $\|\nabla_y \nabla_y g(x, y)\| \leq C_{g_{yy}}$.*

**Assumption 3.** *There exists $m \in \mathbb{R}$ such that $\inf_x \Phi(x) \geq m$.*

Next, we present the main convergence theorems for Algorithm 1 and Algorithm 2, along with key propositions that provide insights into these theorems. **A proof sketch is provided in Appendix C.**

### 4.3 Convergence and Complexity Analysis for Algorithm 1

Firstly, we explain the two-stage framework used in our analysis.

**Proposition 1.** *Suppose the iteration rounds to update $\{x, y, v\}$ are $\{T_1, T_2, T_3\}$ and $\{\alpha_t, \beta_t, \gamma_t\}$ are generated by Algorithm 1 or 2. For any $C_\alpha \geq \alpha_0$, $C_\beta \geq \beta_0$, $C_\gamma \geq \gamma_0$, we have*

*(a) either $\alpha_t \leq C_\alpha$ for any $t \leq T_1$, or $\exists k_1 \leq T_1$ such that $\alpha_{k_1} \leq C_\alpha$, $\alpha_{k_1+1} > C_\alpha$;*

*(b) either $\beta_t \leq C_\beta$ for any $t \leq T_2$, or $\exists k_2 \leq T_2$ such that $\beta_{k_2} \leq C_\beta$, $\beta_{k_2+1} > C_\beta$;*

*(c) either $\gamma_t \leq C_\gamma$ for any $t \leq T_3$, or $\exists k_3 \leq T_3$ such that $\gamma_{k_3} \leq C_\gamma$, $\gamma_{k_3+1} > C_\gamma$.*

The analysis for each stepsize is divided into two cases. Let us take (a) as an illustration example. Case 1: the accumulation $\alpha_t$ of gradient norms is bounded by a constant $C_\alpha$ before the end of the iteration. In this case, the average gradient norm square can be bound as $\frac{C_\alpha^2}{T_1}$, which decreases with $T_1$. Case 2: the accumulation $\alpha_{T_1}$ exceeds $C_\alpha$, and hence $\alpha_t$ experiences two stages: in stage 1, $\alpha_t \leq C_\alpha$, and in stage 2, $\alpha_t > C_\alpha$. The error analysis for stage 1 is similar to that of case 1. In stage 2, the stepsizes are small enough to show the gradient norm decreases via a descent lemma.

**Proposition 2.** *Recall that for $t_{th}$ iteration, the sub-loops in Algorithm 1 aim to find $y_t^{P_t}$ and $v_t^{Q_t}$ such that $\|\nabla_y g(x_t, y_t^{P_t})\|^2 \leq \epsilon_y$ and $\|\nabla_v R(x_t, y_t^{P_t}, v_t^{Q_t})\|^2 \leq \epsilon_v$. Under Assumptions 1, 2, we have*

$$\begin{cases} P_t \leq \dfrac{\log(C_\beta^2/\beta_0^2)}{\log(1 + \epsilon_y/C_\beta^2)} + \dfrac{\beta_{max}}{\mu} \log\big(\dfrac{L_{g,1}^2(\beta_{max} - C_\beta)}{\epsilon_y}\big), \\ Q_t \leq \dfrac{\log(C_\gamma^2/\gamma_0^2)}{\log(1 + \epsilon_v/C_\gamma^2)} + \dfrac{\gamma_{max}}{\mu} \log\big(\dfrac{C_{g_{yy}}^2(\gamma_{max} - C_\gamma)}{\epsilon_v}\big), \end{cases}$$

*where $\{C_\beta, C_\gamma\}$, $\beta_{max}$, $\gamma_{max}$ are denied in eq. (5), eq. (22), eq. (29) in the Appendix, respectively.*

Proposition 2 provides upper bounds on $P_t$ and $Q_t$, which correspond to the total numbers of iterations of the two sub-loops. This result is the same as that of the standard AdaGrad-Norm in the strongly convex setting Xie et al. (2020). For the sub-loop for $y$, in Case 1 above, the loop terminates within $\log(C_\beta^2/\beta_0^2)/\log(1+\epsilon_y/C_\beta^2)$ steps; and in Case 2, it takes at most $\log(C_\beta^2/\beta_0^2)/\log(1+\epsilon_y/C_\beta^2)$ steps for stage 1 and at most $\frac{\beta_{\max}}{\mu}\log(L_{g,1}^2(\beta_{\max}-C_\beta)/\epsilon_y)$ for stage 2. For $\epsilon_y$ small enough, it can be seen that $P_t$ takes an order of $1/\epsilon_y$, which is typically larger than those obtained with well-tuned stepsizes. Based on this proposition, we can derive the following convergence results.

**Theorem 1.** *Suppose Assumptions 1,2,3 are satisfied. By setting $\epsilon_y = 1/T$ and $\epsilon_v = 1/T$, the iterates generated by Algorithm 1 satisfy*

$$\frac{1}{T}\sum_{t=0}^{T-1}\|\nabla\Phi(x_t)\|^2 \leq \frac{c_1(C_\alpha+2c_1)}{T} = \mathcal{O}\Big(\frac{1}{T}\Big),$$

*where $C_\alpha$ and $c_1$ are constants defined in eq. (5) and eq. (37), respectively.*

**Corollary 1.** *Under the same setting Theorem 1, to achieve an $\epsilon$-accurate stationary point, Algorithm 1 needs $T = \mathcal{O}(1/\epsilon)$, $\{P_t, Q_t\} = \mathcal{O}(1/\epsilon)$, and the gradient complexity (i.e., the number of gradient evaluations) is $\mathrm{Gc}(\epsilon) = \mathcal{O}(1/\epsilon^2)$.*

Theorem 1 shows that the convergence rate of Algorithm 1 matches that of the standard double-loop bilevel algorithms (Ji et al., 2021; 2022). According to Proposition 2, the sub-loops for updating $y$ and $v$ require $\mathcal{O}(1/\epsilon_y)$ iterations to ensure an $\epsilon_y$-level approximation accuracy, which is worse than the $\mathcal{O}(1)$ results achieved by well-tuned bilevel optimization methods. This is because more iterations are needed to ensure high accuracy in both sub-loops, due to the lack of information about the Lipschitz parameters and strong convexity parameters. Consequently, the gradient complexity of our D-TFBO method is worse than those of well-tuned double-loop methods by an order of $1/\epsilon$.

### 4.4 Convergence and Complexity Analysis for Algorithm 2

Differently from D-TFBO that uses sub-loops to achieve high-accurate $y$ and $v$ iterates, the main challenge for analyzing S-TFBO lies in dealing with the accumulated approximations errors for updating all variables over iterations. In the following propositions, we will show how we upper-bound such cumulative approximation errors and lower-bound the adaptive stepsizes.

First, we present a descent result for the objective function $\Phi(\cdot)$.

**Proposition 3.** *Under Assumptions 1, 2, for Algorithm 2, suppose the total iteration number is $T$. No matter $k_1$ in Proposition 1 exists or not, we always have*

$$\Phi(x_{t+1}) - \Phi(x_t) \leq -\frac{1}{2\alpha_{t+1}\varphi_{t+1}}\|\nabla\Phi(x_t)\|^2 - \frac{1}{2\alpha_{t+1}\varphi_{t+1}}\Big(1-\frac{L_\Phi}{\alpha_{t+1}\varphi_{t+1}}\Big)\|\bar{\nabla}f(x_t,y_t,v_t)\|^2$$
$$+ \frac{\bar{L}^2}{2\mu^2}\Big[1+\frac{2}{\mu^2}\Big(\frac{L_{g,2}C_{f_y}}{\mu}+L_{f,1}\Big)^2\Big]\frac{\|\nabla_y g(x_t,y_t)\|^2}{\alpha_{t+1}\varphi_{t+1}} + \frac{\bar{L}^2}{\mu^2}\frac{\|\nabla_v R(x_t,y_t,v_t)\|^2}{\alpha_{t+1}\varphi_{t+1}}.$$

*If in addition, $k_1$ in Proposition 1 exists, then for $t \geq k_1$, we further have*

$$\Phi(x_{t+1}) - \Phi(x_t) \leq -\frac{1}{2\alpha_{t+1}\varphi_{t+1}}\|\nabla\Phi(x_t)\|^2 - \frac{1}{4\alpha_{t+1}\varphi_{t+1}}\|\bar{\nabla}f(x_t,y_t,v_t)\|^2$$
$$+ \frac{\bar{L}^2}{2\mu^2}\Big[1+\frac{2}{\mu^2}\Big(\frac{L_{g,2}C_{f_y}}{\mu}+L_{f,1}\Big)^2\Big]\frac{\|\nabla_y g(x_t,y_t)\|^2}{\alpha_{t+1}\varphi_{t+1}} + \frac{\bar{L}^2}{\mu^2}\frac{\|\nabla_v R(x_t,y_t,v_t)\|^2}{\alpha_{t+1}\varphi_{t+1}},$$

*where $\bar{L} := \max\big\{2(C_{f_y}^2 L_{g,2}^2/\mu^2 + L_{f,1}^2)^{\frac{1}{2}}, \sqrt{2}C_{g_{yy}}\big\}$.*

It can be seen from Proposition 1 that we derive two distinct forms of descent results for the objective function based on the relationship between $\alpha_{t+1}$ and $C_\alpha$ (whose form is specified in eq. (41) in the appendix). Their key difference is that the second inequality is tighter for the case $t \geq k_1$ by eliminating a term of $\frac{L_\Phi}{2\alpha_{t+1}^2\varphi_{t+1}^2}\|\bar{\nabla}f(x_t,y_t,v_t)\|^2$. Both upper bounds consist of two parts: (i) the approximation errors $\mathcal{O}(\|\nabla_y g(x_t,y_t)\|^2 + \|\nabla_v R(x_t,y_t,v_t)\|^2)/(\alpha_{t+1}\varphi_{t+1})$ induced by the updates on $y$ and $v$; (ii) the descent term $-\|\nabla\Phi(x_t)\|^2/(\alpha_{t+1}\varphi_{t+1})$. It can be seen that there exists a trade-off: a smaller $\alpha_t\varphi_t$ leads to a more notable descent, but larger approximation errors. However, due to the

lack of information about the problem parameters, the value of $\alpha_t \varphi_t$ remains unknown, making it infeasible to determine the optimal trade-off. Instead, we adjust this trade-off based on an overall bound on the descent and approximation errors, derived by telescoping all descent inequalities.

Next, we investigate the upper bounds on the summations of the positive error terms in Proposition 3.

**Proposition 4.** *Under Assumptions 1, 2, for any $0 \leq k_0 < t$, for the positive error terms in Proposition 3, we have the upper bounds in terms of logarithmic functions as*

$$\sum_{k=k_0}^{t} \frac{\|\nabla_y g(x_k, y_k)\|^2}{\beta_{k+1}} \leq a_2 \log(t+1) + b_2, \quad \sum_{k=k_0}^{t} \frac{\|\nabla_v R(x_k, y_k, v_k)\|^2}{\varphi_{k+1}} \leq a_3 \log(t+1) + b_3,$$

*where $a_2$, $b_2$, $a_3$, $b_3$ are defined in eq. (75) in the Appendix.*

**Proposition 5.** *Under Assumptions 1, 2, 3, suppose the total iteration rounds is $T$. For any case in Proposition 1, we have the upper-bound of $\varphi_t$ and $\alpha_t$ in Algorithm 2 as*

$$\varphi_t \leq a_1 \log(t) + b_1, \quad \alpha_t \leq C_\alpha + \big(a_4 \log(t) + b_4 + 4(\Phi(x_0) - \inf_x \Phi(x))\big)\varphi_t,$$

*where $a_1$, $b_1$ are defined in eq. (65) and $a_4$, $b_4$ are defined in eq. (79) in the Appendix.*

Proposition 4 provides the upper bounds on the accumulated positive error terms in Proposition 3, and Proposition 5 shows that the cumulative gradient norms for all variables increase only logarithmically. By rearranging the terms and taking the average, we have the upper bound for the average squared hypergradient norm $\frac{1}{T} \sum_{t=0}^{T-1} \|\nabla \Phi(x_t)\|^2$, establishing the final convergence rate of Algorithm 2, as shown in the following theorem and corollary.

**Theorem 2.** *Suppose Assumptions 1,2,3 are satisfied. The iterates generated by Algorithm 2 satisfy*

$$\frac{1}{T} \sum_{t=0}^{T} \|\nabla \Phi(x_t)\|^2 \leq \frac{1}{2T} \Big[ \big(a_4 \log(T) + b_4 + 4\big(\Phi(x_0) - \inf_x \Phi(x)\big)\big)^2 \big(a_1 \log(T) + b_1\big)^2$$

$$+ C_\alpha \big(a_4 \log(T) + b_4 + 4\big(\Phi(x_0) - \inf_x \Phi(x)\big)\big)\big(a_1 \log(T) + b_1\big) \Big] = \mathcal{O}\Big(\frac{\log^4(T)}{T}\Big),$$

*where $\{C_\alpha, a_1, b_1, a_4, b_4\} = \mathcal{O}(1)$ are defined in eq. (41), eq. (65), eq. (79) in the Appendix.*

**Corollary 2.** *Under the same setting Theorem 2, to achieve an $\epsilon$-accurate stationary point, Algorithm 2 needs $T = \mathcal{O}\big(\frac{1}{\epsilon} \log^4(\frac{1}{\epsilon})\big)$ and the gradient complexity is $\mathrm{Gc}(\epsilon) = \mathcal{O}\big(\frac{1}{\epsilon} \log^4(\frac{1}{\epsilon})\big)$.*

Theorem 2 shows that the proposed Algorithm 2 achieves a convergence rate of $\mathcal{O}(\log^4(T)/T)$ and a gradient complexity of $\mathcal{O}\big(\frac{1}{\epsilon} \log^4(\frac{1}{\epsilon})\big)$, both of which nearly match the results in Ji et al. (2022) of the standard well-tuned bilevel optimization methods up to polylogarithmic factors.

**Remark 4.** *Note that the difference of $\frac{1}{\epsilon}$ in gradient complexity between double-loop and single-loop methods has not been observed in previous works on well-tuned bilevel optimization. This difference stems from the design of the sub-loops. In previous double-loop works, carefully selected stepsizes are used to ensure that the iterates of each sub-loop converge linearly, up to an approximation error caused by the shift in $x$. However, due to the precise control of stepsizes, tuning-free approaches can only guarantee a sub-linear convergence for each sub-loop (as shown in Proposition 2).*

## 5 EXPERIMENTS

In this section, we evaluate the effectiveness of our proposed algorithm on practical applications including regularization selection, data hyper-cleaning (Franceschi et al., 2017), and coreset selection for continual learning (Hao et al., 2024). Our implementation is based on the benchmark provided in Dagréou et al. (2022) and Hao et al. (2024), respectively. *Please refer to Appendix B for more details about practical implementation, experiment configurations, and additional plots.*

### 5.1 REGULARIZATION SELECTION

The selection of regularization can be framed as a bilevel optimization problem, where the inner objective focuses on optimizing the model parameters $\theta$ on the training set $\mathcal{S}_T = \{(d_i^{train}, y_i^{train})\}_{1 \leq i \leq n}$, while the outer objective aims to determine the best regularization term $\lambda$ on the validation set

$\mathcal{S}_V = \{(d_j^{val}, y_j^{val})\}_{1 \leq j \leq m}$. Denote the model parameters by $\theta \in \mathbb{R}^p$ and regularization term by $\lambda \in \mathbb{R}^p$, then the outer and inner problems can be formulated as

$$f(\theta, \lambda) = \frac{1}{m} \sum_{j=1}^{m} l\big((d_j^{val}, y_j^{val}), \theta\big); \quad g(\theta, \lambda) = \frac{1}{n} \sum_{i=1}^{n} l\big((d_i^{train}, y_i^{train}), \theta\big) + \mathcal{R}(\theta, \lambda),$$

where the loss $l((d_i, y_i), \theta) = \log(1 + \exp(-y_i d_i^\top \theta))$, and $\mathcal{R}(\theta, \lambda) = \frac{1}{2} \sum_{k=1}^{p} \exp(\lambda_k) \theta_k^2$ represents the regularization, where each element $\theta_k$ is regularized with strength $\exp(\lambda_k)$. We compare our proposed algorithm with benchmark bilevel algorithms including AmIGO (Arbel & Mairal, 2022), BSA (Ghadimi & Wang, 2018), FSLA (Li et al., 2022), MRBO (Yang et al., 2021), SOBA (Dagréou et al., 2022), StocBiO (Ji et al., 2021), SUSTAIN (Khanduri et al., 2021), TTSA (Hong et al., 2023b), VRBO (Yang et al., 2021) on the Covtype dataset. More details are provided in Appendix B.

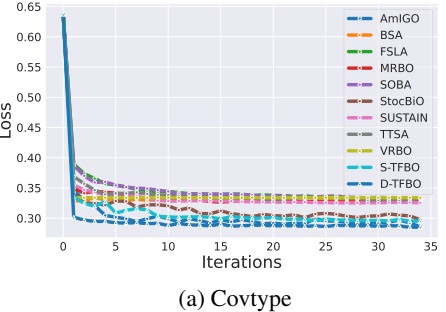 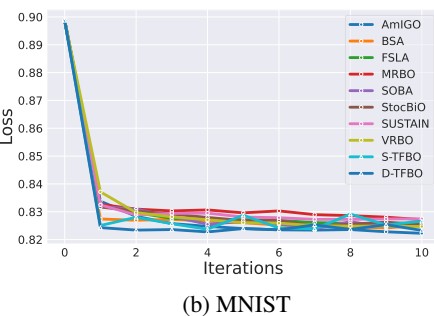

(a) Covtype  (b) MNIST

Figure 1: Comparison with other bilevel methods. (a) Regularization selection on Covtype dataset. (b) Data hyper-cleaning on MNIST dataset.

As shown in Figure 1a, our D-TFBO achieves the fastest convergence rate, while S-TFBO converges slightly more slowly but remains comparable to other well-tuned methods.

## 5.2 DATA HYPER-CLEANING

The training set $\mathcal{S}_T = \{(d_i^{train}, y_i^{train})\}_{1 \leq i \leq n}$ have been corrupted in this scenario, where the label of a data sample could be replaced by a random label with a certain probability $p$. It is important to note that we do not have prior knowledge about which data samples have been corrupted. The objective is to develop a model that can effectively fit the corrupted training set while performing well on the clean validation set $\mathcal{S}_V = \{(d_j^{val}, y_j^{val})\}_{1 \leq j \leq m}$. We conduct experiments on the MNIST dataset, where we aim to learn a set of weights $\lambda$, one for each training sample, in addition to the model parameters $\theta$. Hence, the outer and inner problems are

$$f(\theta, \lambda) = \frac{1}{m} \sum_{j=1}^{m} l\big((d_j^{val}, y_j^{val}), \theta\big); \quad g(\theta, \lambda) = \frac{1}{n} \sum_{i=1}^{n} \sigma(\lambda_i) l\big((d_i^{train}, y_i^{train}), \theta\big) + C\|\theta\|^2,$$

where $\sigma(\cdot)$ is sigmoid function, $C$ is a regularization constant, and loss function $l((d_i, y_i), \theta) = 1/(1 + \exp(-y_i d_i^\top \theta))$. Ideally, we would like the weights to be 0 for the corrupted sample and 1 for the clean sample. More details can be found in Appendix B. We compare the performance with other bilevel optimization methods including AmIGO (Arbel & Mairal, 2022), BSA (Ghadimi & Wang, 2018), FSLA (Li et al., 2022), MRBO (Yang et al., 2021), SOBA (Dagréou et al., 2022), StocBiO (Ji et al., 2021), SUSTAIN (Khanduri et al., 2021), VRBO (Yang et al., 2021). The results presented in Figure 1b demonstrate that our algorithms achieve a convergence rate comparable to other baselines.

## 5.3 CORESET SELECTION FOR CONTINUAL LEARNING

Coreset selection aims to improve training efficiency by selecting a subset of the most informative data samples, which can be used as an approximation of the entire dataset. Thus, the model that minimizes the loss on the coreset can also minimize the loss on the entire dataset. Following the design in Hao et al. (2024), we apply the proposed algorithms to coreset selection for continual learning. The inner problem learns model parameters $\theta$, and the outer problem determines the distribution $\lambda$ ($0 \leq \lambda_{(i)} \leq 1$ and $\|\lambda\|_1 = 1$) over the entire dataset

$$f(\theta, \lambda) = \sum_{i=1}^{n} l_i(\theta) + C\mathcal{R}(\lambda); \quad g(\theta, \lambda) = \sum_{i=1}^{n} \lambda_{(i)} l_i(\theta),$$

Table 1: Results on Split CIFAR100. The best and second best results are in bold and underlined.

| Method | Balanced | | Imbalanced | |
|---|---|---|---|---|
| | $A_T$ | $FGT_T$ | $A_T$ | $FGT_T$ |
| $k$-means features | 57.82±0.69 | 0.070±0.003 | 45.44±0.76 | 0.037±0.002 |
| $k$-means embedding | 59.77±0.24 | 0.061±0.001 | 43.91±0.15 | 0.044±0.001 |
| Uniform Sampling | 58.99±0.54 | 0.074±0.004 | 44.73±0.11 | 0.033±0.007 |
| iCaRL | 60.74±0.09 | 0.044±0.026 | 44.25±2.04 | 0.042±0.019 |
| Grad Matching | 59.17±0.38 | 0.067±0.003 | 45.44±0.64 | 0.038±0.001 |
| GCR | 58.73±0.43 | 0.073±0.013 | 44.48±0.05 | 0.035±0.005 |
| Greedy Coreset | 59.39±0.16 | 0.066±0.017 | 43.80±0.01 | 0.039±0.007 |
| PBCS | 55.64±2.26 | 0.062±0.001 | 39.87±1.12 | 0.076±0.011 |
| BCSR | **61.60±0.14** | 0.051±0.015 | **47.30±0.57** | **0.022±0.005** |
| S-TFBO | 58.90±0.75 | 0.046±0.009 | 45.78±0.70 | 0.036±0.005 |
| D-TFBO | 59.54±0.45 | **0.041±0.005** | 46.68±0.72 | 0.029±0.002 |

Table 2: Experiment results of sensitivity analysis on Split CIFAR100. The initial values refer to the constant learning rates in BCSR or $\alpha_0,\beta_0,\gamma_0$ in S-TFBO and D-TFBO.

| Method | $initial = 2$ | $initial = 4$ | $initial = 6$ | $initial = 8$ | Relative Average Change |
|---|---|---|---|---|---|
| BCSR | 59.42 | 56.25 | 58.75 | 57.55 | 5.8% |
| S-TFBO | 58.85 | 58.55 | 58.69 | 58.47 | **0.4%** |
| D-TFBO | 59.71 | 59.62 | 59.11 | 59.08 | **0.3%** |

where $n$ is the sample size, $C$ is a constant, $\lambda_{(i)}$ is the $i$-th entry. $\mathcal{R}(\lambda) = -\sum_{i=1}^{K} \mathbb{E}(\lambda + \delta z)_{[i]}$ denotes the smoothed top-$K$ regularizer, where $\delta$ is a constant and $z \sim \mathcal{N}(0, 1)$, $\lambda_{[i]}$ is the $i$-th largest component. The regularizer encourages the distribution to have $K$ non-zero entries, corresponding to the size of the selected coreset. Following Zhou et al. (2022), we use the Split CIFAR100 dataset and conduct experiments in the balanced and imbalanced scenarios. We compare the proposed algorithms with various methods, including $k$-means features (Nguyen et al., 2018), $k$-means embedding (Sener & Savarese, 2018), Uniform Sampling, iCaRL (Rebuffi et al., 2017), Grad Matching (Campbell & Broderick, 2019), GCR (Tiwari et al., 2022), Greedy Coreset (Borsos et al., 2020), PBCS (Zhou et al., 2022), and BCSR (Hao et al., 2024), with the last three being bilevel optimization-based methods. We evaluate the performance using the average accuracy and forgetting measure across all tasks after learning task $T$. The former is defined as $A_T = \frac{1}{T} \sum_{i=1}^{T} a_{T,i}$, where $a_{T,i}$ is the test accuracy of the $i$-th task after learning task $T$. The latter is defined as $FGT_T = \frac{1}{T} \sum_{i=1}^{T} [\max_{j \in 1, \cdots, T-1} (a_{j,i} - a_{T,i})]$. The results are shown in Table 1. Each experiment is repeated three times and the average is reported. It can be observed that our D-TFBO achieves the best $FGT_T$ under the balanced setting and the second-best performance under the imbalanced setting.

**Sensitivity analysis w.r.t. different initial learning rates.** The tuning-free design provides another benefit. The proposed algorithms demonstrate more robustness compared to the Hao et al. (2024). We conduct a simple sensitivity analysis under the balanced setting, regarding the learning rates in the inner and outer loops. Specifically, we set the initial learning rates in Hao et al. (2024) and $\alpha_0, \beta_0,$ $\gamma_0$ in S-TFBO and D-TFBO for the inner and outer loops to $\{2, 4, 6, 8\}$, where the original values are set to 5. We run one experiment for each learning rate. Further, we compare the changes in average accuracy $A_T$. We also compute the average and report the relative change compared to the results presented in Table 1.

## 6 CONCLUSION

We introduce two fully tuning-free bilevel optimization algorithms, D-TFBO and S-TFBO. Both methods adaptively update stepsizes without requiring prior knowledge of problem parameters, while achieving convergence rates comparable to their well-tuned counterparts. The experimental results show that our tuning-free design performs comparably to existing well-tuned methods and is more robust to initial stepsizes. We anticipate that the proposed algorithms and the developed analysis can be extended to the stochastic setting, and the proposed algorithms may be applied to other applications such as meta-learning, few-shot learning, and fair machine learning.

ACKNOWLEDGEMENTS

Yifan Yang, Hao Ban and Kaiyi Ji were partially supported by NSF grants CCF-2311274 and ECCS-2326592. Shiqian Ma was supported in part by NSF grants CCF-2311275, ECCS-2326591 and ONR grant N00014-24-1-2705.

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

# Supplementary material

## A  ADDITIONAL DISCUSSION ON RELATED WORK

### A.1  COMPARISON WITH THE EXISTING BILEVEL METHODS

We compare the proposed D-TFBO and S-TFBO with standard bilevel methods in Table 3. Notably, both D-TFBO and S-TFBO achieve a (nearly) equivalent convergence rate to other methods without requiring additional tuning.

| **Algorithm** | Sub-loop $K$ | Convergence Rate $T$ | Gradient Complexity | Hyperparameters to Tune |
|---|---|---|---|---|
| AID-BiO (Ji et al., 2021) | $\mathcal{O}(1)$ | $\mathcal{O}(\frac{1}{\epsilon})$ | $\mathcal{O}(\frac{1}{\epsilon})$ | 5 |
| ITD-BiO (Ji et al., 2021) | $\mathcal{O}(\log(\frac{1}{\epsilon}))$ | $\mathcal{O}(\frac{1}{\epsilon})$ | $\mathcal{O}(\frac{1}{\epsilon}\log(\frac{1}{\epsilon}))$ | 3 |
| SOBA (Dagréou et al., 2022) | $\mathcal{O}(1)$ | $\mathcal{O}(\frac{1}{\epsilon})$ | $\mathcal{O}(\frac{1}{\epsilon})$ | 3 |
| D-TFBO (this paper) | $\mathcal{O}(\frac{1}{\epsilon})$ | $\mathcal{O}(\frac{1}{\epsilon})$ | $\mathcal{O}(\frac{1}{\epsilon^2})$ | 0 |
| S-TFBO (this paper) | $\mathcal{O}(1)$ | $\mathcal{O}(\frac{1}{\epsilon}\log^4(\frac{1}{\epsilon}))$ | $\mathcal{O}(\frac{1}{\epsilon}\log^4(\frac{1}{\epsilon}))$ | 0 |

Table 3: Comparison of the proposed tuning-free methods with existing standard bilevel optimization methods.

### A.2  THE NECESSITY OF THE ITERATION NUMBER $T$

It is possible to eliminate the dependence on the knowledge of iteration $T$ in S-TFBO. In detail, we can modify the "for" loop in S-TFBO (Algorithm 2) to a "repeat until convergence" structure, as in Marumo & Takeda (2024), and this allows S-TFBO to converge to any targeted $\epsilon$-stationary point without the knowledge of total iteration number $T$. However, D-TFBO (Algorithm 1) requires the sub-loop stopping criteria to be set as $\epsilon_y = \mathcal{O}(\frac{1}{T})$, $\epsilon_v = \mathcal{O}(\frac{1}{T})$, which depends on prior knowledge of $T$. Thus, D-TFBO may not be feasible.

### A.3  SUPPLEMENTARY RELATED WORK ON BILEVEL OPTIMIZATION

Initially introduced by Bracken & McGill (1973), bilevel optimization has been extensively studied for decades. Early works (Hansen et al., 1992; Shi et al., 2005; Gould et al., 2016; Sinha et al., 2017) solved the bilevel problem from a constrained optimization perspective. More recently, gradient-based bilevel methods have gained significant attention for their efficiency and effectiveness in addressing machine learning problems. Among them, approaches based on Approximate Implicit Differentiation (AID) (Domke, 2012; Liao et al., 2018; Pedregosa, 2016; Lorraine et al., 2020; Grazzi et al., 2020; Ji et al., 2021; Arbel & Mairal, 2022; Hong et al., 2023b) exploit the implicit derivation of the hypergradient, approximating it by solving a linear system.

On the other hand, approaches based on Iterative Differentiation (ITD) (Maclaurin et al., 2015; Franceschi et al., 2017; Finn et al., 2017; Shaban et al., 2019; Grazzi et al., 2020) estimate the hypergradient by employing automatic differentiation, utilizing either forward or reverse mode.

A series of stochastic bilevel approaches has been developed and analyzed recently, utilizing Neumann series (Chen et al., 2022; Ji et al., 2021; Arbel & Mairal, 2022), recursive momentum (Yang et al., 2021; Huang & Huang, 2021; Guo & Yang, 2021), and variance reduction (Yang et al., 2021; Dagréou et al., 2022), etc. For the lower-level problem with multiple solutions, several approaches were proposed based on upper- and lower-level gradient aggregation (Sabach & Shtern, 2017; Liu et al., 2020; Li et al., 2020), barrier types of regularization (Liu et al., 2021a; 2022), penalty-based formulations (Shen & Chen, 2023), primal-dual techniques (Sow et al., 2022), and dynamic system-based methods (Liu et al., 2021b). Another class of approaches formulated the lower-level problem as a value-function-based constraint (Kwon et al., 2023; Wang et al., 2023) to solve bilevel problems without second-order gradients.

# B SPECIFICATIONS OF EXPERIMENTS

## B.1 PRACTICAL GUIDELINE

In practice, D-TFBO ensures higher accuracy, as shown in most of our experiments but is harder to implement and the sub-loops cause the waiting time to update $x$; S-TFBO achieves slightly worse performance but it has advantages such as simple implementation and no waiting time for updating $x$.

As practical guidance for practitioners, D-TFBO is well-suited for scenarios requiring high accuracy, while S-TFBO is preferable for its simpler implementation and no waiting time when updating the objective variable.

## B.2 PRACTICAL IMPLEMENTATION

For regularization selection and data hyper-cleaning, we use the benchmark provided in Dagréou et al. (2022). For coreset selection, we use the codebase from Hao et al. (2024). We implement D-TFBO using "for loops" as an approximation, since the magnitude of $\|\nabla_v R(x, y, v)\|$ in Algorithm 1 varies across different experiments. Specifically, the number of loops for updating $y$ and $v$ in regularization selection and data hyper-cleaning are both set to 10, while the numbers of loops for updating $y$ and $v$ in coreset selection are 5 and 3, respectively.

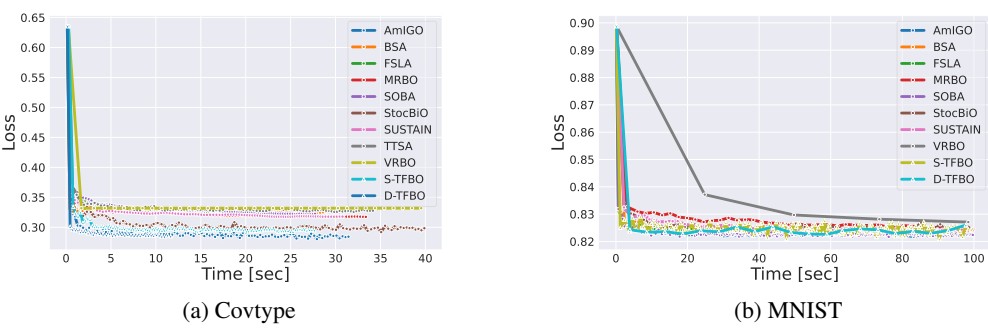

(a) Covtype           (b) MNIST

Figure 2: Comparison of running time on regularization selection and data hyper-cleaning.

## B.3 CONFIGURATION

We adopt the default configuration for regularization selection and data hyper-cleaning. The batch size is 64. The maximum iterations are 2048 and 512, respectively. The data corruption ratio in hyper-cleaning is 0.1. For coreset selection, we also use the default configuration except for the leaning rates, due to the tuning-free design. The $\alpha_0$, $\beta_0$, and $\gamma_0$ values are set to 5.

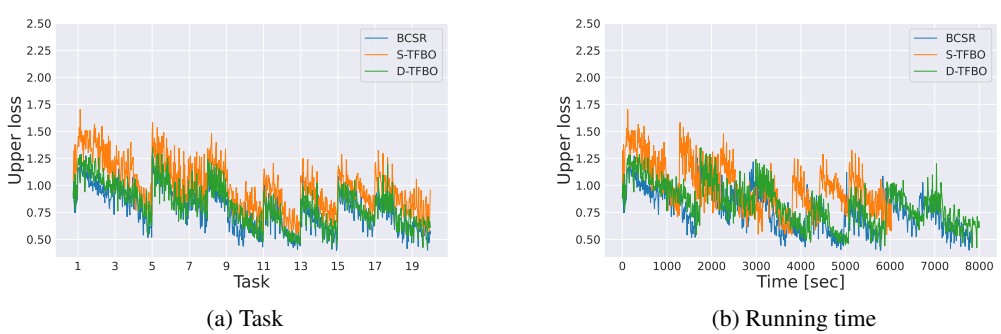

(a) Task           (b) Running time

Figure 3: The upper loss of coreset selection.

### B.4 ADDITIONAL RESULTS

For regularization and data hyper-cleaning, we also present the loss curves regarding running time in Figure 2. Our methods exhibit a faster running time than other baselines on the Covtype dataset. For coreset selection, we adopt the default settings of initial values, such as the constant learning rates in BCSR and $\alpha_0$, $\beta_0$, $\gamma_0$ in S-TFBO and D-TFBO, all set to 5. We re-ran the methods on Split-CIFAR100 under the balanced scenarios and recorded the loss and running time. The loss curves regarding task and running time are shown in Figure 3. Following Hao et al. (2024), we plot the loss value every 5 mini-batches. The loss decreases gradually but increases when a new task is encountered. Additionally, S-TFBO converges faster than BCSR (Hao et al., 2024), while D-TFBO performs comparably to BCSR (Hao et al., 2024).

## C PROOF SKETCH

The proofs of Propositions 1, 2, 3, 4 and 5 can be found in Lemma 4, 9, 11, 15,17, respectively. In this section, we present a high level proof sketch that outlines the convergence and gradient complexity analysis of Algorithm 1 and Algorithm 2, emphasizing the key challenges and our technical innovations.

**Proof sketch of Algorithm 1:**

**Step 1:** We first discuss the two-stage framework in our problem in Lemma 4 and we develop two forms of descent lemma of the objective function in Lemma 7 based on the two stages of $\alpha_t$ in Lemma 4.

**Step 2:** We developed upper bounds of $\alpha_t$ under the two stages in Lemma 4.

**Step 3:** We provide the maximum iteration numbers for the sub-loops approximating $y^*(x_t)$ and $v^*(x_t)$.

**Step 4:** Combining the results in Step 1 and Step 2, we telescope and take the average of the inequalities in the descent lemma of the objective function, then we obtain the convergence rate.

**Step 5:** Combining the maximum iteration numbers in Step 3 and convergence rate in Step 4, we obtain the gradient computation complexity to find $\epsilon$-stationary point. Then the proof is complete.

**Proof sketch of Algorithm 2:**

**Step 1:** We first discuss the two-stage framework in our problem in Lemma 4 and we develop two forms of descent lemma of the objective function in Lemma 11 based on the two stages of $\alpha_t$ in Lemma 4.

**Step 2:** We develop a rough upper bound of two important components in the descent lemma in Lemma 11: $\sum_{k=k_2}^{t} \frac{\|\nabla_y g(x_k, y_k)\|^2}{\beta_{k+1}}$ and $\sum_{k=k_3}^{t} \frac{\|\nabla_v R(x_k, y_k, v_k)\|^2}{\varphi_{k+1}}$, where $k_2$ and $k_3$ represents the second stage in Lemma 4.

**Step 3:** Following the results in Step 2 and the upper bound of $v_t$ in Lemma 10, we obtain a two-way relationship between $\varphi_{t+1}$ and $\sum_{k=0}^{t} \frac{\|\nabla f(x_k, y_k, v_k)\|^2}{\alpha_{k+1}^2}$, which further indicates the logarithmic upper bounds of both terms in Lemma 15 and Lemma 16, respectively.

**Step 4:** Incorporating the results from Step 3 into the rough bounds from Step 2, we can also obtain the logarithmic upper bounds of $\sum_{k=k_2}^{t} \frac{\|\nabla_y g(x_k, y_k)\|^2}{\beta_{k+1}}$ and $\sum_{k=k_3}^{t} \frac{\|\nabla_v R(x_k, y_k, v_k)\|^2}{\varphi_{k+1}}$ in Lemma 16.

**Step 5:** We rearrange the terms in Lemma 11 and incorporate in the results in Step 4, we obtain two forms of the upper bound of $\alpha_t$ in Lemma 17.

**Step 6:** Combining the results in Steps 3, 4, 5, we telescope and take the average of the inequalities in the descent lemma of the objective function, then we obtain the convergence rate.

**Step 7:** Without sub-loops, via the convergence rate in Step 6, we can directly obtain the gradient computation complexity to find $\epsilon$-stationary point. Then the proof is complete.

# D  PROOFS OF PRELIMINARY LEMMAS

**Lemma 1** (Ward et al. (2020) Lemma 3.2). *For any non-negative $a_1, ..., a_T$, and $a_1 \geq 1$, we have*

$$\sum_{l=1}^{T} \frac{a_l}{\sum_{i=1}^{l} a_i} \leq \log \left( \sum_{l=1}^{T} a_l \right) + 1. \tag{4}$$

**Lemma 2.** *Under Assumptions 1, 2, we have basic properties as follows:*

(a) $\Phi(x)$ *is $L_\Phi$-smooth w.r.t $x$, where* $L_\Phi := \left( L_{f,1} + \frac{L_{g,2} C_{f_y}}{\mu} \right) \left( 1 + \frac{C_{g_{xy}}}{\mu} \right)^2$;

(b) $y^*(x)$ *is $L_y$-Lipschitz continuous w.r.t. $x$, where* $L_y := \frac{C_{g_{xy}}}{\mu}$;

(c) *the gradient estimator $\bar{\nabla} f(x, y, v)$ is $(L_{g,2}\|v\| + L_{f,1})$ -Lipschitz continuous w.r.t. $(x, y)$, and $L_{g,1}$-Lipschitz continuous w.r.t. $v$;*

(d) $\bar{\nabla} f(x, y, v)$ *can be bounded as* $\|\bar{\nabla} f(x, y, v)\| \leq C_{g_{xy}}\|v\| + C_{f_x}$.

*Proof.* The proof of (a) and (b) can refer to Ghadimi & Wang (2018). For (c), under Assumption 2, we have

$$\|\bar{\nabla} f(x_1, y_1, v) - \bar{\nabla} f(x_2, y_2, v)\| \leq \|\nabla_x \nabla_y g(x_1, y_1) - \nabla_x \nabla_y g(x_2, y_2)\| \cdot \|v\|$$
$$+ \|\nabla_x f(x_1, y_1) - \nabla_x f(x_2, y_2)\|$$
$$\leq (L_{g,2}\|v\| + L_{f,1})(\|x_1 - x_2\| + \|y_1 - y_2\|)$$
$$\|\bar{\nabla} f(x, y, v_1) - \bar{\nabla} f(x, y, v_2)\| \leq \|\nabla_x \nabla_y g(x, y)\| \cdot \|v_1 - v_2\| \leq L_{g,1}\|v_1 - v_2\|.$$

By Assumption 2 and Remark 3, we can easily prove (d) as

$$\|\bar{\nabla} f(x, y, v)\| \leq \|\nabla_x \nabla_y g(x, y)\| \cdot \|v\| + \|\nabla_x f(x, y)\| \leq C_{g_{xy}}\|v\| + C_{f_x}.$$

Then the proof is complete. $\qquad\square$

**Lemma 3.** *Under Assumptions 1, 2, we have basic properties of linear system function $R$ in eq. (2) as follows:*

(a) $R(x, y, v)$ *is $\mu$-strongly convex and $C_{g_{yy}}$-smooth w.r.t. $v$;*

(b) $\nabla_v R(x, y, v)$ *is $(L_{g,2}\|v\| + L_{f,1})$-Lipschitz continuous w.r.t. $(x, y)$;*

(c) $\nabla_v R(x, y, v)$ *can be bounded as* $\|\nabla_v R(x, y, v)\| \leq C_{g_{yy}}\|v\| + C_{f_y}$;

(d) $v^*(x)$ *in eq. (2) can be bounded as* $\|v^*(x)\| \leq \frac{C_{f_y}}{\mu}$, *and $\hat{v}^*(x, y) := \arg\min_v R(x, y, v)$ can also be bounded as* $\|\hat{v}^*(x, y)\| \leq \frac{C_{f_y}}{\mu}$;

(e) $v^*(x)$ *is $L_v$-Lipschitz continuous w.r.t. $x$ and $\hat{v}^*(x, y)$ is $\bar{L}_v$-Lipschitz continuous w.r.t. $y$, where* $L_v := \left( \frac{L_{f,1}}{\mu} + \frac{C_{f_y} L_{g,2}}{\mu^2} \right)(1 + L_y)$ *and* $\bar{L}_v := \frac{L_{f,1}}{\mu} + \frac{C_{f_y} L_{g,2}}{\mu^2}$.

*Proof.* First of all, since $\nabla_v \nabla_v R(x, y, v) = \nabla_y \nabla_y g(x, y)$, we know $\mu I \preceq \nabla_y \nabla_y g(x, y)$. Thus, according to Assumption 1,2, we have

$$\|\nabla_v \nabla_v R(x, y, v_1) - \nabla_v \nabla_v R(x, y, v_2)\| \leq \|\nabla_y \nabla_y g(x, y)\|\|v_1 - v_2\| \leq C_{g_{yy}}\|v_1 - v_2\|.$$

Then (a) is proved. Next, by using Lipschitz continuity in Assumption 2, we have

$$\|\nabla_v R(x_1, y_1, v) - \nabla_v R(x_2, y_2, v)\| \leq \|\nabla_y \nabla_y g(x_1, y_1) - \nabla_y \nabla_y g(x_2, y_2)\| \cdot \|v\|$$
$$+ \|\nabla_y f(x_1, y_1) - \nabla_y f(x_2, y_2)\|$$
$$\leq (L_{g,2}\|v\| + L_{f,1})(\|x_1 - x_2\| + \|y_1 - y_2\|).$$

Then (b) is proved. By Assumption 2, we can easily prove (c) as

$$\|\nabla_v R(x,y,v)\| \le \|\nabla_y \nabla_y g(x,y)\| \cdot \|v\| + \|\nabla_y f(x,y)\| \le C_{g_{yy}}\|v\| + C_{f_y}.$$

Next, for $\hat{v}^*(x,y)$, we have

$$\nabla_v R\big(x,y,\hat{v}^*(x,y)\big) = \nabla_y \nabla_y g(x,y)\hat{v}^*(x,y) - \nabla_y f(x,y) = 0,$$

which indicates that

$$\|\hat{v}^*(x,y)\| = \big\|\big[\nabla_y \nabla_y g(x,y)\big]^{-1}\nabla_y f(x,y)\big\| \le \big\|\big[\nabla_y \nabla_y g(x,y)\big]^{-1}\big\| \cdot \|\nabla_y f(x,y))\| \le \frac{C_{f_y}}{\mu}.$$

Since $v^*(x)$ is a special case as $v^*(x) = \hat{v}^*(x,y^*(x))$, (d) is proved. The proof of the first part of (e) can refer to Lemma 4 in Yang et al. (2024); for the second part, we have

$$\|\hat{v}^*(x,y_1) - \hat{v}^*(x,y_2)\|$$
$$= \big\|[\nabla_y \nabla_y g(x,y_1)]^{-1}\nabla_y f(x,y_1) - [\nabla_y \nabla_y g(x,y_2)]^{-1}\nabla_y f(x,y_2)\big\|$$
$$\le \big\|[\nabla_y \nabla_y g(x,y_1)]^{-1}\big(\nabla_y f(x,y_1) - \nabla_y f(x,y_2)\big)\big\|$$
$$\quad + \big\|\big([\nabla_y \nabla_y g(x,y_1)]^{-1} - [\nabla_y \nabla_y g(x,y_2)]^{-1}\big)\nabla_y f(x,y_2)\big\|$$
$$\le \frac{L_{f,1}}{\mu}\|y_1 - y_2\| + C_{f_y}\big\|\big([\nabla_y \nabla_y g(x,y_1)]^{-1}\big(\nabla_y \nabla_y g(x,y_2) - \nabla_y \nabla_y g(x,y_1)\big)[\nabla_y \nabla_y g(x,y_2)]^{-1}\big)\big\|$$
$$\le \Big(\frac{L_{f,1}}{\mu} + \frac{C_{f_y}L_{g,2}}{\mu^2}\Big)\|y_1 - y_2\|.$$

Thus, the second part of (e) is proved and the proof of Lemma 3 is complete. □

**Lemma 4.** *Suppose the iteration rounds to update $\{x,y,v\}$ are $\{T_1,T_2,T_3\}$ and $\{\alpha_t,\beta_t,\gamma_t\}$ are generated by Algorithm 1 or 2. For any $C_\alpha \ge \alpha_0$, $C_\beta \ge \beta_0$, $C_\gamma \ge \gamma_0$, we have*

(a) *either $\alpha_t \le C_\alpha$ for any $t \le T_1$, or $\exists k_1 \le T_1$ such that $\alpha_{k_1} \le C_\alpha$, $\alpha_{k_1+1} > C_\alpha$;*

(b) *either $\beta_t \le C_\beta$ for any $t \le T_2$, or $\exists k_2 \le T_2$ such that $\beta_{k_2} \le C_\beta$, $\beta_{k_2+1} > C_\beta$;*

(c) *either $\gamma_t \le C_\gamma$ for any $t \le T_3$, or $\exists k_3 \le T_3$ such that $\gamma_{k_3} \le C_\gamma$, $\gamma_{k_3+1} > C_\gamma$.*

*Proof.* The proof resembles the Lemma 4.1 in Ward et al. (2020). Here we only prove part (a), and the other two are similar. Note that if $\alpha_{T_1} > C_\alpha$, then there must exist $k_1 \le T_1$ such that $\alpha_{k_1} \le C_\alpha$, $\alpha_{k_1+1} > C_\alpha$, because $C_\alpha \ge \alpha_0$ and the sequence $\{\alpha_k\}$ is monotonically increasing. Otherwise, we have $\alpha_t \le \alpha_{T_1} \le C_\alpha$ for any $t \le T_1$. This completes the proof of part (a). □

# E   PROOF OF THEOREM 1

We define some notation for convenience before proving Theorem 1.

## E.1   NOTATION

Here, we define the following constants as thresholds for parameters $\beta_p$, $\gamma_q$, $\alpha_t$ in Algorithm 1 as

$$C_\alpha := \max\{2L_\Phi, \alpha_0\}, \quad C_\beta := \max\{L_{g,1}, \beta_0\}, \quad C_\gamma := \max\{C_{g_{yy}}, \gamma_0\}. \tag{5}$$

## E.2   PROOFS OF PRELIMINARY LEMMAS

**Lemma 5.** *Under Assumptions 1, 2, for any $t \geq 0$ in Algorithm 1, we have*

$$\|y_t^{P_t} - y^*(x_t)\|^2 \leq \frac{\epsilon_y}{\mu^2}, \quad \|v_t^{Q_t} - \hat{v}^*(x_t, y_t^{P_t})\|^2 \leq \frac{\epsilon_v}{\mu^2},$$

*where $\epsilon_y$ and $\epsilon_v$ are sub-loop stopping criteria in Algorithm 1.*

*Proof.* For the $k_{th}$ iteration, according to the stop criteria of the sub-loops, we have

$$\|\nabla_y g(x_t, y_t^{P_t})\|^2 \leq \epsilon_y, \quad \|\nabla_v R(x_t, y_t^{P_t}, v_t^{Q_t})\|^2 \leq \epsilon_v.$$

By using Assumptions 1,2, we have

$$\|y_t^{P_t} - y^*(x_t)\|^2 \leq \frac{1}{\mu^2}\|\nabla_y g(x_t, y_t^{P_t}) - \nabla_y g(x_t, y^*(x_t))\|^2 \leq \frac{\epsilon_y}{\mu^2},$$

$$\|v_t^{Q_t} - \hat{v}^*(x_t, y_t^{P_t})\|^2 \leq \frac{1}{\mu^2}\|\nabla_v R(x_t, y_t^{P_t}, v_t^{Q_t}) - \nabla_v R(x_t, y_t^{P_t}, \hat{v}^*(x_t, y_t^{P_t}))\|^2 \leq \frac{\epsilon_v}{\mu^2},$$

since $\|\nabla_y g(x_t, y^*(x_t))\|^2 = 0$ and $\|\nabla_v R(x_t, y_t^{P_t}, \hat{v}^*(x_t, y_t^{P_t}))\|^2 = 0$. Thus, the proof is complete. $\square$

**Lemma 6.** *Under Assumptions 1, 2, for any $t \geq 0$ in Algorithm 1, we have $\|\bar{\nabla} f(x_t, y_t^{P_t}, v_t^{Q_t})\|^2 \leq C_f^2$, where $C_f := \left(\frac{2C_{g_{xy}}^2 \epsilon_v}{\mu^2} + \frac{4C_{g_{xy}}^2 C_{f_y}^2}{\mu^2} + 4C_{f_y}^2\right)^{\frac{1}{2}}$.*

*Proof.* For the $k_{th}$ iteration, we have

$$\|\bar{\nabla} f(x_t, y_t^{P_t}, v_t^{Q_t})\|^2$$
$$\leq 2\|\bar{\nabla} f(x_t, y_t^{P_t}, v_t^{Q_t}) - \bar{\nabla} f(x_t, y_t^{P_t}, \hat{v}^*(x_t, y_t^{P_t}))\|^2 + 2\|\bar{\nabla} f(x_t, y_t^{P_t}, \hat{v}^*(x_t, y_t^{P_t}))\|^2$$
$$= 2\|\nabla_x \nabla_y g(x_t, y_t^{P_t})(v_t^{Q_t} - \hat{v}^*(x_t, y_t^{P_t}))\|^2 + 2\|\nabla_x \nabla_y g(x_t, y_t^{P_t})\hat{v}^*(x_t, y_t^{P_t}) - \nabla_y f(x_t, y_t^{P_t})\|^2$$
$$\leq 2\|\nabla_x \nabla_y g(x_t, y_t^{P_t})\|^2 \cdot \|v_t^{Q_t} - \hat{v}^*(x_t, y_t^{P_t})\|^2 + 2\|\nabla_x \nabla_y g(x_t, y_t^{P_t})\hat{v}^*(x_t, y_t^{P_t}) - \nabla_y f(x_t, y_t^{P_t})\|^2$$
$$\overset{(a)}{\leq} \frac{2C_{g_{xy}}^2 \epsilon_v}{\mu^2} + \frac{4C_{g_{xy}}^2 C_{f_y}^2}{\mu^2} + 4C_{f_y}^2,$$

where (a) uses Assumption 1, Remark 3, Lemma 3 and Lemma 5. Then, the proof is complete. $\square$

## E.3   DESCENT IN OBJECTIVE FUNCTION

**Lemma 7.** *Under Assumptions 1, 2, for Algorithm 1, suppose the total iteration number is $T$. No matter $k_1$ in Lemma 4 exists or not, we always have*

$$\Phi(x_{t+1}) \leq \Phi(x_t) - \frac{1}{2\alpha_{t+1}}\|\nabla \Phi(x_t)\|^2 - \frac{1}{2\alpha_{t+1}}\left(1 - \frac{L_\Phi}{2\alpha_{t+1}}\right)\|\bar{\nabla} f(x_t, y_t^{P_t}, v_t^{Q_t})\|^2 + \frac{\epsilon'}{2\alpha_{t+1}}. \tag{6}$$

*If in addition, $k_1$ in Lemma 4 exists, then for $t \geq k_1$, we further have*

$$\Phi(x_{t+1}) \leq \Phi(x_t) - \frac{1}{2\alpha_{t+1}}\|\nabla \Phi(x_t)\|^2 - \frac{1}{4\alpha_{t+1}}\|\bar{\nabla} f(x_t, y_t^{P_t}, v_t^{Q_t})\|^2 + \frac{\epsilon'}{2\alpha_{t+1}}, \tag{7}$$

*where $\epsilon' := \frac{\bar{L}^2}{\mu^2}(\epsilon_y + \epsilon_v)$ and $\bar{L} := \max\{2\left(\frac{C_{f_y}^2 L_{g,2}^2}{\mu^2} + L_{f,1}^2 + C_{g_{yy}}^2 \bar{L}_v^2\right)^{\frac{1}{2}}, \sqrt{2}C_{g_{yy}}\}$.*

*Proof.* From Lemma 2, we have $\Phi(x)$ is $L_\Phi$-smooth. So we can apply the descent lemma to $\Phi$ as

$$
\begin{aligned}
\Phi(x_{t+1}) \leq & \Phi(x_t) + \langle \nabla\Phi(x_t), x_{t+1} - x_t \rangle + \frac{L_\Phi}{2}\|x_{t+1} - x_t\|^2 \\
= & \Phi(x_t) - \frac{1}{\alpha_{t+1}}\langle \nabla\Phi(x_t), \bar{\nabla} f(x_t, y_t^{P_t}, v_t^{Q_t})\rangle + \frac{L_\Phi}{2\alpha_{t+1}^2}\|\bar{\nabla} f(x_t, y_t^{P_t}, v_t^{Q_t})\|^2 \\
= & \Phi(x_t) - \frac{1}{2\alpha_{t+1}}\|\nabla\Phi(x)\|^2 - \frac{1}{2\alpha_{t+1}}\|\bar{\nabla} f(x_t, y_t^{P_t}, v_t^{Q_t})\|^2 \\
& + \frac{1}{2\alpha_{t+1}}\|\nabla\Phi(x_t) - \bar{\nabla} f(x_t, y_t^{P_t}, v_t^{Q_t})\|^2 + \frac{L_\Phi}{2\alpha_{t+1}^2}\|\bar{\nabla} f(x_t, y_t^{P_t}, v_t^{Q_t})\|^2, \quad (8)
\end{aligned}
$$

where the approximation error

$$
\begin{aligned}
& \|\nabla\Phi(x_t) - \bar{\nabla} f(x_t, y_t^{P_t}, v_t^{Q_t})\|^2 \\
= & \|\bar{\nabla} f(x_t, y^*(x_t), v^*(x_t)) - \bar{\nabla} f(x_t, y_t^{P_t}, v_t^{Q_t})\|^2 \\
\leq & 2\|\bar{\nabla} f(x_t, y^*(x_t), v^*(x_t)) - \bar{\nabla} f(x_t, y_t^{P_t}, v^*(x_t))\|^2 \\
& + 2\|\bar{\nabla} f(x_t, y_t^{P_t}, v^*(x_t)) - \bar{\nabla} f(x_t, y_t^{P_t}, v_t^{Q_t})\|^2 \\
\leq & 4\|\nabla_y\nabla_y g(x_t, y^*(x_t))v^*(x_t) - \nabla_y\nabla_y g(x_t, y_t^{P_t})v^*(x_t)\|^2 \\
& + 4\|\nabla_y f(x_t, y^*(x_t)) - \nabla_y f(x_t, y_t^{P_t})\|^2 + 2\|\nabla_y\nabla_y g(x_t, y_t^{P_t})(v^*(x_t) - v_t^{Q_t})\|^2 \\
\overset{(a)}{\leq} & 4\Big(\frac{C_{f_y}^2 L_{g,2}^2}{\mu^2} + L_{f,1}^2\Big)\|y_t^{P_t} - y^*(x_t)\|^2 + 2C_{g_{yy}}^2\|v_t^{Q_t} - v^*(x_t)\|^2 \\
\leq & 4\Big(\frac{C_{f_y}^2 L_{g,2}^2}{\mu^2} + L_{f,1}^2\Big)\|y_t^{P_t} - y^*(x_t)\|^2 + 4C_{g_{yy}}^2\|v_t^{Q_t} - \hat{v}^*(x_t, y_t^{P_t})\|^2 + 4C_{g_{yy}}^2\|\hat{v}^*(x_t, y_t^{P_t}) - v^*(x_t)\|^2 \\
\overset{(b)}{\leq} & 4\Big(\frac{C_{f_y}^2 L_{g,2}^2}{\mu^2} + L_{f,1}^2 + C_{g_{yy}}^2\bar{L}_v^2\Big)\|y_t^{P_t} - y^*(x_t)\|^2 + 4C_{g_{yy}}^2\|v_t^{Q_t} - \hat{v}^*(x_t, y_t^{P_t})\|^2 \\
\leq & \bar{L}^2\big(\|y_t^{P_t} - y^*(x_t)\|^2 + \|v_t^{Q_t} - \hat{v}^*(x_t, y^*(x_t))\|^2\big), \quad (9)
\end{aligned}
$$

where (a) uses Assumption 2, Remark 3 and Lemma 3; (b) uses $v^*(x_t) = \hat{v}^*(x_t, y^*(x_t))$ and Lemma 3. By using Lemma 5, we have

$$
\|\nabla\Phi(x_t) - \bar{\nabla} f(x_t, y_t^{P_t}, v_t^{Q_t})\|^2 \leq \frac{\bar{L}^2}{\mu^2}(\epsilon_y + \epsilon_v) =: \epsilon'. \quad (10)
$$

By plugging eq. (10) into eq. (8), we obtain (6).

Now if in addition, $k_1$ in Lemma 4 exists, then for $t \geq k_1$, we have $\alpha_{t+1} > C_\alpha \geq 2L_\Phi$. From (6) we can immediately obtain (7). Thus, the proof is complete. □

### E.4 THE BOUND OF $\alpha_t$

**Lemma 8.** *Under Assumptions 1, 2, 3, suppose the number of total iteration rounds in Algorithm 1 is $T$. If there exists $k_1 \leq T$ described in Lemma 4, we have*

$$
\begin{cases}
\alpha_t \leq C_\alpha, & t \leq k_1; \\
\alpha_t \leq C_\alpha + 2c_0 + \dfrac{2t\epsilon'}{\alpha_0}, & t \geq k_1,
\end{cases}
$$

*where we define*

$$
c_0 := 2\big(\Phi(x_0) - \inf_x \Phi(x)\big) + \frac{L_\Phi C_\alpha^2}{2\alpha_0^2}. \quad (11)
$$

*When such $k_1$ does not exist, we have $\alpha_t \leq C_\alpha$ for any $t \leq T$.*

*Proof.* According to Lemma 4, the proof can be split into the following three cases.

**Case 1:** if $\alpha_T \leq C_\alpha$, for any $t < T$, we have the upper bound of $\alpha_{t+1}$ as $\alpha_{t+1} \leq C_\alpha$.

**Case 2:** if $\alpha_T > C_\alpha$, there exists $k_1 \leq T$ described in Lemma 4. Then we have the upper bound of $\alpha_{t+1}$ as $\alpha_{t+1} \leq C_\alpha$ for any $t < k_1$.

**Case 3:** in the remaining proof, we only consider and explore the case $k_1 \leq t \leq T$ when $\alpha_T > C_\alpha$.

From Lemma 7, for $k \geq k_1$, we have

$$\Phi(x_{k+1}) \leq \Phi(x_k) - \frac{1}{2\alpha_{k+1}}\|\nabla\Phi(x_k)\|^2 - \frac{1}{4\alpha_{k+1}}\|\bar{\nabla}f(x_k, y_k^{P_k}, v_k^{Q_k})\|^2 + \frac{\epsilon'}{2\alpha_{k+1}},$$

which indicates that

$$\frac{\|\bar{\nabla}f(x_k, y_k^{P_k}, v_k^{Q_k})\|^2}{\alpha_{k+1}} \leq 4\big(\Phi(x_k) - \Phi(x_{k+1})\big) + \frac{2\epsilon'}{\alpha_{k+1}}.$$

By taking summation over $k = k_1, \ldots, t$, we have

$$\sum_{k=k_1}^{t} \frac{\|\bar{\nabla}f(x_k, y_k^{P_k}, v_k^{Q_k})\|^2}{\alpha_{k+1}} \leq 4\sum_{k=k_1}^{t}\big(\Phi(x_k) - \Phi(x_{k+1})\big) + \sum_{k=k_1}^{t}\frac{2\epsilon'}{\alpha_{k+1}}$$

$$= 4\big(\Phi(x_{k_1}) - \Phi(x_{t+1})\big) + \sum_{k=k_1}^{t}\frac{2\epsilon'}{\alpha_{k+1}}. \tag{12}$$

For $\Phi(x_{k_1})$, by telescoping (6), we get

$$\Phi(x_{k_1}) \leq \Phi(x_0) + \sum_{k=0}^{k_1-1}\frac{L_\Phi}{4\alpha_{k+1}^2}\|\bar{\nabla}f(x_k, y_k^{P_k}, v_k^{Q_k})\|^2 + \sum_{k=0}^{k_1-1}\frac{\epsilon'}{2\alpha_{k+1}}. \tag{13}$$

Plugging eq. (13) into eq. (12), we obtain

$$\sum_{k=k_1}^{t}\frac{\|\bar{\nabla}f(x_k, y_k^{P_k}, v_k^{Q_k})\|^2}{\alpha_{k+1}} \leq 4\big(\Phi(x_0) - \inf_x\Phi(x)\big) + \sum_{k=0}^{k_1-1}\frac{L_\Phi}{\alpha_{k+1}^2}\|\bar{\nabla}f(x_k, y_k^{P_k}, v_k^{Q_k})\|^2 + \sum_{k=0}^{t}\frac{2\epsilon'}{\alpha_{k+1}}$$

$$\leq 4\big(\Phi(x_0) - \inf_x\Phi(x)\big) + \frac{L_\Phi\sum_{k=0}^{k_1-1}\|\bar{\nabla}f(x_k, y_k^{P_k}, v_k^{Q_k})\|^2}{\alpha_0^2} + \sum_{k=0}^{t}\frac{2\epsilon'}{\alpha_{k+1}}$$

$$\leq 4\big(\Phi(x_0) - \inf_x\Phi(x)\big) + \frac{L_\Phi\alpha_{k_1}^2}{\alpha_0^2} + \frac{2(t+1)\epsilon'}{\alpha_0}$$

$$\leq 4\big(\Phi(x_0) - \inf_x\Phi(x)\big) + \frac{L_\Phi C_\alpha^2}{\alpha_0^2} + \frac{2(t+1)\epsilon'}{\alpha_0}. \tag{14}$$

Inspired by Ward et al. (2020) and using telescoping, we have

$$\alpha_{t+1} = \alpha_t + \frac{\|\bar{\nabla}f(x_t, y_t^{P_t}, v_t^{Q_t})\|^2}{\alpha_{t+1} + \alpha_t}$$

$$\leq \alpha_t + \frac{\|\bar{\nabla}f(x_t, y_t^{P_t}, v_t^{Q_t})\|^2}{\alpha_{t+1}}$$

$$\leq \alpha_{k_1} + \sum_{k=k_1}^{t}\frac{\|\bar{\nabla}f(x_k, y_k^{P_k}, v_k^{Q_k})\|^2}{\alpha_{k+1}}$$

$$\leq C_\alpha + 4\big(\Phi(x_0) - \inf_x\Phi(x)\big) + \frac{L_\Phi C_\alpha^2}{\alpha_0^2} + \frac{2(t+1)\epsilon'}{\alpha_0}.$$

Thus, the proof is complete. $\qquad\square$

### E.5 CONVERGENCE ANALYSIS OF SUB-LOOPS

**Lemma 9.** *Recall that for the $t_{th}$ iteration, the sub-loops in Algorithm 1 aim to find $y_t^{P_t}$ and $v_t^{Q_t}$ such that $\|\nabla_y g(x_t, y_t^{P_t})\|^2 \leq \epsilon_y$ and $\|\nabla_v R(x_t, y_t^{P_t}, v_t^{Q_t})\|^2 \leq \epsilon_v$. Here we prove that*

$$P_t \leq P' := \frac{\log(C_\beta^2/\beta_0^2)}{\log(1+\epsilon_y/C_\beta^2)} + \frac{\beta_{max}}{\mu} \log\big(\frac{L_{g,1}^2(\beta_{max} - C_\beta)}{\epsilon_y}\big), \tag{15a}$$

$$Q_t \leq Q' := \frac{\log(C_\gamma^2/\gamma_0^2)}{\log(1+\epsilon_v/C_\gamma^2)} + \frac{\gamma_{max}}{\mu} \log\big(\frac{C_{g_{yy}}^2(\gamma_{max} - C_\gamma)}{\epsilon_v}\big), \tag{15b}$$

*where $\beta_{max} := C_\beta + L_{g,1}\big(\frac{2\epsilon_y}{\mu^2} + \frac{2C_{g_{xy}}^2 C_f^2}{\mu^2\alpha_0^2} + 2\log(C_\beta/\beta_0) + 1\big)$ and $\gamma_{max} := C_\gamma + C_{g_{yy}}\big(\frac{2\epsilon_y}{\mu^2} + \frac{8C_{f_y}^2}{\mu^2} + 2\log(C_\gamma/\gamma_0) + 1\big)$.*

*Proof.* The proof is split into the following two parts.

**Part I: maximum number for convergence of $g(x_t, y_t^{P_t})$.**

Inspired by Xie et al. (2020), we split the analysis into the following two cases.

**Case 1: $k_2$ does not exist before we find $P_t$.** This indicates $\beta_{P_t} < C_\beta$. Referring to Lemma 2 in Xie et al. (2020), we have $P_t < \frac{\log(C_\beta^2/\beta_0^2)}{\log(1+\epsilon_y/C_\beta^2)}$ and therefore the desired upper bound for $P_t$ holds. This can be proved as follows. If $P_t \geq \frac{\log(C_\beta^2/\beta_0^2)}{\log(1+\epsilon_y/C_\beta^2)}$, we have the following result.

$$\begin{aligned}
\beta_{P_t}^2 &= \beta_{P_t-1}^2 + \|\nabla_y g(x_t, y_t^{P_t-1})\|^2 \\
&= \beta_{P_t-1}^2 \Big(1 + \frac{\|\nabla_y g(x_t, y_t^{P_t-1})\|^2}{\beta_{P_t-1}^2}\Big) \\
&\geq \beta_0^2 \prod_{p=0}^{P_t-1} \Big(1 + \frac{\|\nabla_y g(x_t, y_t^p)\|^2}{\beta_p^2}\Big) \\
&\geq \beta_0^2 \Big(1 + \frac{\epsilon_y}{C_\beta^2}\Big)^{P_t} \geq C_\beta^2. 
\end{aligned} \tag{16}$$

This contradicts $\beta_{P_t} < C_\beta$.

**Case 2: $k_2$ exists and $P_t \geq k_2$.** Here we have $\beta_{k_2} \leq C_\beta$ and $\beta_{k_2+1} > C_\beta$.

**Firstly**, we prove $k_2 \leq \frac{\log(C_\beta^2/\beta_0^2)}{\log(1+\epsilon_y/C_\beta^2)}$. Similar to **Case 1**, if $k_2 > \frac{\log(C_\beta^2/\beta_0^2)}{\log(1+\epsilon_y/C_\beta^2)}$, following eq. (16) by replacing $P_t$ with $k_2$, we have

$$\beta_{k_2}^2 \geq \beta_0^2 \Big(1 + \frac{\epsilon_y}{C_\beta^2}\Big)^{k_2} > C_\beta^2,$$

which contradicts $\beta_{k_2} \leq C_\beta$.

**Secondly**, referring to Lemma 3 in Xie et al. (2020), we have the bound of $\|y_t^{k_2} - y^*(x_t)\|^2$ as

$$\begin{aligned}
&\|y_t^{k_2} - y^*(x_t)\|^2 \\
&= \Big\|y_t^{k_2-1} - \frac{\nabla_y g(x_t, y_t^{k_2-1})}{\beta_{k_2}} - y^*(x_t)\Big\|^2 \\
&= \|y_t^{k_2-1} - y^*(x_t)\|^2 + \Big\|\frac{\nabla_y g(x_t, y_t^{k_2})}{\beta_{k_2}}\Big\|^2 - 2\Big\langle y_t^{k_2-1} - y^*(x_t), \frac{\nabla_y g(x_t, y_t^{k_2-1})}{\beta_{k_2}}\Big\rangle \\
&\overset{(a)}{\leq} \|y_t^{k_2-1} - y^*(x_t)\|^2 + \Big\|\frac{\nabla_y g(x_t, y_t^{k_2-1})}{\beta_{k_2}}\Big\|^2 - \frac{2}{\beta_{k_2} L_{g,1}} \|\nabla_y g(x_t, y_t^{k_2-1}) - \nabla_y g(x_t, y^*(x_t))\|^2
\end{aligned}$$

$$\leq \|y_t^{k_2-1} - y^*(x_t)\|^2 + \frac{\|\nabla_y g(x_t, y_t^{k_2-1})\|^2}{\beta_{k_2}^2}$$

$$\leq \|y_t^0 - y^*(x_t)\|^2 + \sum_{p=0}^{k_2-1} \frac{\|\nabla_y g(x_t, y_t^p)\|^2}{\beta_{p+1}^2}$$

$$\overset{(b)}{\leq} \|y_{t-1}^{P_{t-1}} - y^*(x_t)\|^2 + \sum_{p=0}^{k_2-1} \frac{\|\nabla_y g(x_t, y_t^p)\|^2/\beta_0^2}{\sum_{k=0}^{p} \|\nabla_y g(x_t, y_t^k)\|^2/\beta_0^2}$$

$$\overset{(c)}{\leq} 2\|y_{t-1}^{P_{t-1}} - y^*(x_{t-1})\|^2 + 2\|y^*(x_{t-1}) - y^*(x_t)\|^2 + \log\left(\sum_{p=0}^{k_2-1} \frac{\|\nabla_y g(x_t, y_t^p)\|^2}{\beta_0^2}\right) + 1$$

$$\overset{(d)}{\leq} \frac{2\epsilon_y}{\mu^2} + \frac{2C_{g_{xy}}^2 \|\bar{\nabla} f(x_{t-1}, y_{t-1}^{P_{t-1}}, v_{t-1}^{Q_{t-1}})\|^2}{\mu^2 \alpha_t^2} + \log\left(\sum_{p=0}^{k_2-1} \frac{\|\nabla_y g(x_t, y_t^p)\|^2}{\beta_0^2}\right) + 1$$

$$\overset{(e)}{\leq} \frac{2\epsilon_y}{\mu^2} + \frac{2C_{g_{xy}}^2 C_f^2}{\mu^2 \alpha_0^2} + 2\log(C_\beta/\beta_0) + 1, \tag{17}$$

where (a) uses Assumptions 1,2; (b) refers to the warm start of $y_t^0$; (c) uses Lemma 1; (d) uses Lemmas 2 and 5; (e) follows from Lemma 6 and $\beta_{k_2} \leq C_\beta$.

**Last**, following Xie et al. (2020), for all $P > k_2$, we have the bound of $\|y_t^P - y^*(x_t)\|^2$ as

$$\|y_t^P - y^*(x_t)\|^2 = \|y_t^{P-1} - y^*(x_t)\|^2 + \frac{\|\nabla_y g(x_t, y_t^{P-1})\|^2}{\beta_P^2} - \frac{2\langle y_t^{P-1} - y^*(x_t), \nabla_y g(x_t, y_t^{P-1})\rangle}{\beta_P}$$

$$\leq \|y_t^{P-1} - y^*(x_t)\|^2 - \frac{1}{\beta_P}\left(2 - \frac{L_{g,1}}{\beta_P}\right)\langle y_t^{P-1} - y^*(x_t), \nabla_y g(x_t, y_t^{P-1})\rangle$$

$$\overset{(a)}{\leq} \|y_t^{P-1} - y^*(x_t)\|^2 - \frac{1}{\beta_P}\langle y_t^{P-1} - y^*(x_t), \nabla_y g(x_t, y_t^{P-1})\rangle$$

$$\overset{(b)}{\leq} \left(1 - \frac{\mu}{\beta_P}\right)\|y_t^{P-1} - y^*(x_t)\|^2$$

$$\overset{(c)}{\leq} e^{-\mu(P-k_2)/\beta_P} \|y_t^{k_2} - y^*(x_t)\|^2$$

$$\overset{(d)}{\leq} e^{-\mu(P-k_2)/\beta_P}\left(\frac{2\epsilon_y}{\mu^2} + \frac{2C_{g_{xy}}^2 C_f^2}{\mu^2 \alpha_0^2} + 2\log(C_\beta/\beta_0) + 1\right), \tag{18}$$

where (a) uses $\beta_P \geq C_\beta \geq L_{g,1}$; (b) uses Assumption 1; (c) follows from $\beta_P \geq C_\beta \geq L_{g,1} \geq \mu$ and $1 - m \leq e^{-m}$ for $0 < m < 1$; (d) refers to eq. (17). Inspired by Lemma 4 in Xie et al. (2020), we have the upper-bound of $\beta_P$ as

$$\beta_P = \beta_{P-1} + \frac{\|\nabla_y g(x_t, y_t^{P-1})\|^2}{\beta_P + \beta_{P-1}} \leq \beta_{k_2} + \sum_{p=k_2}^{P-1} \frac{\|\nabla_y g(x_t, y_t^p)\|^2}{\beta_{p+1}}. \tag{19}$$

To further bound the last term of the right-hand side of eq. (19), using Assumption 2, we have the following result:

$$\|y_t^P - y^*(x_t)\|^2$$

$$= \|y_t^{P-1} - y^*(x_t)\|^2 + \frac{\|\nabla_y g(x_t, y_t^{P-1})\|^2}{\beta_P^2} - \frac{2\langle y_t^{P-1} - y^*(x_t), \nabla_y g(x_t, y_t^{P-1})\rangle}{\beta_P}$$

$$\overset{(a)}{\leq} \|y_t^{P-1} - y^*(x_t)\|^2 + \frac{\|\nabla_y g(x_t, y_t^{P-1})\|^2}{\beta_P^2} - \frac{2\|\nabla_y g(x_t, y_t^{P-1}) - \nabla_y g(x_t, y^*(x_t))\|^2}{\beta_P L_{g,1}}$$

$$\overset{(b)}{\leq} \|y_t^{P-1} - y^*(x_t)\|^2 - \frac{\|\nabla_y g(x_t, y_t^{P-1})\|^2}{\beta_P L_{g,1}}$$

$$\leq \|y_t^{k_2} - y^*(x_t)\|^2 - \sum_{p=k_2}^{P-1} \frac{\|\nabla_y g(x_t, y_t^p)\|^2}{\beta_{p+1} L_{g,1}}, \tag{20}$$

where (a) uses Assumptions 1,2 ; (b) refers to $\beta_P \geq C_\beta \geq L_{g,1}$. By rearranging eq. (20) and using eq. (17), we have

$$\sum_{p=k_2}^{P-1} \frac{\|\nabla_y g(x_t, y_t^p)\|^2}{\beta_{p+1}} \leq L_{g,1} \left( \|y_t^{k_2} - y^*(x_t)\|^2 - \|y_t^P - y^*(x_t)\|^2 \right)$$

$$\leq L_{g,1} \|y_t^{k_2} - y^*(x_t)\|^2$$

$$\leq L_{g,1} \left( \frac{2\epsilon_y}{\mu^2} + \frac{2C_{g_{xy}}^2 C_f^2}{\mu^2 \alpha_0^2} + 2 \log(C_\beta/\beta_0) + 1 \right). \tag{21}$$

Plugging eq. (21) into eq. (19), we obtain the upper-bound of $\beta_P$ as

$$\beta_P \leq C_\beta + L_{g,1} \left( \frac{2\epsilon_y}{\mu^2} + \frac{2C_{g_{xy}}^2 C_f^2}{\mu^2 \alpha_0^2} + 2 \log(C_\beta/\beta_0) + 1 \right) =: \beta_{\max}. \tag{22}$$

Then, by plugging eq. (22) into eq. (18), we have the upper bound of $\|y_t^P - y^*(x_t)\|^2$ as

$$\|y_t^P - y^*(x_t)\|^2 \leq e^{-\mu(P-k_2)/\beta_{\max}} \left( \frac{2\epsilon_y}{\mu^2} + \frac{2C_{g_{xy}}^2 C_f^2}{\mu^2 \alpha_0^2} + 2 \log(C_\beta/\beta_0) + 1 \right). \tag{23}$$

Recall we have the upper bound $k_2 \leq \frac{\log(C_\beta^2/\beta_0^2)}{\log(1+\epsilon_y/C_\beta^2)}$. Note that $P'$ defined in (15a) satisfies

$$P' \geq k_2 + \frac{\beta_{\max}}{\mu} \log \left( L_{g,1}^2 (\beta_{\max} - C_\beta)/\epsilon_y \right).$$

By replacing $P$ with $P'$ in eq. (23), we have

$$\|\nabla_y g(x_t, y_t^{P'})\|^2 \leq L_{g,1}^2 \|y_t^{P'} - y^*(x_t)\|^2 \leq e^{-\mu(P'-k_2)/\beta_{\max}} L_{g,1}^2 (\beta_{\max} - C_\beta) \leq \epsilon_y.$$

Therefore, $P_t \leq P'$ and this completes the proof of (15a).

**Part II: maximum number for convergence of $R(x_t, y_t^{P_t}, v_t^{Q_t})$.**

Similarly to **Part I**, we split the analysis into the following two cases.

**Case 1: $k_3$ does not exist before we find $Q_t$.** This indicates $\gamma_{Q_t} < C_\gamma$. Then we have $Q_t < \frac{\log(C_\gamma^2/\gamma_0^2)}{\log(1+\epsilon_v/C_\gamma^2)}$. Otherwise, if $Q_t \geq \frac{\log(C_\gamma^2/\gamma_0^2)}{\log(1+\epsilon_v/C_\gamma^2)}$, we have the following result.

$$\gamma_{Q_t}^2 = \gamma_{Q_t-1}^2 + \|\nabla_v R(x_t, y_t^{P_t}, v_t^{Q_t-1})\|^2$$

$$= \gamma_{Q_t-1}^2 \left( 1 + \frac{\|\nabla_v R(x_t, y_t^{P_t}, v_t^{Q_t-1})\|^2}{\gamma_{Q_t-1}^2} \right)$$

$$\geq \gamma_0^2 \prod_{q=0}^{Q_t-1} \left( 1 + \frac{\|\nabla_v R(x_t, y_t^{P_t}, v_t^{Q_t-1})\|^2}{\gamma_{Q_t-1}^2} \right)$$

$$\geq \gamma_0^2 \left( 1 + \frac{\epsilon_v}{C_\gamma^2} \right)^{Q_t} \geq C_\gamma^2.$$

This contradicts $\gamma_{Q_t} < C_\gamma$.

**Case 2: $k_3$ exists and $Q_t \geq k_3$.** Here we have $\gamma_{k_3} \leq C_\gamma$ and $\gamma_{k_3+1} > C_\gamma$.

**Firstly**, we have $k_3 \leq \frac{\log(C_\gamma^2/\gamma_0^2)}{\log(1+\epsilon_v/C_\gamma^2)}$. Similar to **Case 1**, if $k_3 > \frac{\log(C_\gamma^2/\gamma_0^2)}{\log(1+\epsilon_v/C_\gamma^2)}$, following eq. (16), by replacing $Q_t$ with $k_3$, we have

$$\gamma_{k_3}^2 \geq \gamma_0^2 \left( 1 + \frac{\epsilon_v}{C_\gamma^2} \right)^{k_3} > C_\gamma^2,$$

which contradicts $\gamma_{k_3} \leq C_\gamma$.

**Secondly**, referring to Lemma 3 in Xie et al. (2020), we have the bound of $\|v_t^{k_3} - v^*(x_t)\|^2$ as following:

$$\|v_t^{k_3} - \hat{v}^*(x_t, y_t^{P_t})\|^2 = \left\|v_t^{k_3-1} - \frac{\nabla_v R(x_t, y_t^{P_t}, v_t^{k_3-1})}{\gamma_{k_3}} - \hat{v}^*(x_t, y_t^{P_t})\right\|^2$$

$$= \|v_t^{k_3-1} - \hat{v}^*(x_t, y_t^{P_t})\|^2 + \left\|\frac{\nabla_v R(x_t, y_t^{P_t}, v_t^{k_3-1})}{\gamma_{k_3}}\right\|^2$$

$$- \frac{2}{\gamma_{k_3}} \langle v_t^{k_3-1} - \hat{v}^*(x_t, y_t^{P_t}), \nabla_v R(x_t, y_t^{P_t}, v_t^{k_3-1})\rangle$$

$$\overset{(a)}{\leq} \|v_t^{k_3-1} - \hat{v}^*(x_t, y_t^{P_t})\|^2 + \left\|\frac{\nabla_v R(x_t, y_t^{P_t}, v_t^{k_3-1})}{\gamma_{k_3}}\right\|^2$$

$$- \frac{2}{\gamma_{k_3} C_{g_{yy}}} \|\nabla_v R(x_t, y_t^{P_t}, v_t^{k_3-1}) - \nabla_v R(x_t, y_t^{P_t}, \hat{v}^*(x_t, y_t^{P_t}))\|^2$$

$$\leq \|v_t^{k_3-1} - \hat{v}^*(x_t, y_t^{P_t})\|^2 + \left\|\frac{\nabla_v R(x_t, y_t^{P_t}, v_t^{k_3-1})}{\gamma_{k_3}}\right\|^2$$

$$\leq \|v_t^0 - \hat{v}^*(x_t, y_t^{P_t})\|^2 + \sum_{q=0}^{k_3-1} \left\|\frac{\nabla_v R(x_t, y_t^{P_t}, v_t^q)}{\gamma_{k_3}}\right\|^2$$

$$\overset{(b)}{\leq} \|v_t^0 - \hat{v}^*(x_t, y_t^{P_t})\|^2 + \sum_{q=0}^{k_3-1} \frac{\|\nabla_v R(x_t, y_t^{P_t}, v_t^q)\|^2/\gamma_0^2}{\sum_{k=0}^{q} \|\nabla_v R(x_t, y_t^{P_t}, v_t^k)\|^2/\gamma_0^2}$$

$$\overset{(c)}{\leq} 2\|v_{t-1}^{P_{t-1}} - \hat{v}^*(x_{t-1}, y_{t-1}^{P_{t-1}})\|^2 + 2\|\hat{v}^*(x_{t-1}, y_{t-1}^{P_{t-1}}) - \hat{v}^*(x_t, y_t^{P_t})\|^2$$

$$+ \log\left(\sum_{q=0}^{k_3-1} \|\nabla_v R(x_t, y_t^{P_t}, v_t^k)\|^2/\gamma_0^2\right) + 1$$

$$\leq 2\|v_{t-1}^{P_{t-1}} - \hat{v}^*(x_{t-1}, y_{t-1}^{P_{t-1}})\|^2 + 4\|\hat{v}^*(x_{t-1}, y_{t-1}^{P_{t-1}})\|^2 + 4\|\hat{v}^*(x_t, y_t^{P_t})\|^2$$

$$+ \log\left(\sum_{q=0}^{k_3-1} \|\nabla_v R(x_t, y_t^{P_t}, v_t^k)\|^2/\gamma_0^2\right) + 1$$

$$\overset{(d)}{\leq} \frac{2\epsilon_y}{\mu^2} + \frac{8C_{f_y}^2}{\mu^2} + 2\log(C_\gamma/\gamma_0) + 1, \tag{24}$$

where (a) uses Lemma 3 and $\nabla_v R(x_t, y_t^{P_t}, \hat{v}^*(x_t, y_t^{P_t})) = 0$; (b) refers to the warm start of $v_t^0$; (c) uses Lemma 1; (d) follows from Lemma 3,5 and $\gamma_{k_3} \leq C_\gamma$.

**Last**, similar to **Part I**, for all $Q > k_3$, we explore the bound of $\|v_t^Q - v^*(x_t)\|^2$ as

$$\|v_t^Q - \hat{v}^*(x_t, y_t^{P_t})\|^2$$

$$= \|v_t^{Q-1} - \hat{v}^*(x_t, y_t^{P_t})\|^2 + \frac{\|\nabla_v R(x_t, y_t^{P_t}, v_t^{Q-1})\|^2}{\gamma_Q^2}$$

$$- \frac{2\langle v_t^{Q-1} - \hat{v}^*(x_t, y_t^{P_t}), \nabla_v R(x_t, y_t^{P_t}, v_t^{Q-1})\rangle}{\gamma_Q}$$

$$\overset{(a)}{\leq} \|v_t^{Q-1} - \hat{v}^*(x_t, y_t^{P_t})\|^2 - \frac{1}{\gamma_Q}\left(2 - \frac{C_{g_{yy}}}{\gamma_Q}\right)\langle v_t^{Q-1} - \hat{v}^*(x_t, y_t^{P_t}), \nabla_v R(x_t, y_t^{P_t}, v_t^{Q-1})\rangle$$

$$\overset{(b)}{\leq} \|v_t^{Q-1} - \hat{v}^*(x_t, y_t^{P_t})\|^2 - \frac{1}{\gamma_Q}\langle v_t^{Q-1} - \hat{v}^*(x_t, y_t^{P_t}), \nabla_v R(x_t, y_t^{P_t}, v_t^{Q-1})\rangle$$

$$\overset{(c)}{\leq} \left(1 - \frac{\mu}{\gamma_Q}\right)\|v_t^{Q-1} - \hat{v}^*(x_t, y_t^{P_t})\|^2$$

$$\overset{(d)}{\leq} e^{-\mu(Q-k_3)/\gamma_Q}\|v_t^{k_3} - \hat{v}^*(x_t, y_t^{P_t})\|^2$$

$$\overset{(e)}{\leq} e^{-\mu(Q-k_3)/\gamma_Q} \Big( \frac{2\epsilon_y}{\mu^2} + \frac{8C_{f_y}^2}{\mu^2} + 2\log(C_\gamma/\gamma_0) + 1 \Big), \tag{25}$$

where (a) uses Lemma 3; (b) follows from $\gamma_Q > C_\gamma \geq C_{g_{yy}}$; (c) uses $\nabla_v R(x_t, y_t^{P_t}, \hat{v}^*(x_t, y_t^{P_t})) = 0$ and Lemma 3; (d) follows from $\gamma_Q \geq C_\gamma \geq C_{g_{yy}} \geq \mu$ and $1 - m \leq e^{-m}$ for $0 < m < 1$; (e) uses eq. (24). Similar to eq. (19), we have the upper-bound of $\gamma_Q$ as

$$\gamma_Q = \gamma_{Q-1} + \frac{\|\nabla_v R(x_t, y_t^{P_t}, v_t^{Q-1})\|^2}{\gamma_Q + \gamma_{Q-1}} \leq \gamma_{k_3} + \sum_{q=k_3}^{Q-1} \frac{\|\nabla_v R(x_t, y_t^{P_t}, v_t^q)\|^2}{\gamma_{q+1}}. \tag{26}$$

To further bound the last term on the right-hand side of eq. (26), we can have the following result:

$$\begin{aligned}
\|v_t^Q - \hat{v}^*(x_t, y_t^{P_t})\|^2 =& \|v_t^{Q-1} - \hat{v}^*(x_t, y_t^{P_t})\|^2 + \frac{\|\nabla_v R(x_t, y_t^{P_t}, v_t^{Q-1})\|^2}{\gamma_Q^2} \\
& - \frac{2\langle v_t^Q - \hat{v}^*(x_t, y_t^{P_t}), \nabla_v R(x_t, y_t^{P_t}, v_t^{Q-1})\rangle}{\gamma_Q} \\
\overset{(a)}{\leq}& \|v_t^{Q-1} - \hat{v}^*(x_t, y_t^{P_t})\|^2 + \frac{\|\|\nabla_v R(x_t, y_t^{P_t}, v_t^{Q-1})\|\|^2}{\gamma_Q^2} \\
& - \frac{2\|\nabla_v R(x_t, y_t^{P_t}, v_t^{Q-1}) - \nabla_v R(x_t, y_t^{P_t}, \hat{v}^*(x_t, y_t^{P_t}))\|^2}{\gamma_Q C_{g_{yy}}} \\
\overset{(b)}{\leq}& \|v_t^{Q-1} - \hat{v}^*(x_t, y_t^{P_t})\|^2 - \frac{\|\nabla_v R(x_t, y_t^{P_t}, v_t^{Q-1})\|^2}{\gamma_Q C_{g_{yy}}} \\
\leq& \|v_t^{k_3} - \hat{v}^*(x_t, y_t^{P_t})\|^2 - \sum_{q=k_3}^{Q-1} \frac{\|\nabla_v R(x_t, y_t^{P_t}, v_t^q)\|^2}{\gamma_{q+1} C_{g_{yy}}}, \tag{27}
\end{aligned}$$

where (a) uses Lemma 3; (b) refers to $\gamma_Q \geq C_\gamma \geq C_{g_{yy}}$. By rearranging eq. (27) and using eq. (24), we have

$$\begin{aligned}
\sum_{q=k_3}^{Q-1} \frac{\|\nabla_v R(x_t, y_t^{P_t}, v_t^q)\|^2}{\gamma_{q+1}} \leq& C_{g_{yy}} \big( \|v_t^{k_3} - \hat{v}^*(x_t, y_t^{P_t})\|^2 - \|v_t^Q - \hat{v}^*(x_t, y_t^{P_t})\|^2 \big) \\
\leq& C_{g_{yy}} \Big( \frac{2\epsilon_y}{\mu^2} + \frac{8C_{f_y}^2}{\mu^2} + 2\log(C_\gamma/\gamma_0) + 1 \Big). \tag{28}
\end{aligned}$$

Plugging eq. (23) into eq. (20), we obtain the upper-bound of $\gamma_Q$ as

$$\gamma_Q \leq C_\gamma + C_{g_{yy}} \Big( \frac{2\epsilon_y}{\mu^2} + \frac{8C_{f_y}^2}{\mu^2} + 2\log(C_\gamma/\gamma_0) + 1 \Big) =: \gamma_{\max}. \tag{29}$$

Then, we have the upper bound of $\|v_t^Q - \hat{v}^*(x_t, y_t^{P_t})\|^2$ as

$$\|v_t^Q - \hat{v}^*(x_t, y_t^{P_t})\|^2 \leq e^{-\mu(Q_t - k_3)/\gamma_{\max}} \Big( \frac{2\epsilon_y}{\mu^2} + \frac{8C_{f_y}^2}{\mu^2} + 2\log(C_\gamma/\gamma_0) + 1 \Big). \tag{30}$$

Recall we have the upper bound $k_3 \leq \frac{\log(C_\gamma^2/\gamma_0^2)}{\log(1+\epsilon_v/C_\gamma^2)}$. Note that $Q'$ defined in (15b) satisfies

$$Q' \geq k_3 + \frac{\gamma_{\max}}{\mu} \log \big( C_{g_{yy}}^2 (\gamma_{\max} - C_\gamma)/\epsilon_v \big).$$

By replacing $Q$ with $Q'$ in eq. (30), we have

$$\|\nabla_v R(x_t, y_t^{P_t}, v_t^{Q'})\|^2 \leq C_{g_{yy}}^2 \|v_t^{Q'} - \hat{v}^*(x_t, y_t^{P_t})\|^2 \leq e^{-\mu(Q'-k_3)/\gamma_{\max}} (\gamma_{\max} - C_\gamma) \leq \epsilon_v.$$

Therefore, $Q_t \leq Q'$ and this completes the proof of (15b). Thus, the proof is complete. $\qquad\square$

### E.6 PROOF OF THEOREM 1

Here we suppose the total iteration round is $T$. According to Lemma 4, the proof can be split into the following two cases.

**Case 1:** $k_1$ **does not exist.** Based on Lemma 4, we have $\alpha_T \leq C_\alpha$. Then by Lemma 7 we have

$$\frac{\|\nabla\Phi(x_t)\|^2}{\alpha_{t+1}} \leq 2\big(\Phi(x_t) - \Phi(x_{t+1})\big) + \frac{L_\Phi}{2\alpha_{t+1}^2}\|\bar\nabla f(x_t, y_t^{P_t}, v_t^{Q_t})\|^2 + \frac{\epsilon'}{\alpha_{t+1}},$$

where $\epsilon'$ is defined in Lemma 7. By taking the average, we have

$$\begin{aligned}
\frac{1}{T}\sum_{t=0}^{T-1}\frac{\|\nabla\Phi(x_t)\|^2}{\alpha_{t+1}} \leq & \frac{2}{T}\big(\Phi(x_0) - \Phi(x_T)\big) + \frac{L_\Phi}{2\alpha_0^2}\frac{1}{T}\sum_{t=0}^{T-1}\|\bar\nabla f(x_t, y_t^{P_t}, v_t^{Q_t})\|^2 + \frac{1}{T}\sum_{t=0}^{T-1}\frac{\epsilon'}{\alpha_{t+1}} \\
\leq & \frac{1}{T}\left(2\big(\Phi(x_0) - \inf_x \Phi(x)\big) + \frac{L_\Phi C_\alpha^2}{2\alpha_0^2}\right) + \frac{\epsilon'}{\alpha_0} = \frac{c_0}{T} + \frac{\epsilon'}{\alpha_0},
\end{aligned}\tag{31}$$

where $c_0$ is defined by eq. (11) in Lemma 8.

**Case 2:** $k_1$ **exists.** For $t < k_1$, according to Lemma 7, we still have

$$\frac{\|\nabla\Phi(x_t)\|^2}{\alpha_{t+1}} \leq 2\big(\Phi(x_t) - \Phi(x_{t+1})\big) + \frac{L_\Phi}{2\alpha_{t+1}^2}\|\bar\nabla f(x_t, y_t^{P_t}, v_t^{Q_t})\|^2 + \frac{\epsilon'}{\alpha_{t+1}}.\tag{32}$$

For $t \geq k_1$, we have $\alpha_t \geq C_\alpha$. Using Lemma 7, we have

$$\frac{\|\nabla\Phi(x_t)\|^2}{\alpha_{t+1}} \leq 2\big(\Phi(x_t) - \Phi(x_{t+1})\big) + \frac{\epsilon'}{\alpha_{t+1}}.\tag{33}$$

By merging eq. (32) and eq. (33), and taking an average from $t = 0, ..., T-1$, we have

$$\begin{aligned}
\frac{1}{T}\sum_{t=0}^{T-1}\frac{\|\nabla\Phi(x_t)\|^2}{\alpha_{t+1}} = & \frac{1}{T}\sum_{t=0}^{k_1-1}\frac{\|\nabla\Phi(x_t)\|^2}{\alpha_{t+1}} + \frac{1}{T}\sum_{t=k_1}^{T-1}\frac{\|\nabla\Phi(x_t)\|^2}{\alpha_{t+1}} \\
\leq & \frac{2}{T}\big(\Phi(x_0) - \Phi(x_T)\big) + \frac{L_\Phi}{2\alpha_0^2}\frac{1}{T}\sum_{t=0}^{k_1-1}\|\bar\nabla f(x_t, y_t^{P_t}, v_t^{Q_t})\|^2 + \frac{1}{T}\sum_{t=0}^{T-1}\frac{\epsilon'}{\alpha_{t+1}} \\
\leq & \frac{1}{T}\left(2\big(\Phi(x_0) - \inf_x \Phi(x)\big) + \frac{L_\Phi C_\alpha^2}{2\alpha_0^2}\right) + \frac{\epsilon'}{\alpha_0} = \frac{c_0}{T} + \frac{\epsilon'}{\alpha_0},
\end{aligned}\tag{34}$$

where $c_0$ is defined in Lemma 8. This result is the same as eq. (31). Thus, for both **Case 1** and **Case 2**, we have

$$\frac{1}{T}\sum_{t=0}^{T-1}\frac{\|\nabla\Phi(x_t)\|^2}{\alpha_T} \leq \frac{1}{T}\sum_{t=0}^{T-1}\frac{\|\nabla\Phi(x_t)\|^2}{\alpha_{t+1}} \leq \frac{1}{T}\left(2\big(\Phi(x_0) - \inf_x \Phi(x)\big) + \frac{L_\Phi C_\alpha^2}{2\alpha_0^2}\right) + \frac{\epsilon'}{\alpha_0},$$

which indicates that

$$\begin{aligned}
\frac{1}{T}\sum_{t=0}^{T-1}\|\nabla\Phi(x_t)\|^2 \leq & \left[\frac{1}{T}\left(2\big(\Phi(x_0) - \inf_x \Phi(x)\big) + \frac{L_\Phi C_\alpha^2}{2\alpha_0^2}\right) + \frac{\epsilon'}{\alpha_0}\right]\alpha_T \\
\overset{(a)}{\leq} & \frac{1}{T}\left[\left(2\big(\Phi(x_0) - \inf_x \Phi(x)\big) + \frac{L_\Phi C_\alpha^2}{2\alpha_0^2}\right) + \frac{T\epsilon'}{\alpha_0}\right] \\
& \times \left[C_\alpha + 4\big(\Phi(x_0) - \inf_x \Phi(x)\big) + \frac{L_\Phi C_\alpha^2}{\alpha_0^2} + \frac{2T\epsilon'}{\alpha_0}\right],
\end{aligned}\tag{35}$$

where (a) uses Lemma 8. To achieve the $\mathcal{O}(1/T)$ convergence rate, we need $\epsilon' = \mathcal{O}(1/T)$ in eq. (35). This can be guaranteed by taking $\epsilon_y = 1/T$ and $\epsilon_v = 1/T$, which implies (see Lemma 7)

$$\epsilon' = \frac{1}{T}\left[\left(\frac{2}{\mu^2}\big(\frac{L_{g,2}C_{f_y}}{\mu} + L_{f,1}\big) + 1\right)L_{g,1}^2\bar L^2 + \frac{2\bar L^2}{\mu^2}\right].\tag{36}$$

For symbol convenience, here we define

$$c_1 := c_0 + \frac{1}{\alpha_0}\left[\left(\frac{2}{\mu^2}\left(\frac{L_{g,2}C_{f_y}}{\mu} + L_{f,1}\right) + 1\right)L_{g,1}^2\bar{L}^2 + \frac{2\bar{L}^2}{\mu^2}\right], \tag{37}$$

where $c_0$ is defined in eq. (11). Thus, we can obtain

$$\frac{1}{T}\sum_{t=0}^{T-1}\|\nabla\Phi(x_t)\|^2 \leq \frac{c_1(C_\alpha + 2c_1)}{T} = \mathcal{O}\left(\frac{1}{T}\right).$$

Thus, Theorem 1 is proved.

### E.7 COMPLEXITY ANALYSIS OF ALGORITHM 1 (PROOF OF COROLLARY 1)

Recall in Theorem 1, we take $\epsilon_y = 1/T$, $\epsilon_v = 1/T$, and we obtain

$$\frac{1}{T}\sum_{t=0}^{T-1}\|\nabla\Phi(x_t)\|^2 \leq \frac{c_1(C_\alpha + 2c_1)}{T}.$$

To achieve $\epsilon$-accurate stationary point, we need

$$\frac{1}{T}\sum_{t=0}^{T-1}\|\nabla\Phi(x_t)\|^2 \leq \frac{c_1(C_\alpha + 2c_1)}{T} \leq \epsilon \quad \text{i.e.,} \quad T = \mathcal{O}(1/\epsilon). \tag{38}$$

Recall in Lemma 9, we have

$$\begin{aligned}
P_t &\leq \frac{\log(C_\beta^2/\beta_0^2)}{\log(1 + \epsilon_y/C_\beta^2)} + \frac{\beta_{\max}}{\mu}\log\left(\frac{L_{g,1}^2(\beta_{\max} - C_\beta)}{\epsilon_y}\right) \\
&\leq \frac{\log(C_\beta^2/\beta_0^2)}{\log(1 + 1/C_\beta^2 T)} + \frac{\beta_{\max}}{\mu}\log\left(\frac{TL_{g,1}^2(\beta_{\max} - C_\beta)}{1}\right) = \mathcal{O}\left(\frac{1}{\log(1+\epsilon)} + \log\left(\frac{1}{\epsilon}\right)\right).
\end{aligned}$$

When $\epsilon$ is sufficiently small, we have

$$P_t = \mathcal{O}\left(\frac{1}{\log(1+\epsilon)} + \log\left(\frac{1}{\epsilon}\right)\right) = \mathcal{O}\left(\frac{1}{\epsilon} + \log\left(\frac{1}{\epsilon}\right)\right) = \mathcal{O}(1/\epsilon). \tag{39}$$

Similarly, we have

$$Q_t = \mathcal{O}\left(\frac{1}{\log(1+\epsilon)} + \log\left(\frac{1}{\epsilon}\right)\right) = \mathcal{O}\left(\frac{1}{\epsilon} + \log\left(\frac{1}{\epsilon}\right)\right) = \mathcal{O}(1/\epsilon). \tag{40}$$

We denote $\mathrm{Gc}(\epsilon)$ as the gradient complexity, then we have

$$\mathrm{Gc}(\epsilon) = T \cdot \max_t\{P_t + Q_t\} = \mathcal{O}(1/\epsilon^2).$$

Therefore Corollary 1 is proved.

# F    PROOF OF THEOREM 2

We define some notation for convenience before proving Theorem 2.

## F.1    NOTATION

Below, we define several preset constants for notational convenience at their first use. We first define some Lipschitzness parameters for $\Phi(x)$ as

$$L_\Phi := \Big(L_{f,1} + \frac{L_{g,2}C_{f_y}}{\mu}\Big)\Big(1 + \frac{C_{g_{xy}}}{\mu}\Big)^2$$

$$\bar{L} := \max\Big\{2\Big(\frac{C_{f_y}^2 L_{g,2}^2}{\mu^2} + L_{f,1}^2\Big)^{\frac{1}{2}}, \sqrt{2}C_{g_{yy}}\Big\}.$$

Next, we define the following constants as thresholds for parameters $\beta_k$, $\gamma_k$, $\alpha_k$ as

$$C_\alpha := \max\Big\{\frac{2L_\Phi}{\varphi_0}, \alpha_0\Big\},$$

$$C_\beta := \max\Big\{\mu + L_{g,1}, \frac{2\mu L_{g,1}}{\mu + L_{g,1}}, \beta_0, 64a_0^2, 1\Big\},$$

$$C_\gamma := \max\Big\{2(\mu + C_{g_{yy}}), \frac{\mu C_{g_{yy}}}{\mu + C_{g_{yy}}}, \gamma_0, 64a_0^2, 1, C_{g_{yy}}\Big\},$$

$$C_\varphi := C_\beta + C_\gamma, \tag{41}$$

where the constant $\alpha_0$ is defined as

$$a_0 := \Big(\Big(\frac{4(\mu + C_{g_{yy}})^2}{\mu C_{g_{yy}}} + 8\Big)\Big(\frac{L_{g,2}C_{f_y}}{\mu} + L_{f,1}\Big)^2 \frac{1}{\mu^2} + 1\Big)\frac{(\mu + L_{g,1})^2 L_y^2}{\mu L_{g,1}C_\beta}$$

$$+ \frac{4(\mu + C_{g_{yy}})(\mu + L_{g,1})L_y^2}{\mu^3 L_{g,1}\varphi_0}\Big(\frac{L_{g,2}C_{f_y}}{\mu} + L_{f,1}\Big)^2 + \frac{4(\mu + C_{g_{yy}})^2 L_v^2}{\mu C_{g_{yy}}\gamma_0}.$$

## F.2    A ROUGH BOUND OF $v_k$

**Lemma 10.** *Under Assumptions 1, 2, for any $t \geq 0$ in Algorithm 2, we have $\|v_t\| \leq \frac{\sqrt{2}}{\mu}\varphi_{t+1} + \frac{\sqrt{2}C_{f_y}}{\mu}\sqrt{t}$.*

*Proof.* By strong convexity of $g$ in Assumption 1, we have

$$\sum_{k=1}^{t}\mu^2\|v_k\|^2 \leq \sum_{k=1}^{t}\|\nabla_y\nabla_y g(x_k, y_k)v_k\|^2$$

$$\leq \sum_{k=1}^{t}2\|\nabla_y\nabla_y g(x_k, y_k)v_k - \nabla_y f(x_k, y_k)\|^2 + \sum_{k=1}^{t}2\|\nabla_y f(x_k, y_k)\|^2$$

$$= \sum_{k=1}^{t}2\|\nabla_v R(x_k, y_k, v_k)\|^2 + \sum_{k=1}^{t}2\|\nabla_y f(x_k, y_k)\|^2$$

$$\leq 2\gamma_{t+1}^2 + 2tC_{f_y}^2,$$

which indicates that for any $t \geq 0$, $\|v_t\|$ can be bounded as

$$\|v_t\| \leq \frac{\big(2\gamma_{t+1}^2 + 2tC_{f_y}^2\big)^{\frac{1}{2}}}{\mu} \leq \frac{\big(2\varphi_{t+1}^2 + 2tC_{f_y}^2\big)^{\frac{1}{2}}}{\mu} \leq \frac{\sqrt{2}\big(\varphi_{t+1} + \sqrt{t}C_{f_y}\big)}{\mu}. \tag{42}$$

Then the proof is complete. $\square$

### F.3 DESCENT IN OBJECTIVE FUNCTION

**Lemma 11.** *Under Assumptions 1, 2, for Algorithm 2, suppose the total iteration number is $T$. No matter $k_1$ in Lemma 4 exists or not, we always have*

$$
\begin{aligned}
\Phi(x_{t+1}) \leq & \Phi(x_t) - \frac{1}{2\alpha_{t+1}\varphi_{t+1}}\|\nabla\Phi(x_t)\|^2 - \frac{1}{2\alpha_{t+1}\varphi_{t+1}}\Big(1 - \frac{L_\Phi}{\alpha_{t+1}\varphi_{t+1}}\Big)\|\bar{\nabla}f(x_t, y_t, v_t)\|^2 \\
& + \frac{\bar{L}^2}{2\mu^2}\Big[1 + \frac{2}{\mu^2}\Big(\frac{L_{g,2}C_{f_y}}{\mu} + L_{f,1}\Big)^2\Big]\frac{\|\nabla_y g(x_t, y_t)\|^2}{\alpha_{t+1}\varphi_{t+1}} + \frac{\bar{L}^2}{\mu^2}\frac{\|\nabla_v R(x_t, y_t, v_t)\|^2}{\alpha_{t+1}\varphi_{t+1}}.
\end{aligned}
\tag{43}
$$

*If in addition, $k_1$ in Lemma 4 exists, then for $t \geq k_1$, we further have*

$$
\begin{aligned}
\Phi(x_{t+1}) \leq & \Phi(x_t) - \frac{1}{2\alpha_{t+1}\varphi_{t+1}}\|\nabla\Phi(x_t)\|^2 - \frac{1}{4\alpha_{t+1}\varphi_{t+1}}\|\bar{\nabla}f(x_t, y_t, v_t)\|^2 \\
& + \frac{\bar{L}^2}{2\mu^2}\Big[1 + \frac{2}{\mu^2}\Big(\frac{L_{g,2}C_{f_y}}{\mu} + L_{f,1}\Big)^2\Big]\frac{\|\nabla_y g(x_t, y_t)\|^2}{\alpha_{t+1}\varphi_{t+1}} + \frac{\bar{L}^2}{\mu^2}\frac{\|\nabla_v R(x_t, y_t, v_t)\|^2}{\alpha_{t+1}\varphi_{t+1}},
\end{aligned}
\tag{44}
$$

*where $\bar{L} := \max\big\{2\big(\frac{C_{f_y}^2 L_{g,2}^2}{\mu^2} + L_{f,1}^2\big)^{\frac{1}{2}}, \sqrt{2}C_{g_{yy}}\big\}$.*

*Proof.* From Lemma 2, we have $\Phi(x)$ is $L_\Phi$-smooth. So we can apply the descent lemma to $\Phi$ as

$$
\begin{aligned}
\Phi(x_{t+1}) \leq & \Phi(x_t) + \langle\nabla\Phi(x_t), x_{t+1} - x_t\rangle + \frac{L_\Phi}{2}\|x_{t+1} - x_t\|^2 \\
= & \Phi(x_t) - \frac{1}{\alpha_{t+1}\varphi_{t+1}}\langle\nabla\Phi(x_t), \bar{\nabla}f(x_t, y_t, v_t)\rangle + \frac{L_\Phi}{2\alpha_{t+1}^2\varphi_{t+1}^2}\|\bar{\nabla}f(x_t, y_t, v_t)\|^2 \\
= & \Phi(x_t) - \frac{1}{2\alpha_{t+1}\varphi_{t+1}}\|\nabla\Phi(x)\|^2 - \frac{1}{2\alpha_{t+1}\varphi_{t+1}}\|\bar{\nabla}f(x_t, y_t, v_t)\|^2 \\
& + \frac{1}{2\alpha_{t+1}\varphi_{t+1}}\|\nabla\Phi(x_t) - \bar{\nabla}f(x_t, y_t, v_t)\|^2 + \frac{L_\Phi}{2\alpha_{t+1}^2\varphi_{t+1}^2}\|\bar{\nabla}f(x_t, y_t, v_t)\|^2, \quad (45)
\end{aligned}
$$

and the approximation error

$$
\begin{aligned}
&\|\nabla\Phi(x_t) - \bar{\nabla}f(x_t, y_t, v_t)\|^2 \\
&= \big\|\bar{\nabla}f\big(x_t, y^*(x_t), v^*(x_t)\big) - \bar{\nabla}f(x_t, y_t, v_t)\big\|^2 \\
&\leq 2\big\|\bar{\nabla}f\big(x_t, y^*(x_t), v^*(x_t)\big) - \bar{\nabla}f\big(x_t, y_t, v^*(x_t)\big)\big\|^2 + 2\big\|\bar{\nabla}f\big(x_t, y_t, v^*(x_t)\big) - \bar{\nabla}f(x_t, y_t, v_t)\big\|^2 \\
&\leq 4\big\|\nabla_y\nabla_y g\big(x_t, y^*(x_t)\big)v^*(x_t) - \nabla_y\nabla_y g(x_t, y_t)v^*(x_t)\big\|^2 \\
&\quad + 4\big\|\nabla_y f\big(x_t, y^*(x_t)\big) - \nabla_y f(x_t, y_t)\big\|^2 + 2\big\|\nabla_y\nabla_y g(x_t, y_t)\big(v^*(x_t) - v_t\big)\big\|^2 \\
&\leq 4\Big(\frac{C_{f_y}^2 L_{g,2}^2}{\mu^2} + L_{f,1}^2\Big)\|y_t - y^*(x_t)\|^2 + 2C_{g_{yy}}^2\|v_t - v^*(x_t)\|^2 \\
&\leq \bar{L}^2\big(\|y_t - y^*(x_t)\|^2 + \|v_t - v^*(x_t)\|^2\big),
\end{aligned}
\tag{46}
$$

where the third inequality used results from Lemma 3. By plugging eq. (46) into eq. (45), we have

$$
\begin{aligned}
\Phi(x_{t+1}) \leq & \Phi(x_t) - \frac{1}{2\alpha_{t+1}\varphi_{t+1}}\|\nabla\Phi(x_t)\|^2 - \frac{1}{2\alpha_{t+1}\varphi_{t+1}}\Big(1 - \frac{L_\Phi}{\alpha_{t+1}\varphi_{t+1}}\Big)\|\bar{\nabla}f(x_t, y_t, v_t)\|^2 \\
& + \frac{\bar{L}^2}{2\alpha_{t+1}\varphi_{t+1}}\big(\|y_t - y^*(x_t)\|^2 + \|v_t - v^*(x_t)\|^2\big).
\end{aligned}
\tag{47}
$$

Note that $g(x, y)$ is $\mu$-strongly convex in $y$ and $R(x, y, v)$ is $\mu$-strongly convex in $v$. So here we can bound the approximation gaps $\|y_t - y^*(x_t)\|^2 + \|v_t - v^*(x_t)\|^2$ by $\|\nabla_y g(x_t, y_t)\|^2$ and $\|\nabla_v R(x_t, y_t, v_t)\|^2$ as

$$
\|y_t - y^*(x_t)\|^2 + \|v_t - v^*(x_t)\|^2
$$

$$\overset{(a)}{\leq} \frac{1}{\mu^2}\big\|\nabla_y g(x_t, y_t) - \nabla_y g\big(x_t, y^*(x_t)\big)\big\|^2 + \frac{1}{\mu^2}\big\|\nabla_v R(x_t, y_t, v_t) - \nabla_v R\big(x_t, y_t, v^*(x_t)\big)\big\|^2$$

$$\overset{(b)}{\leq} \frac{1}{\mu^2}\big\|\nabla_y g(x_t, y_t)\big\|^2 + \frac{2}{\mu^2}\big\|\nabla_v R(x_t, y_t, v_t)\big\|^2$$
$$+ \frac{2}{\mu^2}\big\|\nabla_v R\big(x_t, y_t, v^*(x_t)\big) - \nabla_v R\big(x_t, y^*(x_t), v^*(x_t)\big)\big\|^2$$

$$\overset{(c)}{\leq} \frac{1}{\mu^2}\big\|\nabla_y g(x_t, y_t)\big\|^2 + \frac{2}{\mu^2}\big\|\nabla_v R(x_t, y_t, v_t)\big\|^2 + \frac{2}{\mu^2}\left(\frac{L_{g,2}C_{f_y}}{\mu} + L_{f,1}\right)^2 \|y_t - y^*(x_t)\|^2$$

$$\overset{(d)}{\leq} \left[\frac{1}{\mu^2} + \frac{2}{\mu^4}\left(\frac{L_{g,2}C_{f_y}}{\mu} + L_{f,1}\right)^2\right]\big\|\nabla_y g(x_t, y_t)\big\|^2 + \frac{2}{\mu^2}\big\|\nabla_v R(x_t, y_t, v_t)\big\|^2, \tag{48}$$

where (a) and (d) use the strong convexity; (b) and (d) result from $\nabla_y g\big(x, y^*(x)\big) = 0$ and $\nabla_v R\big(x, y^*(x), v^*(x)\big) = 0$; (c) uses Lemma 3. By plugging eq. (48) into eq. (47), we obtain eq. (43).

Now if in addition, $k_1$ in Lemma 4 exists, then for $t \geq k_1$, we have $\alpha_{t+1} > C_\alpha \geq 2L_\Phi/\varphi_0$. From (43) we can immediately obtain (44). Thus, the proof is complete. $\qquad\square$

Note that to further explore the bounds of the right-hand side of eq. (43) and eq. (44) in the above lemma, we next show the (summed) bounds of $\frac{\|\nabla_y g(x_t, y_t)\|^2}{\beta_{t+1}}$ and $\frac{\|\nabla_v R(x_t, y_t, v_t)\|^2}{\varphi_{t+1}}$.

**Lemma 12.** *Under Assumptions 1, 2, for Algorithm 2, suppose the total iteration rounds is $T$. If $k_2$ in Lemma 4 exists within $T$ iterations, for all integer $t \in [k_2, T]$, we have*

$$\sum_{k=k_2}^{t} \frac{\|\nabla_y g(x_k, y_k)\|^2}{\beta_{k+1}} \leq \frac{(\mu + L_{g,1})C_\beta^2}{\mu^2} + \frac{(\mu + L_{g,1})^2 L_y^2}{\mu L_{g,1}\varphi_0} + \frac{(\mu + L_{g,1})^2 L_y^2}{\mu L_{g,1}} \sum_{k=k_2}^{t} \frac{\|\bar{\nabla} f(x_k, y_k, v_k)\|^2}{\alpha_{k+1}^2 \varphi_{k+1}}.$$

*Proof.* For $k_2 \leq t < T$, we have $\beta_{k_2} \leq C_\beta$ and $\beta_{t+1} > C_\beta$. For any positive scalar $\bar{\lambda}_{t+1}$, using Young's inequality, we have

$$\|y_{t+1} - y^*(x_{t+1})\|^2 \leq (1 + \bar{\lambda}_{t+1})\|y_{t+1} - y^*(x_t)\|^2 + \left(1 + \frac{1}{\bar{\lambda}_{t+1}}\right)\|y^*(x_t) - y^*(x_{t+1})\|^2. \tag{49}$$

For the first term on the right hand side of eq. (49), we have

$$\|y_{t+1} - y^*(x_t)\|^2$$
$$= \left\|y_t - \frac{1}{\beta_{t+1}}\nabla_y g(x_t, y_t) - y^*(x_t)\right\|^2$$
$$= \|y_t - y^*(x_t)\|^2 + \frac{1}{\beta_{t+1}^2}\|\nabla_y g(x_t, y_t)\|^2 - \frac{2}{\beta_{t+1}}\big\langle y_t - y^*(x_t), \nabla_y g(x_t, y_t)\big\rangle$$
$$\overset{(a)}{\leq} \left(1 - \frac{2\mu L_{g,1}}{\beta_{t+1}(\mu + L_{g,1})}\right)\|y_t - y^*(x_t)\|^2 + \frac{1}{\beta_{t+1}}\left(\frac{1}{\beta_{t+1}} - \frac{2}{\mu + L_{g,1}}\right)\|\nabla_y g(x_t, y_t)\|^2$$
$$\overset{(b)}{\leq} \left(1 - \frac{2\mu L_{g,1}}{\beta_{t+1}(\mu + L_{g,1})}\right)\|y_t - y^*(x_t)\|^2 - \frac{1}{\beta_{t+1}(\mu + L_{g,1})}\|\nabla_y g(x_t, y_t)\|^2, \tag{50}$$

where (a) uses Lemma 3.11 in Bubeck et al. (2015); (b) follows from $\beta_{t+1} \geq C_\beta \geq \mu + L_{g,1}$. By plugging eq. (50) into eq. (49), we have

$$\|y_{t+1} - y^*(x_{t+1})\|^2$$
$$\leq (1 + \bar{\lambda}_{t+1})\left(1 - \frac{2\mu L_{g,1}}{\beta_{t+1}(\mu + L_{g,1})}\right)\|y_t - y^*(x_t)\|^2 - (1 + \bar{\lambda}_{t+1})\frac{1}{\beta_{t+1}(\mu + L_{g,1})}\|\nabla_y g(x_t, y_t)\|^2$$
$$+ \left(1 + \frac{1}{\bar{\lambda}_{t+1}}\right)\|y^*(x_t) - y^*(x_{t+1})\|^2. \tag{51}$$

By rearranging the terms in eq. (51), we have

$$(1 + \bar{\lambda}_{t+1})\frac{1}{\beta_{t+1}(\mu + L_{g,1})}\|\nabla_y g(x_t, y_t)\|^2$$

$$\leq (1 + \bar{\lambda}_{t+1}) \left( 1 - \frac{2\mu L_{g,1}}{\beta_{t+1}(\mu + L_{g,1})} \right) \|y_t - y^*(x_t)\|^2 - \|y_{t+1} - y^*(x_{t+1})\|^2$$
$$+ \left( 1 + \frac{1}{\bar{\lambda}_{t+1}} \right) \|y^*(x_t) - y^*(x_{t+1})\|^2.$$

We take $\bar{\lambda}_{t+1} := \frac{2\mu L_{g,1}}{\beta_{t+1}(\mu + L_{g,1})}$. Since $\beta_{t+1} > C_\beta \geq \frac{2\mu L_{g,1}}{\mu + L_{g,1}}$ in eq. (41), we have $\bar{\lambda}_{t+1} \leq 1$. Then we have

$$\frac{\|\nabla_y g(x_t, y_t)\|^2}{\beta_{t+1}} \leq (1 + \bar{\lambda}_{t+1}) \frac{\|\nabla_y g(x_t, y_t)\|^2}{\beta_{t+1}}$$
$$\leq (\mu + L_{g,1}) \big( \|y_t - y^*(x_t)\|^2 - \|y_{t+1} - y^*(x_{t+1})\|^2 \big)$$
$$+ \frac{2(\mu + L_{g,1})}{\bar{\lambda}_{t+1}} \|y^*(x_t) - y^*(x_{t+1})\|^2$$
$$= (\mu + L_{g,1}) \big( \|y_t - y^*(x_t)\|^2 - \|y_{t+1} - y^*(x_{t+1})\|^2 \big)$$
$$+ \frac{(\mu + L_{g,1})^2 \beta_{t+1}}{\mu L_{g,1}} \|y^*(x_t) - y^*(x_{t+1})\|^2$$
$$\overset{(a)}{\leq} (\mu + L_{g,1}) \big( \|y_t - y^*(x_t)\|^2 - \|y_{t+1} - y^*(x_{t+1})\|^2 \big)$$
$$+ \frac{(\mu + L_{g,1})^2 L_y^2 \beta_{t+1}}{\mu L_{g,1}} \|x_t - x_{t+1}\|^2,$$

where (a) uses Lemma 2. Summing the above inequality over $k = k_2, \ldots, t$, we have

$$\sum_{k=k_2}^{t} \frac{\|\nabla_y g(x_k, y_k)\|^2}{\beta_{k+1}}$$
$$\leq \sum_{k=k_2-1}^{t} \frac{\|\nabla_y g(x_k, y_k)\|^2}{\beta_{k+1}}$$
$$\leq (\mu + L_{g,1}) \|y_{k_2-1} - y^*(x_{k_2-1})\|^2 + \frac{(\mu + L_{g,1})^2 L_y^2}{\mu L_{g,1}} \sum_{k=k_2-1}^{t} \beta_{k+1} \|x_k - x_{k+1}\|^2$$
$$\overset{(a)}{\leq} \frac{\mu + L_{g,1}}{\mu^2} \big\| \nabla_y g(x_{k_2-1}, y_{k_2-1}) - \nabla_y g\big(x_{k_2-1}, y^*(x_{k_2-1})\big) \big\|^2$$
$$+ \frac{(\mu + L_{g,1})^2 L_y^2}{\mu L_{g,1}} \sum_{k=k_2-1}^{t} \frac{\beta_{k+1}}{\alpha_{k+1}^2 \varphi_{k+1}^2} \|\bar{\nabla} f(x_k, y_k, v_k)\|^2$$
$$\leq \frac{\mu + L_{g,1}}{\mu^2} \|\nabla_y g(x_{k_2-1}, y_{k_2-1})\|^2 + \frac{(\mu + L_{g,1})^2 L_y^2}{\mu L_{g,1}} \sum_{k=k_2-1}^{t} \frac{\|\bar{\nabla} f(x_k, y_k, v_k)\|^2}{\alpha_{k+1}^2 \varphi_{k+1}}$$
$$\overset{(b)}{\leq} \frac{(\mu + L_{g,1}) C_\beta^2}{\mu^2} + \frac{(\mu + L_{g,1})^2 L_y^2}{\mu L_{g,1}} \sum_{k=k_2-1}^{t} \frac{\|\bar{\nabla} f(x_k, y_k, v_k)\|^2}{\alpha_{k+1}^2 \varphi_{k+1}}$$
$$\overset{(c)}{\leq} \frac{(\mu + L_{g,1}) C_\beta^2}{\mu^2} + \frac{(\mu + L_{g,1})^2 L_y^2}{\mu L_{g,1} \varphi_0} + \frac{(\mu + L_{g,1})^2 L_y^2}{\mu L_{g,1}} \sum_{k=k_2}^{t} \frac{\|\bar{\nabla} f(x_k, y_k, v_k)\|^2}{\alpha_{k+1}^2 \varphi_{k+1}}, \qquad (52)$$

where (a) uses Assumption 1; (b) results from $\|\nabla_y g(x_{k_2-1}, y_{k_2-1})\|^2 \leq \beta_{k_2}^2 \leq C_\beta^2$; (c) denotes $\varphi_0 = \max\{\beta_0, \gamma_0\}$. Then, the proof is complete. $\qquad \square$

**Lemma 13.** *Under Assumptions 1, 2, for Algorithm 2, suppose the total iteration rounds is $T$. If $k_3$ in Lemma 4 exists within $T$ iterations, for all integer $t \in [k_3, T)$, we have*

$$\sum_{k=k_3}^{t} \frac{\|\nabla_v R(x_k, y_k, v_k)\|^2}{\varphi_{k+1}}$$

$$\leq \frac{4(\mu + C_{g_{yy}})C_\beta^2}{\mu^4}\left(\frac{L_{g,2}C_{f_y}}{\mu} + L_{f,1}\right)^2 + \frac{4(\mu + C_{g_{yy}})C_\gamma^2}{\mu^2}$$

$$+ \frac{4(\mu + C_{g_{yy}})(\mu + L_{g,1})L_y^2}{\mu^3 L_{g,1}\varphi_0}\left(\frac{L_{g,2}C_{f_y}}{\mu} + L_{f,1}\right)^2 \sum_{k=k_2-1}^{k_3-2} \frac{\|\bar\nabla f(x_k, y_k, v_k)\|^2}{\alpha_{k+1}^2}$$

$$+ \frac{4(\mu + C_{g_{yy}})^2 L_v^2}{\mu C_{g_{yy}} C_\gamma} \sum_{k=k_3-1}^{t} \frac{\|\bar\nabla f(x_k, y_k, v_k)\|^2}{\alpha_{k+1}^2}$$

$$+ \left(\frac{4(\mu + C_{g_{yy}})^2}{\mu C_{g_{yy}}} + 8\right)\left(\frac{L_{g,2}C_{f_y}}{\mu} + L_{f,1}\right)^2 \frac{1}{\mu^2}\sum_{k=k_3-1}^{t} \frac{\|\nabla_y g(x_k, y_k)\|^2}{\beta_{k+1}}.$$

*Proof.* For $k_3 \leq t < T$, we have $\gamma_{t+1} > C_\gamma$. For any positive scalar $\hat\lambda_{t+1}$, using Young's inequality, we have

$$\|v_{t+1} - v^*(x_{t+1})\|^2 \leq (1 + \hat\lambda_{t+1})\|v_{t+1} - v^*(x_t)\|^2 + \left(1 + \frac{1}{\hat\lambda_{t+1}}\right)\|v^*(x_t) - v^*(x_{t+1})\|^2. \quad (53)$$

For the first term on the right hand side of eq. (53), we have

$$\|v_{t+1} - v^*(x_t)\|^2$$
$$= \left\|v_t - \frac{1}{\varphi_{t+1}}\nabla_v R(x_t, y_t, v_t) - v^*(x_t)\right\|^2$$
$$= \|v_t - v^*(x_t)\|^2 + \frac{1}{\varphi_{t+1}^2}\|\nabla_v R(x_t, y_t, v_t)\|^2 - \frac{2}{\varphi_{t+1}}\langle v_t - v^*(x_t), \nabla_v R(x_t, y_t, v_t)\rangle. \quad (54)$$

For the last term of the right-hand side of eq. (54), we have

$$-\langle v_t - v^*(x_t), \nabla_v R(x_t, y_t, v_t)\rangle$$
$$= -\langle v_t - v^*(x_t), \nabla_v R(x_t, y_t, v_t) - \nabla_v R(x_t, y_t, v^*(x_t))\rangle$$
$$\quad - \langle v_t - v^*(x_t), \nabla_v R(x_t, y_t, v^*(x_t)) - \nabla_v R(x_t, y^*(x_t), v^*(x_t))\rangle$$
$$\overset{(a)}{\leq} -\frac{1}{\mu + C_{g_{yy}}}\|\nabla_v R(x_t, y_t, v_t) - \nabla_v R(x_t, y_t, v^*(x_t))\|^2 - \frac{\mu C_{g_{yy}}}{\mu + C_{g_{yy}}}\|v_t - v^*(x_t)\|^2$$
$$\quad + \frac{\mu + C_{g_{yy}}}{2\mu C_{g_{yy}}}\|\nabla_v R(x_t, y_t, v^*(x_t)) - \nabla_v R(x_t, y^*(x_t), v^*(x_t))\|^2$$
$$\quad + \frac{\mu C_{g_{yy}}}{2(\mu + C_{g_{yy}})}\|v_t - v^*(x_t)\|^2$$
$$\overset{(b)}{\leq} -\frac{1}{2(\mu + C_{g_{yy}})}\|\nabla_v R(x_t, y_t, v_t)\|^2 + \frac{1}{\mu + C_{g_{yy}}}\|\nabla_v R(x_t, y_t, v^*(x_t))\|^2$$
$$\quad + \frac{\mu + C_{g_{yy}}}{2\mu C_{g_{yy}}}\|\nabla_v R(x_t, y_t, v^*(x_t)) - \nabla_v R(x_t, y^*(x_t), v^*(x_t))\|^2$$
$$\quad - \frac{\mu C_{g_{yy}}}{2(\mu + C_{g_{yy}})}\|v_t - v^*(x_t)\|^2$$
$$\overset{(c)}{=} -\frac{1}{2(\mu + C_{g_{yy}})}\|\nabla_v R(x_t, y_t, v_t)\|^2 - \frac{\mu C_{g_{yy}}}{2(\mu + C_{g_{yy}})}\|v_t - v^*(x_t)\|^2$$
$$\quad + \left(\frac{1}{\mu + C_{g_{yy}}} + \frac{\mu + C_{g_{yy}}}{2\mu C_{g_{yy}}}\right)\|\nabla_v R(x_t, y_t, v^*(x_t)) - \nabla_v R(x_t, y^*(x_t), v^*(x_t))\|^2$$
$$\overset{(d)}{\leq} -\frac{1}{2(\mu + C_{g_{yy}})}\|\nabla_v R(x_t, y_t, v_t)\|^2 - \frac{\mu C_{g_{yy}}}{2(\mu + C_{g_{yy}})}\|v_t - v^*(x_t)\|^2$$
$$\quad + \left(\frac{1}{\mu + C_{g_{yy}}} + \frac{\mu + C_{g_{yy}}}{2\mu C_{g_{yy}}}\right)(L_{g,2}\|v^*(x_t)\| + L_{f,1})^2\|y_t - y^*(x_t)\|^2$$

$$\stackrel{(e)}{\leq} -\frac{1}{2(\mu + C_{g_{yy}})}\|\nabla_v R(x_t, y_t, v_t)\|^2 - \frac{\mu C_{g_{yy}}}{2(\mu + C_{g_{yy}})}\|v_t - v^*(x_t)\|^2$$
$$+ \left(\frac{1}{\mu + C_{g_{yy}}} + \frac{\mu + C_{g_{yy}}}{2\mu C_{g_{yy}}}\right)\left(\frac{L_{g,2}C_{f_y}}{\mu} + L_{f,1}\right)^2\|y_t - y^*(x_t)\|^2, \tag{55}$$

where (a) follows from Lemma 3.11 in Bubeck et al. (2015); (b) uses $-\|a - b\|^2 \leq -\frac{1}{2}\|a\|^2 + \|b\|^2$ since $\|a - b + b\|^2 \leq 2\|a - b\|^2 + 2\|b\|^2$; (c) uses $\nabla_v R(x_t, y^*(x_t), v^*(x_t)) = 0$; (d) and (e) follow from Lemma 3. Plugging eq. (55) into eq. (54), we have

$$\|v_{t+1} - v^*(x_t)\|^2$$
$$\leq \left(1 - \frac{\mu C_{g_{yy}}}{(\mu + C_{g_{yy}})\varphi_{t+1}}\right)\|v_t - v^*(x_t)\|^2 + \frac{1}{\varphi_{t+1}}\left(\frac{1}{\varphi_{t+1}} - \frac{1}{\mu + C_{g_{yy}}}\right)\|\nabla_v R(x_t, y_t, v_t)\|^2$$
$$+ \left(\frac{2}{\mu + C_{g_{yy}}} + \frac{\mu + C_{g_{yy}}}{\mu C_{g_{yy}}}\right)\left(\frac{L_{g,2}C_{f_y}}{\mu} + L_{f,1}\right)^2\frac{1}{\varphi_{t+1}}\|y_t - y^*(x_t)\|^2$$
$$\stackrel{(a)}{\leq} \left(1 - \frac{\mu C_{g_{yy}}}{(\mu + C_{g_{yy}})\varphi_{t+1}}\right)\|v_t - v^*(x_t)\|^2 - \frac{1}{2(\mu + C_{g_{yy}})\varphi_{t+1}}\|\nabla_v R(x_t, y_t, v_t)\|^2$$
$$+ \left(\frac{2}{\mu + C_{g_{yy}}} + \frac{\mu + C_{g_{yy}}}{\mu C_{g_{yy}}}\right)\left(\frac{L_{g,2}C_{f_y}}{\mu} + L_{f,1}\right)^2\frac{1}{\varphi_{t+1}}\|y_t - y^*(x_t)\|^2, \tag{56}$$

where (a) follows from $\varphi_{t+1} \geq \gamma_{t+1} \geq C_\gamma \geq 2(\mu + C_{g_{yy}})$. Combining eq. (56) with eq. (53), we have

$$\|v_{t+1} - v^*(x_{t+1})\|^2$$
$$\leq (1 + \hat{\lambda}_{t+1})\left(1 - \frac{\mu C_{g_{yy}}}{(\mu + C_{g_{yy}})\varphi_{t+1}}\right)\|v_t - v^*(x_t)\|^2$$
$$- (1 + \hat{\lambda}_{t+1})\frac{1}{2(\mu + C_{g_{yy}})\varphi_{t+1}}\|\nabla_v R(x_t, y_t, v_t)\|^2$$
$$+ (1 + \hat{\lambda}_{t+1})\left(\frac{2}{\mu + C_{g_{yy}}} + \frac{\mu + C_{g_{yy}}}{\mu C_{g_{yy}}}\right)\left(\frac{L_{g,2}C_{f_y}}{\mu} + L_{f,1}\right)^2\frac{1}{\varphi_{t+1}}\|y_t - y^*(x_t)\|^2$$
$$+ \left(1 + \frac{1}{\hat{\lambda}_{t+1}}\right)\|v^*(x_t) - v^*(x_{t+1})\|^2. \tag{57}$$

By rearranging the terms in eq. (57), we have

$$(1 + \hat{\lambda}_{t+1})\frac{1}{2(\mu + C_{g_{yy}})\varphi_{t+1}}\|\nabla_v R(x_t, y_t, v_t)\|^2$$
$$\leq (1 + \hat{\lambda}_{t+1})\left(1 - \frac{\mu C_{g_{yy}}}{(\mu + C_{g_{yy}})\varphi_{t+1}}\right)\|v_t - v^*(x_t)\|^2 - \|v_{t+1} - v^*(x_{t+1})\|^2$$
$$+ (1 + \hat{\lambda}_{t+1})\left(\frac{2}{\mu + C_{g_{yy}}} + \frac{\mu + C_{g_{yy}}}{\mu C_{g_{yy}}}\right)\left(\frac{L_{g,2}C_{f_y}}{\mu} + L_{f,1}\right)^2\frac{1}{\varphi_{t+1}}\|y_t - y^*(x_t)\|^2$$
$$+ \left(1 + \frac{1}{\hat{\lambda}_{t+1}}\right)\|v^*(x_t) - v^*(x_{t+1})\|^2. \tag{58}$$

We now take $\hat{\lambda}_{t+1} := \frac{\mu C_{g_{yy}}}{(\mu + C_{g_{yy}})\varphi_{t+1}}$. Since $\varphi_{t+1} \geq \gamma_{t+1} \geq C_\gamma \geq \frac{\mu C_{g_{yy}}}{\mu + C_{g_{yy}}}$ in eq. (41), we have $\hat{\lambda}_{t+1} \leq 1$. Then we get

$$\frac{\|\nabla_v R(x_t, y_t, v_t)\|^2}{\varphi_{t+1}} < (1 + \hat{\lambda}_{t+1})\frac{\|\nabla_v R(x_t, y_t, v_t)\|^2}{\varphi_{t+1}}$$
$$\stackrel{(a)}{\leq} 2(\mu + C_{g_{yy}})\left(\|v_t - v^*(x_t)\|^2 - \|v_{t+1} - v^*(x_{t+1})\|^2\right)$$
$$+ 4(\mu + C_{g_{yy}})\left(\frac{2}{\mu + C_{g_{yy}}} + \frac{\mu + C_{g_{yy}}}{\mu C_{g_{yy}}}\right)\left(\frac{L_{g,2}C_{f_y}}{\mu} + L_{f,1}\right)^2\frac{\|y_t - y^*(x_t)\|^2}{\varphi_{t+1}}$$

$$+ 2(\mu + C_{g_{yy}})\left(1 + \frac{(\mu + C_{g_{yy}})\varphi_{t+1}}{\mu C_{g_{yy}}}\right)L_v^2 \|x_t - x_{t+1}\|^2$$

$$\overset{(b)}{\leq} 2(\mu + C_{g_{yy}})\left(\|v_t - v^*(x_t)\|^2 - \|v_{t+1} - v^*(x_{t+1})\|^2\right)$$

$$+ \left(\frac{4(\mu + C_{g_{yy}})^2}{\mu C_{g_{yy}}} + 8\right)\left(\frac{L_{g,2}C_{f_y}}{\mu} + L_{f,1}\right)^2 \frac{\|y_t - y^*(x_t)\|^2}{\varphi_{t+1}}$$

$$+ \frac{4(\mu + C_{g_{yy}})^2 L_v^2 \varphi_{t+1}}{\mu C_{g_{yy}}} \|x_t - x_{t+1}\|^2, \tag{59}$$

where (a) multiplies both sides of eq. (58) by $2(\mu + C_{g_{xy}})$ and uses $\hat{\lambda}_{t+1} \leq 1$; (b) uses $\varphi_{t+1} \geq \gamma_{t+1} \geq C_\gamma \geq \frac{\mu C_{g_{yy}}}{\mu + C_{g_{yy}}}$. Take summation of eq. (59) and we have

$$\sum_{k=k_3}^{t} \frac{\|\nabla_v R(x_k, y_k, v_k)\|^2}{\varphi_{k+1}}$$

$$\leq \sum_{k=k_3-1}^{t} \frac{\|\nabla_v R(x_k, y_k, v_k)\|^2}{\varphi_{k+1}}$$

$$\leq 2(\mu + C_{g_{yy}})\|v_{k_3-1} - v^*(x_{k_3-1})\|^2 + \frac{4(\mu + C_{g_{yy}})^2 L_v^2}{\mu C_{g_{yy}}} \sum_{k=k_3-1}^{t} \varphi_{k+1}\|x_k - x_{k+1}\|^2$$

$$+ \left(\frac{4(\mu + C_{g_{yy}})^2}{\mu C_{g_{yy}}} + 8\right)\left(\frac{L_{g,2}C_{f_y}}{\mu} + L_{f,1}\right)^2 \sum_{k=k_3-1}^{t} \frac{\|y_k - y^*(x_k)\|^2}{\varphi_{k+1}}$$

$$\leq 2(\mu + C_{g_{yy}})\|v_{k_3-1} - v^*(x_{k_3-1})\|^2 + \frac{4(\mu + C_{g_{yy}})^2 L_v^2}{\mu C_{g_{yy}}} \sum_{k=k_3-1}^{t} \frac{\|\bar{\nabla}f(x_k, y_k, v_k)\|^2}{\alpha_{k+1}^2 \varphi_{k+1}}$$

$$+ \left(\frac{4(\mu + C_{g_{yy}})^2}{\mu C_{g_{yy}}} + 8\right)\left(\frac{L_{g,2}C_{f_y}}{\mu} + L_{f,1}\right)^2 \sum_{k=k_3-1}^{t} \frac{\|y_k - y^*(x_k)\|^2}{\varphi_{k+1}}$$

$$\leq 2(\mu + C_{g_{yy}})\|v_{k_3-1} - v^*(x_{k_3-1})\|^2 + \frac{4(\mu + C_{g_{yy}})^2 L_v^2}{\mu C_{g_{yy}}} \sum_{k=k_3-1}^{t} \frac{\|\bar{\nabla}f(x_k, y_k, v_k)\|^2}{\alpha_{k+1}^2 \varphi_{k+1}}$$

$$+ \left(\frac{4(\mu + C_{g_{yy}})^2}{\mu C_{g_{yy}}} + 8\right)\left(\frac{L_{g,2}C_{f_y}}{\mu} + L_{f,1}\right)^2 \sum_{k=k_3-1}^{t} \frac{\|y_k - y^*(x_k)\|^2}{\beta_{k+1}}$$

$$\overset{(a)}{\leq} 2(\mu + C_{g_{yy}})\|v_{k_3-1} - v^*(x_{k_3-1})\|^2 + \frac{4(\mu + C_{g_{yy}})^2 L_v^2}{\mu C_{g_{yy}}} \sum_{k=k_3-1}^{t} \frac{\|\bar{\nabla}f(x_k, y_k, v_k)\|^2}{\alpha_{k+1}^2 \varphi_{k+1}}$$

$$+ \left(\frac{4(\mu + C_{g_{yy}})^2}{\mu C_{g_{yy}}} + 8\right)\left(\frac{L_{g,2}C_{f_y}}{\mu} + L_{f,1}\right)^2 \frac{1}{\mu^2} \sum_{k=k_3-1}^{t} \frac{\|\nabla_y g(x_k, y_k) - \nabla_y g(x_k, y^*(x_k))\|^2}{\beta_{k+1}}$$

$$\overset{(b)}{\leq} 2(\mu + C_{g_{yy}})\|v_{k_3-1} - v^*(x_{k_3-1})\|^2 + \frac{4(\mu + C_{g_{yy}})^2 L_v^2}{\mu C_{g_{yy}}} \sum_{k=k_3-1}^{t} \frac{\|\bar{\nabla}f(x_k, y_k, v_k)\|^2}{\alpha_{k+1}^2 \varphi_{k+1}}$$

$$+ \left(\frac{4(\mu + C_{g_{yy}})^2}{\mu C_{g_{yy}}} + 8\right)\left(\frac{L_{g,2}C_{f_y}}{\mu} + L_{f,1}\right)^2 \frac{1}{\mu^2} \sum_{k=k_3-1}^{t} \frac{\|\nabla_y g(x_k, y_k)\|^2}{\beta_{k+1}}$$

$$\overset{(c)}{\leq} \frac{2(\mu + C_{g_{yy}})}{\mu^2} \left\|\nabla_v R(x_{k_3-1}, y_{k_3-1}, v_{k_3-1}) - \nabla_v R(x_{k_3-1}, y_{k_3-1}, v^*(x_{k_3-1}))\right\|^2$$

$$+ \frac{4(\mu + C_{g_{yy}})^2 L_v^2}{\mu C_{g_{yy}}} \sum_{k=k_3-1}^{t} \frac{\|\bar{\nabla}f(x_k, y_k, v_k)\|^2}{\alpha_{k+1}^2 \varphi_{k+1}}$$

$$+ \left(\frac{4(\mu + C_{g_{yy}})^2}{\mu C_{g_{yy}}} + 8\right)\left(\frac{L_{g,2}C_{f_y}}{\mu} + L_{f,1}\right)^2 \frac{1}{\mu^2} \sum_{k=k_3-1}^{t} \frac{\|\nabla_y g(x_k, y_k)\|^2}{\beta_{k+1}}$$

$$\overset{(d)}{\leq} \frac{4(\mu + C_{g_{yy}})}{\mu^2} \left\|\nabla_v R(x_{k_3-1}, y^*(x_{k_3-1}), v^*(x_{k_3-1})) - \nabla_v R(x_{k_3-1}, y_{k_3-1}, v^*(x_{k_3-1}))\right\|^2$$

$$+ \frac{4(\mu + C_{g_{yy}})}{\mu^2} \left\| \nabla_v R(x_{k_3-1}, y_{k_3-1}, v_{k_3-1}) - \nabla_v R\big(x_{k_3-1}, y^*(x_{k_3-1}), v^*(x_{k_3-1})\big) \right\|^2$$

$$+ \frac{4(\mu + C_{g_{yy}})^2 L_v^2}{\mu C_{g_{yy}}} \sum_{k=k_3-1}^{t} \frac{\|\bar{\nabla} f(x_k, y_k, v_k)\|^2}{\alpha_{k+1}^2 \varphi_{k+1}}$$

$$+ \left( \frac{4(\mu + C_{g_{yy}})^2}{\mu C_{g_{yy}}} + 8 \right) \left( \frac{L_{g,2} C_{f_y}}{\mu} + L_{f,1} \right)^2 \frac{1}{\mu^2} \sum_{k=k_3-1}^{t} \frac{\|\nabla_y g(x_k, y_k)\|^2}{\beta_{k+1}}$$

$$\overset{(e)}{\le} \frac{4(\mu + C_{g_{yy}})}{\mu^2} \left( \frac{L_{g,2} C_{f_y}}{\mu} + L_{f,1} \right)^2 \|y_{k_3-1} - y^*(x_{k_3-1})\|^2$$

$$+ \frac{4(\mu + C_{g_{yy}})}{\mu^2} \|\nabla_v R(x_{k_3-1}, y_{k_3-1}, v_{k_3-1})\|^2 + \frac{4(\mu + C_{g_{yy}})^2 L_v^2}{\mu C_{g_{yy}}} \sum_{k=k_3-1}^{t} \frac{\|\bar{\nabla} f(x_k, y_k, v_k)\|^2}{\alpha_{k+1}^2 \varphi_{k+1}}$$

$$+ \left( \frac{4(\mu + C_{g_{yy}})^2}{\mu C_{g_{yy}}} + 8 \right) \left( \frac{L_{g,2} C_{f_y}}{\mu} + L_{f,1} \right)^2 \frac{1}{\mu^2} \sum_{k=k_3-1}^{t} \frac{\|\nabla_y g(x_k, y_k)\|^2}{\beta_{k+1}}, \tag{60}$$

where (a) uses Assumption 1; (b) results from $\nabla_y g\big(x, y^*(x)\big) = 0$; (c) uses the strong convexity in Lemma 3; (d) uses $\nabla_v R\big(x, y^*(x), v^*(x)\big) = 0$; (e) follows from Lemma 3.

Our next step is bounding $\|y_{k_3-1} - y^*(x_{k_3-1})\|^2$ on the right hand side of eq. (60) in two cases. **The first case is $\beta_{k_3} \le C_\beta$.** In this case, by using strong convexity of $g$ and the definition of $\beta_{k_3}$, we can easily have

$$\|y_{k_3-1} - y^*(x_{k_3-1})\|^2 \le \frac{1}{\mu^2} \left\| \nabla_y g\big(x_{k_3-1}, y_{k_3-1}\big) - \nabla_y g\big(x_{k_3-1}, y^*(x_{k_3-1})\big) \right\|^2$$

$$= \frac{1}{\mu^2} \|\nabla_y g(x_{k_3-1}, y_{k_3-1}))\|^2 \le \frac{\beta_{k_3}^2}{\mu^2} \le \frac{C_\beta^2}{\mu^2}. \tag{61}$$

**The second case is $\beta_{k_3} > C_\beta$.** This indicates that $k_2$ exists and $k_3 > k_2$ based on Lemma 4. By plugging $\bar{\lambda}_{k_3-1} := \frac{2\mu L_{g,1}}{\beta_{k_3-1}(\mu + L_{g,1})}$ into eq. (51), and noting $\bar{\lambda}_{k_3-1} \le 1$, we have

$$\|y_{k_3-1} - y^*(x_{k_3-1})\|^2 \le \|y_{k_3-2} - y^*(x_{k_3-2})\|^2 + \frac{(\mu + L_{g,1})\beta_{k_3-1}}{\mu L_{g,1}} \|y^*(x_{k_3-2}) - y^*(x_{k_3-1})\|^2$$

$$\overset{(a)}{\le} \|y_{k_3-2} - y^*(x_{k_3-2})\|^2 + \frac{(\mu + L_{g,1})L_y^2 \beta_{k_3-1}}{\mu L_{g,1}} \|x_{k_3-2} - x_{k_3-1}\|^2$$

$$= \|y_{k_3-2} - y^*(x_{k_3-2})\|^2 + \frac{(\mu + L_{g,1})L_y^2 \beta_{k_3-1}}{\mu L_{g,1}} \frac{\|\bar{\nabla} f(x_{k_3-2}, y_{k_3-2}, v_{k_3-2})\|^2}{\alpha_{k_3-1}^2 \varphi_{k_3-1}^2}$$

$$\le \|y_{k_3-2} - y^*(x_{k_3-2})\|^2 + \frac{(\mu + L_{g,1})L_y^2}{\mu L_{g,1}} \frac{\|\bar{\nabla} f(x_{k_3-2}, y_{k_3-2}, v_{k_3-2})\|^2}{\alpha_{k_3-1}^2 \varphi_{k_3-1}}$$

$$\le \|y_{k_3-2} - y^*(x_{k_3-2})\|^2 + \frac{(\mu + L_{g,1})L_y^2}{\mu L_{g,1} \varphi_0} \frac{\|\bar{\nabla} f(x_{k_3-2}, y_{k_3-2}, v_{k_3-2})\|^2}{\alpha_{k_3-1}^2}$$

$$\le \|y_{k_2-1} - y^*(x_{k_2-1})\|^2 + \frac{(\mu + L_{g,1})L_y^2}{\mu L_{g,1} \varphi_0} \sum_{k=k_2-1}^{k_3-2} \frac{\|\bar{\nabla} f(x_k, y_k, v_k)\|^2}{\alpha_{k+1}^2}$$

$$\overset{(b)}{\le} \frac{C_\beta^2}{\mu^2} + \frac{(\mu + L_{g,1})L_y^2}{\mu L_{g,1} \varphi_0} \sum_{k=k_2-1}^{k_3-2} \frac{\|\bar{\nabla} f(x_k, y_k, v_k)\|^2}{\alpha_{k+1}^2}, \tag{62}$$

where (a) uses Lemma 2; (b) uses eq. (61) by replacing $k_3$ by $k_2$ since $\beta_{k_2} \le C_\beta$ (see Lemma 4). By combining eq. (61) and eq. (62), we obtain a **general** upper bound of $\|y_{k_3-1} - y^*(x_{k_3-1})\|^2$ as

$$\|y_{k_3-1} - y^*(x_{k_3-1})\|^2 \le \frac{C_\beta^2}{\mu^2} + \frac{(\mu + L_{g,1})L_y^2}{\mu L_{g,1} \varphi_0} \sum_{k=k_2-1}^{k_3-2} \frac{\|\bar{\nabla} f(x_k, y_k, v_k)\|^2}{\alpha_{k+1}^2}, \tag{63}$$

where we define $\sum_{k=m}^{n} l_k = 0$ for any $m > n$ and non-negative sequence $\{l_k\}$. By plugging eq. (63) into eq. (60) and using $\|\nabla_v R(x_{k_3-1}, y_{k_3-1}, v_{k_3-1})\|^2 \leq \gamma_{k_3}^2 \leq C_\gamma^2$, we have

$$
\sum_{k=k_3}^{t} \frac{\|\nabla_v R(x_k, y_k, v_k)\|^2}{\varphi_{k+1}}
$$

$$
\leq \frac{4(\mu + C_{g_{yy}})C_\beta^2}{\mu^4}\left(\frac{L_{g,2}C_{f_y}}{\mu} + L_{f,1}\right)^2 + \frac{4(\mu + C_{g_{yy}})C_\gamma^2}{\mu^2}
$$

$$
+ \frac{4(\mu + C_{g_{yy}})(\mu + L_{g,1})L_y^2}{\mu^3 L_{g,1}\varphi_0}\left(\frac{L_{g,2}C_{f_y}}{\mu} + L_{f,1}\right)^2 \sum_{k=k_2-1}^{k_3-2} \frac{\|\bar{\nabla}f(x_k, y_k, v_k)\|^2}{\alpha_{k+1}^2}
$$

$$
+ \frac{4(\mu + C_{g_{yy}})^2 L_v^2}{\mu C_{g_{yy}} C_\gamma} \sum_{k=k_3-1}^{t} \frac{\|\bar{\nabla}f(x_k, y_k, v_k)\|^2}{\alpha_{k+1}^2}
$$

$$
+ \left(\frac{4(\mu + C_{g_{yy}})^2}{\mu C_{g_{yy}}} + 8\right)\left(\frac{L_{g,2}C_{f_y}}{\mu} + L_{f,1}\right)^2 \frac{1}{\mu^2} \sum_{k=k_3-1}^{t} \frac{\|\nabla_y g(x_k, y_k)\|^2}{\beta_{k+1}}.
$$

Then, the proof is complete. $\qquad\square$

Supported by Lemma 12 and Lemma 13, we derive upper bounds of $\beta_t$ and $\varphi_t$.

**Lemma 14.** *Suppose the total iteration rounds of Algorithm 2 is $T$. Under Assumptions 1, 2, if $k_2$ in Lemma 4 exists within $T$ iterations, we have*

$$
\begin{cases}
\beta_{t+1} \leq C_\beta, & t < k_2; \\
\beta_{t+1} \leq \left(C_\beta + \dfrac{(\mu + L_{g,1})C_\beta^2}{\mu^2} + \dfrac{(\mu + L_{g,1})^2 L_y^2}{\mu L_{g,1}\varphi_0}\right) + \dfrac{(\mu + L_{g,1})^2 L_y^2}{\mu L_{g,1}C_\beta} \displaystyle\sum_{k=k_2}^{t} \dfrac{\|\bar{\nabla}f(x_k, y_k, v_k)\|^2}{\alpha_{k+1}^2}, & t \geq k_2.
\end{cases}
$$

*When such $k_2$ does not exist, $\beta_{t+1} \leq C_\beta$ holds for any $t < T$.*

*Proof.* According to Lemma 4, the proof can be split into the following three cases.

**Case 1: $k_2$ does not exist:** In this case, based on Lemma 4, we have $\beta_T \leq C_\beta$, and hence $\beta_{t+1} \leq C_\beta$ for any $t < T$ because $\beta_t$ is non-decreasing with $t$.

**Case 2: $k_2$ exists and $t < k_2$:** In this case, based on Lemma 4, we have $\beta_{t+1} \leq C_\beta$.

**Case 3: $k_2$ exists and $t \geq k_2$:** Inspired by Ward et al. (2020) and using telescoping, we have

$$
\begin{aligned}
\beta_{t+1} &= \beta_t + \frac{\|\nabla_y g(x_t, y_t)\|^2}{\beta_{t+1} + \beta_t} \\
&\leq \beta_t + \frac{\|\nabla_y g(x_t, y_t)\|^2}{\beta_{t+1}} \\
&\leq \beta_{k_2} + \sum_{k=k_2}^{t} \frac{\|\nabla_y g(x_k, y_k)\|^2}{\beta_{k+1}} \\
&\overset{(a)}{\leq} \left(C_\beta + \frac{(\mu + L_{g,1})C_\beta^2}{\mu^2} + \frac{(\mu + L_{g,1})^2 L_y^2}{\mu L_{g,1}\varphi_0}\right) + \frac{(\mu + L_{g,1})^2 L_y^2}{\mu L_{g,1}C_\beta} \sum_{k=k_2}^{t} \frac{\|\bar{\nabla}f(x_k, y_k, v_k)\|^2}{\alpha_{k+1}^2},
\end{aligned}
$$
(64)

where (a) uses lemma 12. Thus, the proof is complete. $\qquad\square$

**Lemma 15.** *Under Assumptions 1, 2, suppose the total iteration rounds of Algorithm 2 is $T$. If at least one of $k_2$ and $k_3$ in Lemma 4 exists, we denote $k_{min} := \min\{k_2, k_3\}$. Then we have the upper bound of $\varphi_t$ as*

$$
\begin{cases}
\varphi_t \leq C_\varphi, & t \leq k_{min}; \\
\varphi_t \leq a_1 \log(t) + b_1, & t > k_{min},
\end{cases}
$$

*where $a_1$, $b_1$ are defined as*

$$a_1 := 6a_0, \quad b_1 := 4a_0 \log\left(1 + \frac{C_{g_{xy}}\bar{b} + C_{f_x} + \alpha_0}{C_{g_{xy}}\bar{a}}\right) + 4a_0 \log(C_{g_{xy}}\bar{a}) + 4a_0 + 2b_0, \quad (65)$$

*in which we define constants*

$$
\begin{aligned}
\bar{a} &:= \frac{\sqrt{2}}{\mu}, \quad \bar{b} := \frac{\sqrt{2}C_{f_y}}{\mu}, \\
a_0 &:= \left(\left(\frac{4(\mu + C_{g_{yy}})^2}{\mu C_{g_{yy}}} + 8\right)\left(\frac{L_{g,2}C_{f_y}}{\mu} + L_{f,1}\right)^2 \frac{1}{\mu^2} + 1\right)\frac{(\mu + L_{g,1})^2 L_y^2}{\mu L_{g,1} C_\beta} \\
&\quad + \frac{4(\mu + C_{g_{yy}})(\mu + L_{g,1})L_y^2}{\mu^3 L_{g,1}\varphi_0}\left(\frac{L_{g,2}C_{f_y}}{\mu} + L_{f,1}\right)^2 + \frac{4(\mu + C_{g_{yy}})^2 L_v^2}{\mu C_{g_{yy}}\gamma_0}, \\
b_0 &:= C_\beta + C_\gamma + \frac{4(\mu + C_{g_{yy}})C_\beta^2}{\mu^4}\left(\frac{L_{g,2}C_{f_y}}{\mu} + L_{f,1}\right)^2 + \frac{4(\mu + C_{g_{yy}})C_\gamma^2}{\mu^2} \\
&\quad + \left(\frac{4(\mu + C_{g_{yy}})^2}{\mu C_{g_{yy}}} + 8\right)\left(\frac{L_{g,2}C_{f_y}}{\mu} + L_{f,1}\right)^2 \frac{1}{\mu^2}\left(\frac{C_\beta^2}{\beta_0} - \beta_0\right) \\
&\quad + \left[\left(\frac{4(\mu + C_{g_{yy}})^2}{\mu C_{g_{yy}}} + 8\right)\left(\frac{L_{g,2}C_{f_y}}{\mu} + L_{f,1}\right)^2 \frac{1}{\mu^2} + 1\right]\left(\frac{(\mu + L_{g,1})C_\beta^2}{\mu^2} + \frac{(\mu + L_{g,1})^2 L_y^2}{\mu L_{g,1}\varphi_0}\right). \quad (66)
\end{aligned}
$$

*When such $k_2$ and $k_3$ do not exist, we have $\varphi_t \leq C_\varphi$ for all $t \leq T$.*

*Proof.* To begin with, we first show the following result as the first two lines of eq. (64): since $\beta_t$ and $\gamma_t$ are positive and increasing monotonically with $t$, we can easily have

$$
\begin{aligned}
0 &\leq \min\{\beta_{t+1}^2, \gamma_{t+1}^2\} - \min\{\beta_t^2, \gamma_t^2\} \\
&= \left(\beta_{t+1}^2 + \gamma_{t+1}^2 - \max\{\beta_{t+1}^2, \gamma_{t+1}^2\}\right) - \left(\beta_t^2 + \gamma_t^2 - \max\{\beta_t^2, \gamma_t^2\}\right) \\
&\overset{(a)}{=} (\beta_{t+1}^2 + \gamma_{t+1}^2) - (\beta_t^2 + \gamma_t^2) - (\varphi_{t+1}^2 - \varphi_t^2),
\end{aligned}
$$

where (a) uses the definition $\varphi_t := \max\{\beta_t, \gamma_t\}$. Similar to eq. (64), we have

$$\varphi_{t+1}^2 - \varphi_t^2 \leq (\beta_{t+1}^2 - \beta_t^2) + (\gamma_{t+1}^2 - \gamma_t^2) = \|\nabla_y g(x_t, y_t)\|^2 + \|\nabla_v R(x_t, y_t, v_t)\|^2,$$

which indicates that

$$
\begin{aligned}
\varphi_{t+1} &\leq \varphi_t + \frac{\|\nabla_y g(x_t, y_t)\|^2}{\varphi_{t+1} + \varphi_t} + \frac{\|\nabla_v R(x_t, y_t, v_t)\|^2}{\varphi_{t+1} + \varphi_t} \\
&\leq \varphi_t + \frac{\|\nabla_y g(x_t, y_t)\|^2}{\beta_{t+1} + \beta_t} + \frac{\|\nabla_v R(x_t, y_t, v_t)\|^2}{\varphi_{t+1}} \\
&\leq \varphi_t + \frac{\|\nabla_y g(x_t, y_t)\|^2}{\beta_{t+1}} + \frac{\|\nabla_v R(x_t, y_t, v_t)\|^2}{\varphi_{t+1}}. \quad (67)
\end{aligned}
$$

Note that, to simplify the proof, we define $\sum_{k=m}^n l_k = 0$ for any $m > n$ and non-negative sequence $\{l_k\}$. According to the definitions of $k_2$ and $k_3$ in Lemma 4, the proof can be split into the following four cases.

**Case 1: neither $k_2$ nor $k_3$ exists:** for any $t \in (0, T)$, we can easily have $\varphi_t = \max\{\beta_t, \gamma_t\} \leq \max\{C_\beta, C_\gamma\} \leq C_\varphi$.

**Case 2: $k_2$ exists but $k_3$ does not:** by using the third line of eq. (64), for any $t \in (0, T)$, we have

$$\varphi_{t+1} \leq \beta_{t+1} + \gamma_{t+1} \leq C_\beta + \sum_{k=k_2}^t \frac{\|\nabla_y g(x_k, y_k)\|^2}{\beta_{k+1}} + C_\gamma, \quad (68)$$

where we take $\sum_{k=k_2}^t \frac{\|\nabla_y g(x_k, y_k)\|^2}{\beta_{k+1}} = 0$ for any $t < k_2$.

**Case 3: $k_3$ exists but $k_2$ does not:** from the second line of eq. (67), for any $t \in (0, T)$, we have

$$
\begin{aligned}
\varphi_{t+1} &\overset{(a)}{\leq} \varphi_t + \frac{\|\nabla_y g(x_t, y_t)\|^2}{\beta_{t+1} + \beta_t} + \frac{\|\nabla_v R(x_t, y_t, v_t)\|^2}{\varphi_{t+1}} \\
&\leq \varphi_{k_3} + \sum_{k=k_3}^t \frac{\|\nabla_y g(x_k, y_k)\|^2}{\beta_{k+1} + \beta_k} + \sum_{k=k_3}^t \frac{\|\nabla_v R(x_k, y_k, v_k)\|^2}{\varphi_{k+1}} \\
&\leq \beta_{k_3} + \gamma_{k_3} + \sum_{k=k_3}^t \frac{\|\nabla_y g(x_k, y_k)\|^2}{\beta_{k+1} + \beta_k} + \sum_{k=k_3}^t \frac{\|\nabla_v R(x_k, y_k, v_k)\|^2}{\varphi_{k+1}} \\
&\overset{(b)}{=} \beta_{t+1} + \gamma_{k_3} + \sum_{k=k_3}^t \frac{\|\nabla_v R(x_k, y_k, v_k)\|^2}{\varphi_{k+1}} \\
&\leq \beta_{t+1} + C_\gamma + \sum_{k=k_3}^t \frac{\|\nabla_v R(x_k, y_k, v_k)\|^2}{\varphi_{k+1}} \\
&\leq C_\beta + C_\gamma + \sum_{k=k_3}^t \frac{\|\nabla_v R(x_k, y_k, v_k)\|^2}{\varphi_{k+1}},
\end{aligned}
\tag{69}
$$

where (a) uses the second line of eq. (67); and we take $\sum_{k=k_3}^t \frac{\|\nabla_v R(x_k, y_k, v_k)\|^2}{\varphi_{k+1}} = 0$ for any $t < k_3$; (b) uses the first line of eq. (64).

**Case 4: both $k_2$ and $k_3$ exist:** from the third line of eq. (69), for any $t \in (0, T)$, we have

$$
\begin{aligned}
\varphi_{t+1} &\leq \beta_{k_3} + \gamma_{k_3} + \sum_{k=k_3}^t \frac{\|\nabla_y g(x_k, y_k)\|^2}{\beta_{k+1}} + \sum_{k=k_3}^t \frac{\|\nabla_v R(x_k, y_k, v_k)\|^2}{\varphi_{k+1}} \\
&\overset{(a)}{\leq} \beta_{k_2} + \sum_{k=k_2}^{k_3-1} \frac{\|\nabla_y g(x_k, y_k)\|^2}{\beta_{k+1}} + C_\gamma + \sum_{k=k_3}^t \frac{\|\nabla_y g(x_k, y_k)\|^2}{\beta_{k+1}} + \sum_{k=k_3}^t \frac{\|\nabla_v R(x_k, y_k, v_k)\|^2}{\varphi_{k+1}} \\
&= C_\beta + C_\gamma + \sum_{k=k_2}^t \frac{\|\nabla_y g(x_k, y_k)\|^2}{\beta_{k+1}} + \sum_{k=k_3}^t \frac{\|\nabla_v R(x_k, y_k, v_k)\|^2}{\varphi_{k+1}},
\end{aligned}
\tag{70}
$$

where (a) uses the third line of eq. (64); and we take $\sum_{k=k_2}^{k_3-1} \frac{\|\nabla_y g(x_k, y_k)\|^2}{\beta_{k+1}} = 0$ when $k_2 \geq k_3$, $\sum_{k=k_2}^t \frac{\|\nabla_y g(x_k, y_k)\|^2}{\beta_{k+1}} = 0$ for any $t < k_2$ and $\sum_{k=k_3}^t \frac{\|\nabla_v R(x_k, y_k, v_k)\|^2}{\gamma_{k+1}} = 0$ for any $t < k_3$. It is easy to see that the upper bound of $\varphi_{t+1}$ in eq. (70) is the largest among all cases. Thus, in the remaining proof, we only explore the upper bound of $\varphi_t$ in **Case 4**.

To further explore the bound of $\varphi_t$, we need to use some auxiliary results and bounds. So we split them into three parts as follows.

**Part I: an auxiliary bound of $\sum \frac{\|\bar{\nabla} f(x_k, y_k, v_k)\|^2}{\alpha_{k+1}^2}$.**

To further explore **Case 4**, we begin with a common term $\sum_{k=k_0}^t \frac{\|\bar{\nabla} f(x_k, y_k, v_k)\|^2}{\alpha_{k+1}^2}$ for any $k_0 \leq t$. Recall in Lemma 10, we have

$$
\|v_k\| \leq \frac{\sqrt{2}}{\mu} \varphi_{k+1} + \frac{\sqrt{2} C_{f_y}}{\mu} \sqrt{k} =: \bar{a} \varphi_{k+1} + \bar{b} \sqrt{k},
$$

where $\bar{a}$ and $\bar{b}$ refer to eq. (66). According to Lemma 1, since $\alpha_0 \geq 1$, for any integer $t > 0$, we have

$$
\begin{aligned}
\sum_{k=k_0}^t \frac{\|\bar{\nabla} f(x_k, y_k, v_k)\|^2}{\alpha_{k+1}^2} &\leq \sum_{k=0}^t \frac{\|\bar{\nabla} f(x_k, y_k, v_k)\|^2}{\alpha_{k+1}^2} \\
&\leq \log \left( \sum_{k=0}^t \|\bar{\nabla} f(x_k, y_k, v_k)\|^2 + \alpha_0^2 \right) + 1
\end{aligned}
$$

$$\overset{(a)}{\leq} \log\left(\sum_{k=0}^{t}\left(C_{g_{xy}}\bar{a}\varphi_{k+1} + C_{g_{xy}}\bar{b}\sqrt{k} + C_{f_x}\right)^2 + \alpha_0^2\right) + 1$$

$$\leq \log\left(\left(\sum_{k=0}^{t} C_{g_{xy}}\bar{a}\varphi_{k+1} + C_{g_{xy}}\bar{b}\sqrt{k} + C_{f_x} + \alpha_0\right)^2\right) + 1$$

$$= 2\log\left(\sum_{k=0}^{t} C_{g_{xy}}\bar{a}\varphi_{k+1} + C_{g_{xy}}\bar{b}\sqrt{k} + C_{f_x} + \alpha_0\right) + 1$$

$$\leq 2\log\left((t+1)(C_{g_{xy}}\bar{a}\varphi_{t+1} + C_{g_{xy}}\bar{b}\sqrt{t} + C_{f_x} + \alpha_0)\right) + 1$$

$$= 2\log(t+1) + 2\log\left(C_{g_{xy}}\bar{a}\varphi_{t+1} + C_{g_{xy}}\bar{b}\sqrt{t} + C_{f_x} + \alpha_0\right) + 1$$

$$\leq 2\log(t+1) + 2\log\left((C_{g_{xy}}\bar{a}\varphi_{t+1} + C_{g_{xy}}\bar{b} + C_{f_x} + \alpha_0)\sqrt{t}\right) + 1$$

$$\leq 3\log(t+1) + 2\log(C_{g_{xy}}\bar{a}\varphi_{t+1} + C_{g_{xy}}\bar{b} + C_{f_x} + \alpha_0) + 1, \tag{71}$$

where (a) follows from Remark 3 and Lemma 10. Therefore, we obtain the upper bound of $\sum_{k=k_0}^{t}\frac{\|\bar{\nabla}f(x_k,y_k,v_k)\|^2}{\alpha_{k+1}^2}$ for any $k_0 \leq t$ in eq. (71). **Part I** is completed.

**Part II: a more general bound of $\sum \frac{\|\nabla_y g(x_k,y_k)\|^2}{\beta_{k+1}}$.**

In Lemma 12, we show the bound of $\sum_{k=k_2}^{t}\frac{\|\nabla_y g(x_k,y_k)\|^2}{\beta_{k+1}}$ when $k_2$ exists. In **Part II**, we further provide a rough bound of $\sum_{k=\tilde{k}}^{t}\frac{\|\nabla_y g(x_k,y_k)\|^2}{\beta_{k+1}}$ for any potential $\tilde{k} \leq T$. Firstly, if $\tilde{k} \geq k_2$, it is easy to have

$$\sum_{k=\tilde{k}}^{t}\frac{\|\nabla_y g(x_k,y_k)\|^2}{\beta_{k+1}} \leq \sum_{k=k_2}^{t}\frac{\|\nabla_y g(x_k,y_k)\|^2}{\beta_{k+1}};$$

secondly, if $\tilde{k} < k_2$, we have

$$\sum_{k=\tilde{k}}^{t}\frac{\|\nabla_y g(x_k,y_k)\|^2}{\beta_{k+1}} \leq \sum_{k=\tilde{k}}^{k_2-1}\frac{\|\nabla_y g(x_k,y_k)\|^2}{\beta_{k+1}} + \sum_{k=k_2}^{t}\frac{\|\nabla_y g(x_k,y_k)\|^2}{\beta_{k+1}}$$

$$\leq \frac{\sum_{k=\tilde{k}}^{k_2-1}\|\nabla_y g(x_k,y_k)\|^2}{\beta_0} + \sum_{k=k_2}^{t}\frac{\|\nabla_y g(x_k,y_k)\|^2}{\beta_{k+1}}$$

$$\leq \frac{\beta_{k_2}^2 - \beta_{\tilde{k}}^2}{\beta_0} + \sum_{k=k_2}^{t}\frac{\|\nabla_y g(x_k,y_k)\|^2}{\beta_{k+1}}$$

$$\leq \frac{C_\beta^2 - \beta_0^2}{\beta_0} + \sum_{k=k_2}^{t}\frac{\|\nabla_y g(x_k,y_k)\|^2}{\beta_{k+1}}$$

$$= \frac{C_\beta^2}{\beta_0} - \beta_0 + \sum_{k=k_2}^{t}\frac{\|\nabla_y g(x_k,y_k)\|^2}{\beta_{k+1}}.$$

Combining these two situations, since $C_\beta \geq \beta_0$, for any $\tilde{k} \leq t$, we have

$$\sum_{k=\tilde{k}}^{t}\frac{\|\nabla_y g(x_k,y_k)\|^2}{\beta_{k+1}} \leq \frac{C_\beta^2}{\beta_0} - \beta_0 + \sum_{k=k_2}^{t}\frac{\|\nabla_y g(x_k,y_k)\|^2}{\beta_{k+1}}$$

$$\overset{(a)}{\leq} \frac{C_\beta^2}{\beta_0} - \beta_0 + \frac{(\mu+L_{g,1})C_\beta^2}{\mu^2} + \frac{(\mu+L_{g,1})^2 L_y^2}{\mu L_{g,1}\varphi_0}$$

$$+ \frac{(\mu+L_{g,1})^2 L_y^2}{\mu L_{g,1}}\sum_{k=k_2}^{t}\frac{\|\bar{\nabla}f(x_k,y_k,v_k)\|^2}{\alpha_{k+1}^2\varphi_{k+1}}, \tag{72}$$

where (a) uses Lemma 12. Thus, **Part II** is completed.

**Part III: the bound of $\varphi_t$ in Case 4.**

Here, we explore the upper bound of $\varphi_t$ in **Case 4**. Recalling eq. (70), we have

$$\varphi_{t+1} \leq C_\beta + C_\gamma + \sum_{k=k_2}^{t} \frac{\|\nabla_y g(x_k, y_k)\|^2}{\beta_{k+1}} + \sum_{k=k_3}^{t} \frac{\|\nabla_v R(x_k, y_k, v_k)\|^2}{\varphi_{k+1}} = C_\beta + C_\gamma = C_\varphi,$$

for $t \leq k_{\min} := \min\{k_2, k_3\}$. For $t > k_{\min}$, we have

$$\varphi_{t+1} \leq C_\beta + C_\gamma + \sum_{k=k_2}^{t} \frac{\|\nabla_y g(x_k, y_k)\|^2}{\beta_{k+1}} + \sum_{k=k_3}^{t} \frac{\|\nabla_v R(x_k, y_k, v_k)\|^2}{\varphi_{k+1}}$$

$$\overset{(a)}{\leq} C_\beta + C_\gamma + \sum_{k=k_2}^{t} \frac{\|\nabla_y g(x_k, y_k)\|^2}{\beta_{k+1}}$$

$$+ \frac{4(\mu + C_{g_{yy}})C_\beta^2}{\mu^4}\left(\frac{L_{g,2}C_{f_y}}{\mu} + L_{f,1}\right)^2 + \frac{4(\mu + C_{g_{yy}})C_\gamma^2}{\mu^2}$$

$$+ \frac{4(\mu + C_{g_{yy}})(\mu + L_{g,1})L_y^2}{\mu^3 L_{g,1}\varphi_0}\left(\frac{L_{g,2}C_{f_y}}{\mu} + L_{f,1}\right)^2 \sum_{k=k_2-1}^{k_3-2} \frac{\|\bar\nabla f(x_k, y_k, v_k)\|^2}{\alpha_{k+1}^2}$$

$$+ \frac{4(\mu + C_{g_{yy}})^2 L_v^2}{\mu C_{g_{yy}} C_\gamma} \sum_{k=k_3-1}^{t} \frac{\|\bar\nabla f(x_k, y_k, v_k)\|^2}{\alpha_{k+1}^2}$$

$$+ \left(\frac{4(\mu + C_{g_{yy}})^2}{\mu C_{g_{yy}}} + 8\right)\left(\frac{L_{g,2}C_{f_y}}{\mu} + L_{f,1}\right)^2 \frac{1}{\mu^2} \sum_{k=k_3-1}^{t} \frac{\|\nabla_y g(x_k, y_k)\|^2}{\beta_{k+1}}$$

$$\overset{(b)}{\leq} \left[\left(\frac{4(\mu + C_{g_{yy}})^2}{\mu C_{g_{yy}}} + 8\right)\left(\frac{L_{g,2}C_{f_y}}{\mu} + L_{f,1}\right)^2 \frac{1}{\mu^2} + 1\right] \sum_{k=k_2}^{t} \frac{\|\nabla_y g(x_k, y_k)\|^2}{\beta_{k+1}}$$

$$+ C_\beta + C_\gamma + \frac{4(\mu + C_{g_{yy}})C_\beta^2}{\mu^4}\left(\frac{L_{g,2}C_{f_y}}{\mu} + L_{f,1}\right)^2 + \frac{4(\mu + C_{g_{yy}})C_\gamma^2}{\mu^2}$$

$$+ \left(\frac{4(\mu + C_{g_{yy}})^2}{\mu C_{g_{yy}}} + 8\right)\left(\frac{L_{g,2}C_{f_y}}{\mu} + L_{f,1}\right)^2 \frac{1}{\mu^2}\left(\frac{C_\beta^2}{\beta_0} - \beta_0\right)$$

$$+ \frac{4(\mu + C_{g_{yy}})(\mu + L_{g,1})L_y^2}{\mu^3 L_{g,1}\varphi_0}\left(\frac{L_{g,2}C_{f_y}}{\mu} + L_{f,1}\right)^2 \sum_{k=k_2-1}^{t} \frac{\|\bar\nabla f(x_k, y_k, v_k)\|^2}{\alpha_{k+1}^2}$$

$$+ \frac{4(\mu + C_{g_{yy}})^2 L_v^2}{\mu C_{g_{yy}} C_\gamma} \sum_{k=k_3-1}^{t} \frac{\|\bar\nabla f(x_k, y_k, v_k)\|^2}{\alpha_{k+1}^2}$$

$$\overset{(c)}{\leq} \left(\left(\frac{4(\mu + C_{g_{yy}})^2}{\mu C_{g_{yy}}} + 8\right)\left(\frac{L_{g,2}C_{f_y}}{\mu} + L_{f,1}\right)^2 \frac{1}{\mu^2} + 1\right)\frac{(\mu + L_{g,1})^2 L_y^2}{\mu L_{g,1}} \sum_{k=k_2}^{t} \frac{\|\bar\nabla f(x_k, y_k, v_k)\|^2}{\alpha_{k+1}^2 \varphi_{k+1}}$$

$$+ \frac{4(\mu + C_{g_{yy}})(\mu + L_{g,1})L_y^2}{\mu^3 L_{g,1}\varphi_0}\left(\frac{L_{g,2}C_{f_y}}{\mu} + L_{f,1}\right)^2 \sum_{k=k_2-1}^{t} \frac{\|\bar\nabla f(x_k, y_k, v_k)\|^2}{\alpha_{k+1}^2}$$

$$+ \frac{4(\mu + C_{g_{yy}})^2 L_v^2}{\mu C_{g_{yy}} C_\gamma} \sum_{k=k_3-1}^{t} \frac{\|\bar\nabla f(x_k, y_k, v_k)\|^2}{\alpha_{k+1}^2}$$

$$+ C_\beta + C_\gamma + \frac{4(\mu + C_{g_{yy}})C_\beta^2}{\mu^4}\left(\frac{L_{g,2}C_{f_y}}{\mu} + L_{f,1}\right)^2 + \frac{4(\mu + C_{g_{yy}})C_\gamma^2}{\mu^2}$$

$$+ \left(\frac{4(\mu + C_{g_{yy}})^2}{\mu C_{g_{yy}}} + 8\right)\left(\frac{L_{g,2}C_{f_y}}{\mu} + L_{f,1}\right)^2 \frac{1}{\mu^2}\left(\frac{C_\beta^2}{\beta_0} - \beta_0\right)$$

$$+ \left[\left(\frac{4(\mu + C_{g_{yy}})^2}{\mu C_{g_{yy}}} + 8\right)\left(\frac{L_{g,2}C_{f_y}}{\mu} + L_{f,1}\right)^2 \frac{1}{\mu^2} + 1\right]\left(\frac{(\mu + L_{g,1})C_\beta^2}{\mu^2} + \frac{(\mu + L_{g,1})^2 L_y^2}{\mu L_{g,1}\varphi_0}\right)$$

$$\leq \left[\left(\left(\frac{4(\mu + C_{g_{yy}})^2}{\mu C_{g_{yy}}} + 8\right)\left(\frac{L_{g,2}C_{f_y}}{\mu} + L_{f,1}\right)^2 \frac{1}{\mu^2} + 1\right)\frac{(\mu + L_{g,1})^2 L_y^2}{\mu L_{g,1}C_\beta}\right.$$

$$+ \frac{4(\mu + C_{g_{yy}})(\mu + L_{g,1})L_y^2}{\mu^3 L_{g,1}\varphi_0}\left(\frac{L_{g,2}C_{f_y}}{\mu} + L_{f,1}\right)^2\right]\sum_{k=k_2-1}^{t}\frac{\|\bar{\nabla}f(x_k, y_k, v_k)\|^2}{\alpha_{k+1}^2}$$

$$+ \frac{4(\mu + C_{g_{yy}})^2 L_v^2}{\mu C_{g_{yy}}\gamma_0}\sum_{k=k_3-1}^{t}\frac{\|\bar{\nabla}f(x_k, y_k, v_k)\|^2}{\alpha_{k+1}^2}$$

$$+ C_\beta + C_\gamma + \frac{4(\mu + C_{g_{yy}})C_\beta^2}{\mu^4}\left(\frac{L_{g,2}C_{f_y}}{\mu} + L_{f,1}\right)^2 + \frac{4(\mu + C_{g_{yy}})C_\gamma^2}{\mu^2}$$

$$+ \left(\frac{4(\mu + C_{g_{yy}})^2}{\mu C_{g_{yy}}} + 8\right)\left(\frac{L_{g,2}C_{f_y}}{\mu} + L_{f,1}\right)^2\frac{1}{\mu^2}\left(\frac{C_\beta^2}{\beta_0} - \beta_0\right)$$

$$+ \left[\left(\frac{4(\mu + C_{g_{yy}})^2}{\mu C_{g_{yy}}} + 8\right)\left(\frac{L_{g,2}C_{f_y}}{\mu} + L_{f,1}\right)^2\frac{1}{\mu^2} + 1\right]\left(\frac{(\mu + L_{g,1})C_\beta^2}{\mu^2} + \frac{(\mu + L_{g,1})^2 L_y^2}{\mu L_{g,1}\varphi_0}\right)$$

$$\overset{(d)}{=:} a_0 \sum_{k=\min\{k_2-1,k_3-1\}}^{t}\frac{\|\bar{\nabla}f(x_k, y_k, v_k)\|^2}{\alpha_{k+1}^2} + b_0$$

$$\leq a_0 \sum_{k=\min\{k_2,k_3\}}^{t}\frac{\|\bar{\nabla}f(x_k, y_k, v_k)\|^2}{\alpha_{k+1}^2} + a_0 + b_0$$

$$\overset{(e)}{\leq} a_0\left[3\log(t+1) + 2\log\left(\varphi_{t+1} + \frac{C_{g_{xy}}\bar{b} + C_{f_x} + \alpha_0}{C_{g_{xy}}\bar{a}}\right) + 2\log(C_{g_{xy}}\bar{a}) + 1\right] + a_0 + b_0, \quad (73)$$

where (a) uses Lemma 13; (b) uses the first line in eq. (72) by replacing $\tilde{k}$ with $k_3 - 1$; (c) results from eq. (52); (d) refers to eq. (66); (e) uses eq. (71). Since $\min\{k_2, k_3\} \leq T$, we have $\varphi_{t+1} \geq \min\{C_\beta, C_\gamma\} \geq \max\{64a_0^2, 1\}$, which indicate that

(i) if $8a_0 \leq 1$, we have

$$4a_0 \log(\varphi_{t+1}) \leq \frac{\log(\varphi_{t+1})}{2} \leq \frac{\varphi_{t+1}}{2} \leq \varphi_{t+1};$$

(ii) if $8a_0 > 1$, we have

$$\varphi_{t+1} - 4a_0 \log(\varphi_{t+1}) = \varphi_{t+1} - 8a_0\log(\sqrt{\varphi_{t+1}}) \geq 8a_0\left(\sqrt{\varphi_{t+1}} - \log(\sqrt{\varphi_{t+1}})\right) \geq 0.$$

Combining (i) and (ii), we have $4a_0 \log(\varphi_{t+1}) \leq \varphi_{t+1}$. Then we obtain

$$\varphi_{t+1} \leq a_0\left[3\log(t+1) + 2\log\left(\varphi_{t+1} + \frac{C_{g_{xy}}\bar{b} + C_{f_x} + \alpha_0}{C_{g_{xy}}\bar{a}}\right) + 2\log(C_{g_{xy}}\bar{a}) + 1\right] + a_0 + b_0$$

$$\leq a_0\left[3\log(t+1) + 2\log(\varphi_{t+1}) + 2\log\left(1 + \frac{C_{g_{xy}}\bar{b} + C_{f_x} + \alpha_0}{C_{g_{xy}}\bar{a}}\right) + 2\log(C_{g_{xy}}\bar{a}) + 1\right] + a_0 + b_0$$

$$\leq \frac{1}{2}\varphi_{t+1} + a_0\left[3\log(t+1) + 2\log\left(1 + \frac{C_{g_{xy}}\bar{b} + C_{f_x} + \alpha_0}{C_{g_{xy}}\bar{a}}\right) + 2\log(C_{g_{xy}}\bar{a}) + 1\right] + a_0 + b_0,$$

which indicates that

$$\varphi_{t+1} \leq 6a_0\log(t+1) + 4a_0\log\left(1 + \frac{C_{g_{xy}}\bar{b} + C_{f_x} + \alpha_0}{C_{g_{xy}}\bar{a}}\right) + 4a_0\log(C_{g_{xy}}\bar{a}) + 4a_0 + 2b_0$$

$$\overset{(a)}{=:} a_1\log(t+1) + b_1, \quad (74)$$

where (a) refers to eq. (65). Thus, **Part III** is completed and the proof of this lemma is completed. $\square$

**Lemma 16.** *Under Assumptions 1, 2, for any integer $k_0 \in [0, t)$, we have the upper bounds in terms of logarithmic functions as*

$$\sum_{k=k_0}^{t}\frac{\|\bar{\nabla}f(x_k, y_k, v_k)\|^2}{\alpha_{k+1}^2} \leq 5\log(t+1) + c_2,$$

$$\sum_{k=k_0}^{t}\frac{\|\nabla_y g(x_k, y_k)\|^2}{\beta_{k+1}} \leq a_2\log(t+1) + b_2,$$

$$\sum_{k=k_0}^{t} \frac{\|\nabla_v R(x_k, y_k, v_k)\|^2}{\varphi_{k+1}} \leq a_3 \log(t+1) + b_3,$$

*where referring to eq.* (65), *eq.* (66), $c_2$, $a_2$, $b_2$, $a_3$, $b_3$ *are defined as*

$$c_2 := 2 \log \left( C_{g_{xy}} \bar{a} a_1 + C_{g_{xy}} \bar{a} b_1 + C_{g_{xy}} \bar{b} + C_{f_x} + \alpha_0 \right) + 1,$$

$$a_2 := \frac{5(\mu + L_{g,1})^2 L_y^2}{\mu L_{g,1} C_\beta}, \quad b_2 := \frac{(\mu + L_{g,1})^2 L_y^2}{\mu L_{g,1} C_\beta} c_2 + \left( \frac{C_\beta^2}{\beta_0} - \beta_0 + \frac{(\mu + L_{g,1}) C_\beta^2}{\mu^2} + \frac{(\mu + L_{g,1})^2 L_y^2}{\mu L_{g,1} \varphi_0} \right),$$

$$a_3 := \frac{20(\mu + C_{g_{yy}})(\mu + L_{g,1}) L_y^2}{\mu^3 L_{g,1} \varphi_0} \left( \frac{L_{g,2} C_{f_y}}{\mu} + L_{f,1} \right)^2 + \frac{20(\mu + C_{g_{yy}})^2 L_v^2}{\mu C_{g_{yy}} C_\gamma}$$

$$+ \left( \frac{4(\mu + C_{g_{yy}})^2}{\mu C_{g_{yy}}} + 8 \right) \left( \frac{L_{g,2} C_{f_y}}{\mu} + L_{f,1} \right)^2 \frac{a_2}{\mu^2},$$

$$b_3 := \frac{C_\gamma^2}{\gamma_0} - \gamma_0 + \frac{4(\mu + C_{g_{yy}}) C_\beta^2}{\mu^4} \left( \frac{L_{g,2} C_{f_y}}{\mu} + L_{f,1} \right)^2 + \frac{4(\mu + C_{g_{yy}}) C_\gamma^2}{\mu^2}$$

$$+ \left( \frac{4(\mu + C_{g_{yy}})(\mu + L_{g,1}) L_y^2}{\mu^3 L_{g,1} \varphi_0} \left( \frac{L_{g,2} C_{f_y}}{\mu} + L_{f,1} \right)^2 + \frac{4(\mu + C_{g_{yy}})^2 L_v^2}{\mu C_{g_{yy}} C_\gamma} \right) c_2$$

$$+ \left( \frac{4(\mu + C_{g_{yy}})^2}{\mu C_{g_{yy}}} + 8 \right) \left( \frac{L_{g,2} C_{f_y}}{\mu} + L_{f,1} \right)^2 \frac{b_2}{\mu^2}. \tag{75}$$

*Proof.* Based on the logarithmic-function form bound in Lemma 15, we can further have the logarithmic-function form bounds of the components in Lemma 11 as the following 3 parts.

**Part I: the bound of $\sum \frac{\|\bar{\nabla} f(x_k, y_k, v_k)\|^2}{\alpha_{k+1}^2}$ in terms of logarithmic function.**

Firstly, we bound $\sum_{k=k_0}^{t} \frac{\|\bar{\nabla} f(x_k, y_k, v_k)\|^2}{\alpha_{k+1}^2}$ for arbitrary $k_0 < t$. Back to eq. (71), by plugging in eq. (74), we have

$$\sum_{k=k_0}^{t} \frac{\|\bar{\nabla} f(x_k, y_k, v_k)\|^2}{\alpha_{k+1}^2}$$

$$\leq 3 \log(t+1) + 2 \log(C_{g_{xy}} \bar{a} \varphi_{t+1} + C_{g_{xy}} \bar{b} + C_{f_x} + \alpha_0) + 1$$

$$\overset{(a)}{\leq} 3 \log(t+1) + 2 \log \left( C_{g_{xy}} \bar{a} a_1 \log(t+1) + C_{g_{xy}} \bar{a} b_1 + C_{g_{xy}} \bar{b} + C_{f_x} + \alpha_0 \right) + 1$$

$$\leq 3 \log(t+1) + 2 \log \left( C_{g_{xy}} \bar{a} a_1 (t+1) + C_{g_{xy}} \bar{a} b_1 + C_{g_{xy}} \bar{b} + C_{f_x} + \alpha_0 \right) + 1$$

$$\leq 3 \log(t+1) + 2 \log \left( (C_{g_{xy}} \bar{a} a_1 + C_{g_{xy}} \bar{a} b_1 + C_{g_{xy}} \bar{b} + C_{f_x} + \alpha_0)(t+1) \right) + 1$$

$$\leq 5 \log(t+1) + 2 \log \left( C_{g_{xy}} \bar{a} a_1 + C_{g_{xy}} \bar{a} b_1 + C_{g_{xy}} \bar{b} + C_{f_x} + \alpha_0 \right) + 1$$

$$\overset{(b)}{=:} 5 \log(t+1) + c_2, \tag{76}$$

where (a) results from eq. (74); (b) refers to eq. (75).

**Part II: the bound of $\sum \frac{\|\nabla_y g(x_k, y_k)\|^2}{\beta_{k+1}}$ in terms of logarithmic function.**

Secondly, we bound $\sum_{k=k_0}^{t} \frac{\|\nabla_y g(x_k, y_k)\|^2}{\beta_{k+1}}$. We split this part into two cases using Lemma 4.

**Case 1:** If $\beta_{t+1} \leq C_\beta$, we have

$$\sum_{k=k_0}^{t} \frac{\|\nabla_y g(x_k, y_k)\|^2}{\beta_{k+1}} \leq \frac{\sum_{k=k_0}^{t} \|\nabla_y g(x_k, y_k)\|^2}{\beta_0} \leq \frac{\beta_{t+1}^2 - \beta_{k_0}^2}{\beta_0} \leq \frac{C_\beta^2 - \beta_0^2}{\beta_0} = \frac{C_\beta^2}{\beta_0} - \beta_0 \leq b_2.$$

**Case 2:** If $\beta_{t+1} > C_\beta$, we have $k_2 \leq t$, where $k_2$ refers to Lemma 4. Then we can use eq. (72), which indicates

$$\sum_{k=k_0}^{t} \frac{\|\nabla_y g(x_k, y_k)\|^2}{\beta_{k+1}}$$

$$\leq \left( \frac{C_\beta^2}{\beta_0} - \beta_0 + \frac{(\mu + L_{g,1})C_\beta^2}{\mu^2} + \frac{(\mu + L_{g,1})^2 L_y^2}{\mu L_{g,1}\varphi_0} \right) + \frac{(\mu + L_{g,1})^2 L_y^2}{\mu L_{g,1}C_\beta} \sum_{k=k_2}^{t} \frac{\|\bar{\nabla} f(x_k, y_k, v_k)\|^2}{\alpha_{k+1}^2}$$

$$\leq \frac{5(\mu + L_{g,1})^2 L_y^2}{\mu L_{g,1}C_\beta} \log(t+1) + \frac{(\mu + L_{g,1})^2 L_y^2}{\mu L_{g,1}C_\beta} c_2 + \left( \frac{C_\beta^2}{\beta_0} - \beta_0 + \frac{(\mu + L_{g,1})C_\beta^2}{\mu^2} + \frac{(\mu + L_{g,1})^2 L_y^2}{\mu L_{g,1}\varphi_0} \right)$$

$$\overset{(a)}{=:} a_2 \log(t+1) + b_2, \tag{77}$$

where the second inequality uses (76), and (a) refers to eq. (75). Since the upper bound of **Case 2** is larger, we take eq. (77) as our final result.

**Part III: the bound of $\sum \frac{\|\nabla_v R(x_k, y_k, v_k)\|^2}{\varphi_{k+1}}$ in terms of logarithmic function.**

Last, we bound $\sum_{k=k_0}^{t} \frac{\|\nabla_v R(x_k, y_k, v_k)\|^2}{\varphi_{k+1}}$. We split this part into two cases using Lemma 4.

**Case 1:** If $\gamma_{t+1} \leq C_\gamma$, we have

$$\sum_{k=k_0}^{t} \frac{\|\nabla_v R(x_k, y_k, v_k)\|^2}{\varphi_{k+1}} \leq \frac{\sum_{k=k_0}^{t} \|\nabla_v R(x_k, y_k, v_k)\|^2}{\varphi_0} \leq \frac{C_\gamma^2 - \gamma_0^2}{\gamma_0} \leq \frac{C_\gamma^2}{\gamma_0} - \gamma_0 \leq b_3.$$

**Case 2:** If $\gamma_{t+1} > C_\gamma$, we have $k_3 \leq t$, where $k_3$ refers to Lemma 4.

$$\sum_{k=k_0}^{t} \frac{\|\nabla_v R(x_k, y_k, v_k)\|^2}{\varphi_{k+1}}$$

$$\overset{(a)}{\leq} \sum_{k=k_0}^{k_3-1} \frac{\|\nabla_v R(x_k, y_k, v_k)\|^2}{\varphi_{k+1}} + \sum_{k=k_3}^{t} \frac{\|\nabla_v R(x_k, y_k, v_k)\|^2}{\varphi_{k+1}}$$

$$\overset{(b)}{\leq} \frac{C_\gamma^2}{\gamma_0} - \gamma_0 + \frac{4(\mu + C_{g_{yy}})C_\beta^2}{\mu^4} \left( \frac{L_{g,2}C_{f_y}}{\mu} + L_{f,1} \right)^2 + \frac{4(\mu + C_{g_{yy}})C_\gamma^2}{\mu^2}$$

$$+ \frac{4(\mu + C_{g_{yy}})(\mu + L_{g,1})L_y^2}{\mu^3 L_{g,1}\varphi_0} \left( \frac{L_{g,2}C_{f_y}}{\mu} + L_{f,1} \right)^2 \sum_{k=k_2-1}^{k_3-2} \frac{\|\bar{\nabla} f(x_k, y_k, v_k)\|^2}{\alpha_{k+1}^2}$$

$$+ \frac{4(\mu + C_{g_{yy}})^2 L_v^2}{\mu C_{g_{yy}} C_\gamma} \sum_{k=k_3-1}^{t} \frac{\|\bar{\nabla} f(x_k, y_k, v_k)\|^2}{\alpha_{k+1}^2}$$

$$+ \left( \frac{4(\mu + C_{g_{yy}})^2}{\mu C_{g_{yy}}} + 8 \right) \left( \frac{L_{g,2}C_{f_y}}{\mu} + L_{f,1} \right)^2 \frac{1}{\mu^2} \sum_{k=k_3-1}^{t} \frac{\|\nabla_y g(x_k, y_k)\|^2}{\beta_{k+1}}$$

$$\overset{(c)}{\leq} \frac{C_\gamma^2}{\gamma_0} - \gamma_0 + \frac{4(\mu + C_{g_{yy}})C_\beta^2}{\mu^4} \left( \frac{L_{g,2}C_{f_y}}{\mu} + L_{f,1} \right)^2 + \frac{4(\mu + C_{g_{yy}})C_\gamma^2}{\mu^2}$$

$$+ \left( \frac{4(\mu + C_{g_{yy}})(\mu + L_{g,1})L_y^2}{\mu^3 L_{g,1}\varphi_0} \left( \frac{L_{g,2}C_{f_y}}{\mu} + L_{f,1} \right)^2 + \frac{4(\mu + C_{g_{yy}})^2 L_v^2}{\mu C_{g_{yy}} C_\gamma} \right) \left( 5 \log(t+1) + c_2 \right)$$

$$+ \left( \frac{4(\mu + C_{g_{yy}})^2}{\mu C_{g_{yy}}} + 8 \right) \left( \frac{L_{g,2}C_{f_y}}{\mu} + L_{f,1} \right)^2 \frac{1}{\mu^2} \left( a_2 \log(t+1) + b_2 \right)$$

$$\overset{(d)}{=:} a_3 \log(t+1) + b_3, \tag{78}$$

where (a) allows $\sum_{k=k_0}^{k_3-1} \frac{\|\nabla_v R(x_k, y_k, v_k)\|^2}{\varphi_{k+1}} = 0$ when $k_0 \geq k_3$; (b) uses $C_\gamma \geq \gamma_0$ and Lemma 13; (c) follows from eq. (76) and eq. (77); (d) refers to eq. (75). Since the upper bound of **Case 2** is larger, we take eq. (78) as our final result.

Thus, the proof is complete. $\qquad \square$

Next, we show the upper bound of $\alpha_t$.

**Lemma 17** (The upper bound of $\alpha_t$). *Under Assumptions 1, 2, 3, suppose the number of total iteration rounds in Algorithm 2 is $T$. If there exists $k_1 \leq T$ described in Lemma 4, we have*

$$
\begin{cases}
\alpha_t \leq C_\alpha, & t \leq k_1; \\
\alpha_t \leq C_\alpha + \Big(a_4 \log(t) + b_4 + 4\big(\Phi(x_0) - \inf_x \Phi(x)\big)\Big)\varphi_t, & t \geq k_1,
\end{cases}
$$

*where $a_4$, $b_4$ are defined as*

$$
\begin{aligned}
a_4 &:= \frac{2\bar{L}^2 a_2}{\mu^2 C_\alpha}\Big[1 + \frac{2}{\mu^2}\big(\frac{L_{g,2}C_{f_y}}{\mu} + L_{f,1}\big)^2\Big] + \frac{4\bar{L}^2 a_3}{\mu^2 C_\alpha} \\
b_4 &:= \frac{2\bar{L}^2 b_2}{\mu^2 C_\alpha}\Big[1 + \frac{2}{\mu^2}\big(\frac{L_{g,2}C_{f_y}}{\mu} + L_{f,1}\big)^2\Big] + \frac{4\bar{L}^2 b_3}{\mu^2 C_\alpha} + \frac{2L_\Phi}{\varphi_0^2}\frac{C_\alpha^2}{\alpha_0^2},
\end{aligned}
\tag{79}
$$

*and the upper bound of $\varphi_t := \max\{\beta_t, \gamma_t\}$ refers to Lemma 15. When such $k_1$ does not exist, we have $\alpha_t \leq C_\alpha$ for any $t \leq T$.*

*Proof.* According to Lemma 4, the proof can be split into the following three cases.

**Case 1:** if $\alpha_T \leq C_\alpha$, for any $t < T$, we have the upper bound of $\alpha_{t+1}$ as $\alpha_{t+1} \leq C_\alpha$.

**Case 2:** if $\alpha_T > C_\alpha$, there exists $k_1 \leq T$ described in Lemma 4. Then we have the upper bound of $\alpha_{t+1}$ as $\alpha_{t+1} \leq C_\alpha$ for any $t < k_1$.

**Case 3:** in the remaining proof, we only consider and explore the case $k_1 \leq t \leq T$ when $\alpha_T > C_\alpha$. From Lemma 11, for $k \geq k_1$, we have

$$
\begin{aligned}
\Phi(x_{k+1}) \leq &\Phi(x_k) - \frac{1}{2\alpha_{k+1}\varphi_{k+1}}\|\nabla\Phi(x_k)\|^2 - \frac{1}{4\alpha_{k+1}\varphi_{k+1}}\|\bar{\nabla}f(x_k, y_k, v_k)\|^2 \\
&+ \frac{\bar{L}^2}{2\mu^2}\Big[1 + \frac{2}{\mu^2}\big(\frac{L_{g,2}C_{f_y}}{\mu} + L_{f,1}\big)^2\Big]\frac{\|\nabla_y g(x_k, y_k)\|^2}{\alpha_{k+1}\varphi_{k+1}} + \frac{\bar{L}^2}{\mu^2}\frac{\|\nabla_v R(x_k, y_k, v_k)\|^2}{\alpha_{k+1}\varphi_{k+1}}
\end{aligned}
$$

which indicates that

$$
\begin{aligned}
\frac{\|\bar{\nabla}f(x_k, y_k, v_k)\|^2}{\alpha_{k+1}\varphi_{k+1}} \leq &4\big(\Phi(x_k) - \Phi(x_{k+1})\big) \\
&+ \frac{2\bar{L}^2}{\mu^2}\Big[1 + \frac{2}{\mu^2}\big(\frac{L_{g,2}C_{f_y}}{\mu} + L_{f,1}\big)^2\Big]\frac{\|\nabla_y g(x_k, y_k)\|^2}{\alpha_{k+1}\varphi_{k+1}} + \frac{4\bar{L}^2}{\mu^2}\frac{\|\nabla_v R(x_k, y_k, v_k)\|^2}{\alpha_{k+1}\varphi_{k+1}}.
\end{aligned}
$$

By taking summation, we have

$$
\begin{aligned}
&\sum_{k=k_1}^{t} \frac{\|\bar{\nabla}f(x_k, y_k, v_k)\|^2}{\alpha_{k+1}\varphi_{k+1}} \\
&\leq 4\big(\Phi(x_{k_1}) - \inf_x \Phi(x)\big) + \frac{2\bar{L}^2}{\mu^2 C_\alpha}\Big[1 + \frac{2}{\mu^2}\big(\frac{L_{g,2}C_{f_y}}{\mu} + L_{f,1}\big)^2\Big]\sum_{k=k_1}^{t}\frac{\|\nabla_y g(x_k, y_k)\|^2}{\varphi_{k+1}} \\
&+ \frac{4\bar{L}^2}{\mu^2 C_\alpha}\sum_{k=k_1}^{t}\frac{\|\nabla_v R(x_k, y_k, v_k)\|^2}{\varphi_{k+1}}.
\end{aligned}
\tag{80}
$$

For $\Phi(x_{k_1})$, by telescoping eq. (43) in Lemma 11, we get

$$
\begin{aligned}
\Phi(x_{k_1}) \leq &\Phi(x_0) + \frac{L_\Phi}{2}\sum_{k=0}^{k_1-1}\frac{\|\bar{\nabla}f(x_t, y_t, v_t)\|^2}{\alpha_{t+1}^2\varphi_{t+1}^2} \\
&+ \frac{\bar{L}^2}{2\mu^2}\Big[1 + \frac{2}{\mu^2}\big(\frac{L_{g,2}C_{f_y}}{\mu} + L_{f,1}\big)^2\Big]\sum_{k=0}^{k_1-1}\frac{\|\nabla_y g(x_t, y_t)\|^2}{\alpha_{t+1}\varphi_{t+1}} \\
&+ \frac{\bar{L}^2}{\mu^2}\sum_{k=0}^{k_1-1}\frac{\|\nabla_v R(x_t, y_t, v_t)\|^2}{\alpha_{t+1}\varphi_{t+1}}.
\end{aligned}
\tag{81}
$$

By plugging eq. (81) into eq. (80), we have

$$
\sum_{k=k_1}^{t} \frac{\|\bar{\nabla}f(x_k, y_k, v_k)\|^2}{\alpha_{k+1}\varphi_{k+1}}
$$

$$
\leq 4\big(\Phi(x_0) - \inf_x \Phi(x)\big) + \frac{2\bar{L}^2}{\mu^2 C_\alpha}\left[1 + \frac{2}{\mu^2}\Big(\frac{L_{g,2}C_{f_y}}{\mu} + L_{f,1}\Big)^2\right]\sum_{k=0}^{t} \frac{\|\nabla_y g(x_k, y_k)\|^2}{\varphi_{k+1}}
$$

$$
+ \frac{4\bar{L}^2}{\mu^2 C_\alpha}\sum_{k=0}^{t} \frac{\|\nabla_v R(x_k, y_k, v_k)\|^2}{\varphi_{k+1}} + \frac{2L_\Phi}{\varphi_0^2}\frac{C_\alpha^2}{\alpha_0^2}
$$

$$
\leq 4\big(\Phi(x_0) - \inf_x \Phi(x)\big) + \frac{2\bar{L}^2}{\mu^2 C_\alpha}\left[1 + \frac{2}{\mu^2}\Big(\frac{L_{g,2}C_{f_y}}{\mu} + L_{f,1}\Big)^2\right]\sum_{k=0}^{t} \frac{\|\nabla_y g(x_k, y_k)\|^2}{\beta_{k+1}}
$$

$$
+ \frac{4\bar{L}^2}{\mu^2 C_\alpha}\sum_{k=0}^{t} \frac{\|\nabla_v R(x_k, y_k, v_k)\|^2}{\varphi_{k+1}} + \frac{2L_\Phi}{\varphi_0^2}\frac{C_\alpha^2}{\alpha_0^2}
$$

$$
\overset{(a)}{\leq} 4\big(\Phi(x_0) - \inf_x \Phi(x)\big) + \frac{2\bar{L}^2}{\mu^2 C_\alpha}\left[1 + \frac{2}{\mu^2}\Big(\frac{L_{g,2}C_{f_y}}{\mu} + L_{f,1}\Big)^2\right]\big(a_2 \log(t+1) + b_2\big)
$$

$$
+ \frac{4\bar{L}^2}{\mu^2 C_\alpha}\big(a_3 \log(t+1) + b_3\big) + \frac{2L_\Phi}{\varphi_0^2}\frac{C_\alpha^2}{\alpha_0^2}
$$

$$
\overset{(b)}{=:} a_4 \log(t+1) + b_4 + 4\big(\Phi(x_0) - \inf_x \Phi(x)\big), \tag{82}
$$

where (a) plugs in eq. (77) and eq. (78); (b) refers to eq. (79). This immediately implies

$$
\sum_{k=k_1}^{t} \frac{\|\bar{\nabla}f(x_k, y_k, v_k)\|^2}{\alpha_{k+1}} \leq \Big(a_4 \log(t+1) + b_4 + 4\big(\Phi(x_0) - \inf_x \Phi(x)\big)\Big)\varphi_{t+1}. \tag{83}
$$

Similarly, we can have the upper bound of $\alpha_{t+1}$ as

$$
\alpha_{t+1} \leq \alpha_{k_1} + \sum_{k=k_1}^{t} \frac{\|\bar{\nabla}f(x_k, y_k, v_k)\|^2}{\alpha_{k+1}}
$$

$$
\leq C_\alpha + \Big(a_4 \log(t+1) + b_4 + 4\big(\Phi(x_0) - \inf_x \Phi(x)\big)\Big)\varphi_{t+1}. \tag{84}
$$

Then the upper bound of $\alpha_{t+1}$ is proved. $\qquad\square$

### F.4 Proof of Theorem 2

Here we still assume the total iteration rounds of Algorithm 2 is $T$. According to Lemma 4, the proof can be split into the following two cases.

**Case 1:** If $\alpha_T \leq C_\alpha$, then by Lemma 11 and Lemma 17, we have

$$
\frac{\|\nabla\Phi(x_t)\|^2}{\alpha_{t+1}\varphi_{t+1}} \leq 2\big(\Phi(x_t) - \Phi(x_{t+1})\big) + \frac{L_\Phi}{\alpha_{t+1}^2\varphi_{t+1}^2}\|\bar{\nabla}f(x_t, y_t, v_t)\|^2
$$

$$
+ \frac{\bar{L}^2}{\mu^2}\left[1 + \frac{2}{\mu^2}\Big(\frac{L_{g,2}C_{f_y}}{\mu} + L_{f,1}\Big)^2\right]\frac{\|\nabla_y g(x_t, y_t)\|^2}{\alpha_{t+1}\varphi_{t+1}} + \frac{2\bar{L}^2}{\mu^2}\frac{\|\nabla_v R(x_t, y_t, v_t)\|^2}{\alpha_{t+1}\varphi_{t+1}},
$$

By taking the average, we have

$$
\frac{1}{T}\sum_{t=0}^{T-1} \frac{\|\nabla\Phi(x_t)\|^2}{\alpha_{t+1}\varphi_{t+1}} \leq \frac{2}{T}\big(\Phi(x_0) - \Phi(x_T)\big) + \frac{L_\Phi}{\alpha_0^2\varphi_0^2}\frac{1}{T}\sum_{t=0}^{T-1}\|\bar{\nabla}f(x_t, y_t, v_t)\|^2
$$

$$
+ \frac{\bar{L}^2}{\mu^2}\left[1 + \frac{2}{\mu^2}\Big(\frac{L_{g,2}C_{f_y}}{\mu} + L_{f,1}\Big)^2\right]\frac{1}{T}\sum_{t=0}^{T-1} \frac{\|\nabla_y g(x_t, y_t)\|^2}{\alpha_{t+1}\varphi_{t+1}}
$$

$$+ \frac{2\bar{L}^2}{\mu^2} \frac{1}{T} \sum_{t=0}^{T-1} \frac{\left\| \nabla_v R(x_t, y_t, v_t) \right\|^2}{\alpha_{t+1}\varphi_{t+1}}$$

$$\leq \frac{2}{T}\big(\Phi(x_0) - \Phi(x_T)\big) + \frac{L_\Phi C_\alpha^2}{T\alpha_0^2\varphi_0^2}$$

$$+ \frac{\bar{L}^2}{\mu^2\alpha_0 T}\left[1 + \frac{2}{\mu^2}\Big(\frac{L_{g,2}C_{f_y}}{\mu} + L_{f,1}\Big)^2\right]\sum_{t=0}^{T-1}\frac{\left\| \nabla_y g(x_t, y_t) \right\|^2}{\beta_{t+1}}$$

$$+ \frac{2\bar{L}^2}{\mu^2\alpha_0 T}\sum_{t=0}^{T-1}\frac{\left\| \nabla_v R(x_t, y_t, v_t) \right\|^2}{\varphi_{t+1}}$$

$$\overset{(a)}{\leq} \frac{2}{T}\big(\Phi(x_0) - \inf_x \Phi(x)\big) + \frac{L_\Phi C_\alpha^2}{T\alpha_0^2\varphi_0^2}$$

$$+ \frac{\bar{L}^2}{\mu^2\alpha_0 T}\left[1 + \frac{2}{\mu^2}\Big(\frac{L_{g,2}C_{f_y}}{\mu} + L_{f,1}\Big)^2\right]\big(a_2\log(T) + b_2\big)$$

$$+ \frac{2\bar{L}^2}{\mu^2\alpha_0 T}\big(a_3\log(T) + b_3\big)$$

$$= \frac{1}{2T}\Big(a_4\log(T) + b_4 + 4\big(\Phi(x_0) - \inf_x \Phi(x)\big)\Big), \tag{85}$$

where (a) uses Lemma 16 with $k_0 = 0$.

**Case 2:** If $\alpha_T > C_\alpha$, by Lemma 4, there exists $k_1 \leq T_0$ such that $\alpha_{k_1} \leq C_\alpha$, $\alpha_{k_1+1} > C_\alpha$.

Then for $t < k_1$ when $\alpha_T > C_\alpha$, from Lemma 11, we have

$$\frac{\left\| \nabla\Phi(x_t) \right\|^2}{\alpha_{t+1}\varphi_{t+1}} \leq 2\big(\Phi(x_t) - \Phi(x_{t+1})\big) + \frac{L_\Phi}{\alpha_{t+1}^2\varphi_{t+1}^2}\|\bar{\nabla}f(x_t, y_t, v_t)\|^2$$

$$+ \frac{\bar{L}^2}{\mu^2}\left[1 + \frac{2}{\mu^2}\Big(\frac{L_{g,2}C_{f_y}}{\mu} + L_{f,1}\Big)^2\right]\frac{\left\| \nabla_y g(x_t, y_t) \right\|^2}{\alpha_{t+1}\varphi_{t+1}} + \frac{2\bar{L}^2}{\mu^2}\frac{\left\| \nabla_v R(x_t, y_t, v_t) \right\|^2}{\alpha_{t+1}\varphi_{t+1}}.$$

For $t \geq k_1$ when $\alpha_T > C_\alpha$, from Lemma 11, we have

$$\frac{\left\| \nabla\Phi(x_t) \right\|^2}{\alpha_{t+1}\varphi_{t+1}} \leq 2\big(\Phi(x_t) - \Phi(x_{t+1})\big)$$

$$+ \frac{\bar{L}^2}{\mu^2}\left[1 + \frac{2}{\mu^2}\Big(\frac{L_{g,2}C_{f_y}}{\mu} + L_{f,1}\Big)^2\right]\frac{\left\| \nabla_y g(x_t, y_t) \right\|^2}{\alpha_{t+1}\varphi_{t+1}} + \frac{2\bar{L}^2}{\mu^2}\frac{\left\| \nabla_v R(x_t, y_t, v_t) \right\|^2}{\alpha_{t+1}\varphi_{t+1}}.$$

By taking the average, we can merge $t < k_1$ and $t \geq k_1$ as

$$\frac{1}{T}\sum_{t=0}^{T-1}\frac{\left\| \nabla\Phi(x_t) \right\|^2}{\alpha_{t+1}\varphi_{t+1}} = \frac{1}{T}\sum_{t=0}^{k_1-1}\frac{\left\| \nabla\Phi(x_t) \right\|^2}{\alpha_{t+1}\varphi_{t+1}} + \frac{1}{T}\sum_{t=k_1}^{T-1}\frac{\left\| \nabla\Phi(x_t) \right\|^2}{\alpha_{t+1}\varphi_{t+1}}$$

$$\leq \frac{2}{T}\big(\Phi(x_0) - \Phi(x_{k_1})\big) + \frac{L_\Phi}{\alpha_0^2\varphi_0^2}\frac{1}{T}\sum_{t=0}^{k_1-1}\|\bar{\nabla}f(x_t, y_t, v_t)\|^2$$

$$+ \frac{\bar{L}^2}{\mu^2}\left[1 + \frac{2}{\mu^2}\Big(\frac{L_{g,2}C_{f_y}}{\mu} + L_{f,1}\Big)^2\right]\frac{1}{T}\sum_{t=0}^{k_1-1}\frac{\left\| \nabla_y g(x_t, y_t) \right\|^2}{\alpha_{t+1}\varphi_{t+1}}$$

$$+ \frac{2\bar{L}^2}{\mu^2}\frac{1}{T}\sum_{t=0}^{k_1-1}\frac{\left\| \nabla_v R(x_t, y_t, v_t) \right\|^2}{\alpha_{t+1}\varphi_{t+1}}$$

$$+ \frac{2}{T}\big(\Phi(x_{k_1}) - \Phi(x_T)\big)$$

$$+ \frac{\bar{L}^2}{\mu^2}\left[1 + \frac{2}{\mu^2}\Big(\frac{L_{g,2}C_{f_y}}{\mu} + L_{f,1}\Big)^2\right]\frac{1}{T}\sum_{t=k_1}^{T-1}\frac{\left\| \nabla_y g(x_t, y_t) \right\|^2}{\alpha_{t+1}\varphi_{t+1}}$$

$$+ \frac{2\bar{L}^2}{\mu^2} \frac{1}{T} \sum_{t=k_1}^{T-1} \frac{\left\|\nabla_v R(x_t, y_t, v_t)\right\|^2}{\alpha_{t+1}\varphi_{t+1}}$$

$$\leq \frac{2}{T}\big(\Phi(x_0) - \inf_x \Phi(x)\big) + \frac{L_\Phi}{\alpha_0^2\varphi_0^2} \frac{1}{T} \sum_{t=0}^{k_1-1} \|\bar{\nabla} f(x_t, y_t, v_t)\|^2$$

$$+ \frac{\bar{L}^2}{\mu^2}\left[1 + \frac{2}{\mu^2}\Big(\frac{L_{g,2}C_{f_y}}{\mu} + L_{f,1}\Big)^2\right] \frac{1}{T} \sum_{t=0}^{T-1} \frac{\left\|\nabla_y g(x_t, y_t)\right\|^2}{\alpha_{t+1}\varphi_{t+1}}$$

$$+ \frac{2\bar{L}^2}{\mu^2} \frac{1}{T} \sum_{t=0}^{T-1} \frac{\left\|\nabla_v R(x_t, y_t, v_t)\right\|^2}{\alpha_{t+1}\varphi_{t+1}}$$

$$\leq \frac{2}{T}\big(\Phi(x_0) - \inf_x \Phi(x)\big) + \frac{L_\Phi}{\alpha_0^2\varphi_0^2} \frac{1}{T} \sum_{t=0}^{k_1-1} \|\bar{\nabla} f(x_t, y_t, v_t)\|^2$$

$$+ \frac{\bar{L}^2}{\mu^2\alpha_0 T}\left[1 + \frac{2}{\mu^2}\Big(\frac{L_{g,2}C_{f_y}}{\mu} + L_{f,1}\Big)^2\right] \sum_{t=0}^{T-1} \frac{\left\|\nabla_y g(x_t, y_t)\right\|^2}{\varphi_{t+1}}$$

$$+ \frac{2\bar{L}^2}{\mu^2\alpha_0 T} \sum_{t=0}^{T-1} \frac{\left\|\nabla_v R(x_t, y_t, v_t)\right\|^2}{\varphi_{t+1}}$$

$$\overset{(a)}{\leq} \frac{2}{T}\big(\Phi(x_0) - \inf_x \Phi(x)\big) + \frac{L_\Phi C_\alpha^2}{T\alpha_0^2\varphi_0^2}$$

$$+ \frac{\bar{L}^2}{\mu^2\alpha_0 T}\left[1 + \frac{2}{\mu^2}\Big(\frac{L_{g,2}C_{f_y}}{\mu} + L_{f,1}\Big)^2\right]\big(a_2\log(T) + b_2\big)$$

$$+ \frac{2\bar{L}^2}{\mu^2\alpha_0 T}\big(a_3\log(T) + b_3\big)$$

$$= \frac{1}{2T}\Big(a_4\log(T) + b_4 + 4\big(\Phi(x_0) - \inf_x \Phi(x)\big)\Big), \tag{86}$$

where (a) uses Lemma 16 by plugging in $k_0 = 0$.

Note that **Case 1** and **Case 2** indicate the same result. Thus, we have

$$\frac{1}{T} \sum_{t=0}^{T-1} \|\nabla\Phi(x_t)\|^2 \leq \frac{1}{2T}\Big(a_4\log(T) + b_4 + 4\big(\Phi(x_0) - \inf_x \Phi(x)\big)\Big)\alpha_T\varphi_T$$

$$\overset{(a)}{\leq} \frac{1}{2T}\Big[\Big(a_4\log(T) + b_4 + 4\big(\Phi(x_0) - \inf_x \Phi(x)\big)\Big)^2\varphi_T^2$$

$$+ C_\alpha\Big(a_4\log(T) + b_4 + 4\big(\Phi(x_0) - \inf_x \Phi(x)\big)\Big)\varphi_T\Big]$$

$$\overset{(b)}{\leq} \frac{1}{2T}\Big[\Big(a_4\log(T) + b_4 + 4\big(\Phi(x_0) - \inf_x \Phi(x)\big)\Big)^2\big(a_1\log(T) + b_1\big)^2$$

$$+ C_\alpha\Big(a_4\log(T) + b_4 + 4\big(\Phi(x_0) - \inf_x \Phi(x)\big)\Big)\big(a_1\log(T) + b_1\big)\Big]$$

$$= \mathcal{O}\Big(\frac{\log^4(T)}{T}\Big).$$

where (a) follows from Lemma 17; (b) results from Lemma 15. Thus, the proof is finished.

### F.5 COMPLEXITY ANALYSIS OF ALGORITHM 2 (PROOF OF COROLLARY 2)

Recall in Theorem 2, we know that there exist a constant $M$ such that

$$\frac{1}{T} \sum_{t=0}^{T-1} \|\nabla\Phi(x_t)\|^2 \leq \frac{M\log^4(T)}{T}.$$

When we set the iteration number $T = \frac{MN}{\epsilon} \log^4(\frac{M}{\epsilon})$ and assume the constant $N = 12^4$, we have

$$
\begin{aligned}
\frac{M \log^4(T)}{T} &= \frac{M \log^4(\frac{MN}{\epsilon} \log^4(\frac{M}{\epsilon}))}{\frac{MN}{\epsilon} \log^4(\frac{M}{\epsilon})} \\
&\leq \frac{[\log(N) + \log(\frac{M}{\epsilon}) + 4 \log(\log(\frac{M}{\epsilon}))]^4}{N \log^4(\frac{M}{\epsilon})} \cdot \epsilon \\
&\overset{(a)}{\leq} \left( \frac{\log(N) + 2 \log(\frac{M}{\epsilon})}{N^{\frac{1}{4}} \log(\frac{M}{\epsilon})} \right)^4 \cdot \epsilon \overset{(b)}{\leq} \epsilon,
\end{aligned}
$$

where (a) follows from the inequality $\log(\log(\frac{M}{\epsilon})) \leq \frac{1}{4} \log(\frac{M}{\epsilon})$ for sufficiently small $\epsilon$; (b) holds because $\log(N) + 2 \log(\frac{M}{\epsilon}) \leq N^{\frac{1}{4}} \log(\frac{M}{\epsilon})$ for $N = 12^4$ and $\epsilon$ is sufficiently small. Thus, to achieve $\epsilon$-accurate stationary point, we require $T = \frac{MN}{\epsilon} \log^4(\frac{M}{\epsilon}) = \mathcal{O}(\frac{1}{\epsilon} \log^4(\frac{1}{\epsilon}))$, and the gradient complexity is given by $\text{Gc}(\epsilon) = \Omega(T) = \mathcal{O}(\frac{1}{\epsilon} \log^4(\frac{1}{\epsilon}))$.

