# OpenReview forum: "Tuning-Free Bilevel Optimization: New Algorithms and Convergence Analysis"
_ICLR.cc/2025/Conference — ICLR 2025 Poster_

### Official Review · Reviewer_fFYc · 2024-10-23

**Soundness:** 3
**Presentation:** 3
**Contribution:** 3
**Rating:** 6
**Confidence:** 3

**Summary:**

This paper studies parameter-free algorithms for solving BLO, where the lower level problem is strongly convex and the upper level problem can be nonconvex. The authors propose two tuning-free algorithms that achieve the nearly same rate of the state of the art without knowing the parameter. The stationarity measure is the standard hyper gradient norm. Numerical experiments are conducted to show the effectiveness of the proposed methods.

**Strengths:**

1.This paper combines the parameter free methodology with BLO, which is a good direction to explore.

2.The convergence rates achieved by proposed methods are nearly optimal.

**Weaknesses:**

1. Under the strong convexity of the lower level problem, the BLO is similar to single level problems. This is due to the fact that there is a unique solution $y^*(x)$ for the lower-level problem given an arbitrary upper-level variable $x$, and then the BLO is reduced to the single level problem $\min_{x}\phi(x)=f(x,y^*(x))$. Even further, the gradient of $\phi$ can be computed, as presented in Sec. 3.1 of this paper. Given this, the authors should point out the novelty of the techniques used in this paper compared with single-level parameter-free methods.

2. Second-order information is needed for each iteration, which is computationally expensive. The authors may discuss how to approximate the used hessian matrix or a furture direction to develop methods that are both tuning-free and hessian-free.

**Questions:**

Non.

---

> ### Author Response · Authors · 2024-11-21
>
> We thank the reviewer fFYc for the time and valuable feedback!
>
> **W1:
> Under the strong convexity of the lower-level problem, the BLO is similar to single-level problems. This is due to the fact that there is a unique solution** $y^*(x)$ **for the lower-level problem given an arbitrary upper-level variable** $x$ **, and then the BLO is reduced to the single-level problem** $\min_x \phi(x)-f(x,y^*(x))$ **. Even further, the gradient of $\phi$ can be computed, as presented in Sec. 3.1 of this paper. Given this, the authors should point out the novelty of the techniques used in this paper compared with single-level parameter-free methods.**
>
> **A:** Converting a bilevel problem into a single-level problem is not straightforward. This is mainly because: first, we need to approximate $y^*(x)$; second, we need to approximate the Hessian inverse vector product when we use the implicit function theorem (Section 3.1). Therefore, we need to ensure the convergence of three sequences $y_t$, $v_t$ and $x_t$, where the error bound of $v$ depends on the error bound of $y$, and the error bound of $x$ depends on the error bounds of both $y$ and $v$. Existing single-level methods only need to update a single sequence, so such considerations are unnecessary.
>
> D-TFBO is directly motivated by this idea, employing two sub-loops to achieve precise approximations of  $y^*(x)$ and $v^*(x)$ before updating $x$. By utilizing cold-start stepsizes and stopping criteria for the sub-loops, we establish both upper and lower bounds for the stepsizes. Without additional loops, S-TFBO eliminates additional loops and ensures uniform convergence of $y$, $v$, $x$ through a joint design of adaptive stepsizes. For example, S-TFBO use $\frac{1}{\max\\{\beta_t, \gamma_t\\}}$ as the stepsize to update $v_{t+1}$ and $\frac{1}{\alpha_t\max\\{\beta_t, \gamma_t\\}}$ as the stepsize to update $x_{t+1}$. However, to match the performance of existing well-tuned methods, adaptive approaches require more precise analysis.
>
> Note the proposed methods are the first to achieve completely tuning-free bilevel optimization. Moreover, the convergence rates of the proposed methods (nearly) match those of well-tuned algorithms.
>
> **W2:
> Second-order information is needed for each iteration, which is computationally expensive. The authors may discuss how to approximate the used hessian matrix or a future direction to develop methods that are both tuning-free and hessian-free.**
>
> **A:** Thank you for your valuable point. Computing the full Hessian matrix is indeed expensive. However, we only need to approximate the Hessian-matrix-vector product, which can be efficiently computed as follows:
> 1. First, we compute $\partial f(x)$ and multiply it by $v$ to obtain $\partial f(x)v$, which only involves first-order computations.
> 2. Next, we compute the derivative of $\partial f(x)v$ (a scalar), yielding $\partial^2 f(x)v = \partial[\partial f(x)v]$. This process involves only gradient-like computations and avoids directly computing the Hessian matrix, making it less computationally expensive than anticipated.
>
> However, there are two possible solutions via leveraging [1,2]. The main challenge lies in handling additional parameters, such as the Lagrange multiplier $\lambda$ in [1] and the finite-difference parameter $\delta_\epsilon$ in [2].
> Designing adaptive strategies to update these additional parameters alongside the stepsizes is still an unresolved problem. We aim to tackle these challenges in future research.
>
> [1] A Fully First-Order Method for Stochastic Bilevel Optimization. J. Kwon, D. Kwon, S. Wright, R. Nowak. ICML 2023.
>
> [2] Achieving $\mathcal{O}(\epsilon^{-1.5})$ Complexity in Hessian/Jacobian-free Stochastic Bilevel Optimization. Y. Yang, P. Xiao, K, Ji. NeurIPS 2023.

---

> ### Comment · Reviewer_fFYc · 2024-12-02
>
> The authors' feedback bascally address my concern. I maintain my score.

---

> > ### Author Response · Authors · 2024-12-02
> >
> > Thanks again. We are delighted that our responses effectively addressed your questions and provided the necessary clarification.

---

### Official Review · Reviewer_ZFyM · 2024-10-28

**Soundness:** 3
**Presentation:** 3
**Contribution:** 3
**Rating:** 8
**Confidence:** 3

**Summary:**

In this paper, the authors introduce two tuning-free algorithms, D-TFBO and S-TFBO to solve bilevel problems. D-TFBO employs a double-loop structure with stepsizes adaptively adjusted by the "inverse of cumulative gradient norms'' strategy. S-TFBO features a simpler fully singleloop structure that updates three variables simultaneously with a theory-motivated joint design of adaptive stepsizes for all variables. The authors demonstrate that D-TFBO and S-TFBO respectively require $\mathcal{O}(1/\epsilon)$ and $\mathcal{O}(1/\epsilon \log^4 (1/\epsilon))$ iterations to reach an $\epsilon$-accurate stationary point. The methods are the first to eliminate the need for stepsize tuning while achieving theoretical guarantees.

**Strengths:**

1. The paper is well-written and is easy to read.

2. This paper introduce two tuning-free algorithms, D-TFBO and S-TFBO to solve bilevel problems, which eliminate the need for stepsize tuning while achieving theoretical guarantees.

3. The complexity bounds of the proposed method (nearly) match their well-tuned counterparts using the information of problem parameters.

**Weaknesses:**

1. My primary concern is the update mode of the proposed algorithms. Many tuning-free algorithms do not require prior knowledge of the total number of iterations $T$ (e.g., [1]). However, this is not a significant drawback for me, as addressing bilevel problems with tuning-free algorithms in this context seems to be new.

2. The literature review should comprehensively compare the complexity results and the corresponding problem settings with other methods in a table.

[1] Marumo, Naoki, and Akiko Takeda. "Parameter-free accelerated gradient descent for nonconvex minimization." SIAM Journal on Optimization 34.2 (2024): 2093-2120.

**Questions:**

1. I do not understand the proposed observations in Remarks 1 and 2. The authors could discuss these observations in greater depth, such as by providing more detailed explanations in the main theorems and experiments.

2. I would like to know whether the proposed algorithm can work without the prior knowledge of the total number of iterations $T$ (cf. Weakness 1).

3. I would like to observe the progression of the loss function over time in the experimental results.

4. For other questions, please see the weaknesses above.

---

> ### Author Response · Authors · 2024-11-21
>
> We thank the reviewer ZFyM for the time and valuable feedback!
>
> **W:
> The literature review should comprehensively compare the complexity results and the corresponding problem settings with other methods in a table.**
>
> **A:**
> Thanks for your suggestion. We attached a table illustrating the sub-loop number, total interaction number, gradeint complexity and Hyperparameters tuning requirement to find an $\epsilon$-stationart point. We have added this in our revision (please see Appendix A.1).
>
> | Algorithms | Sub-loop $K$ | Iterations $T$ | Gradient Complexity | Hyperparameters to tune |
> |------------|------------|------------|------------|------------|
> | AID-BiO [2] | $\mathcal{O}(1)$ | $\mathcal{O}(\frac{1}{\epsilon})$ | $\mathcal{O}(\frac{1}{\epsilon})$ | 5 |
> | ITD-BiO [2] | $\mathcal{O}(\log(\frac{1}{\epsilon}))$ | $\mathcal{O}(\frac{1}{\epsilon})$ | $\mathcal{O}(\frac{1}{\epsilon}\log(\frac{1}{\epsilon}))$ | 3 |
> | SOBA [3] | $\mathcal{O}(1)$ | $\mathcal{O}(\frac{1}{\epsilon})$ | $\mathcal{O}(\frac{1}{\epsilon})$ | 3 |
> | D-TFBO (Ours) | $\mathcal{O}(\frac{1}{\epsilon})$ | $\mathcal{O}(\frac{1}{\epsilon})$ | $\mathcal{O}(\frac{1}{\epsilon^2})$ | 0 |
> | S-TFBO (Ours) | $\mathcal{O}(1)$ | $\mathcal{O}(\frac{1}{\epsilon}\log^4(\frac{1}{\epsilon}))$ | $\mathcal{O}(\frac{1}{\epsilon}\log^4(\frac{1}{\epsilon}))$ | 0 |
>
> [2] Bilevel Optimization: Convergence Analysis and Enhanced Design. Kaiyi Ji, Junjie Yang, Yingbin Liang. ICML 2021.
>
> [3] A framework for bilevel optimization that enables stochastic and global variance reduction algorithms. Mathieu Dagréou, Pierre Ablin, Samuel Vaiter, Thomas Moreau. NeurIPS 2022.
>
> **Q1:
> I do not understand the proposed observations in Remarks 1 and 2. The authors could discuss these observations in greater depth, such as by providing more detailed explanations of the main theorems and experiments.**
>
> **A:** Thank you for the question.
> Although the primary goal of this paper is to design tuning-free algorithms, Remarks 1 and 2 provide flexibility for practitioners to tune the algorithms by adjusting constants in the stepsizes and stopping criteria.
> These tunable constants such as $\eta_x$, $\eta_y$ are completely independent of the problem parameters and they do not impact the convergence rate or gradient complexity.
>
> In theory, we can prove convergence rate or gradient complexity are the same and the only difference is that these constants are incorporated into terms such as $\\{C_\alpha, c_1\\}$ in Theorem 1 and $\\{C_\alpha, a_1, b_1, a_4, b_4\\}$ in Theorem 2.
> In the experiments, the initial values in Table 2 represent the various constants discussed in Remarks 1 and 2. The results demonstrate slight performance variation, highlighting the robustness of our algorithms to these tunable constants.
>
> **Q2:
> I would like to know whether the proposed algorithm can work without the prior knowledge of the total number of iterations $T$ (cf. Weakness 1).**
>
> **A:** Thank you for this insightful question. After checking [1], we observe that it is possible to eliminate the dependence on the knowledge of iteration $T$ in S-TFBO. In detail, we can modify the "for" loop in S-TFBO (Algorithm 2) to a "repeat until convergence" structure, as in [1], and this allows S-TFBO to converge to any targeted $\epsilon$-stationary point.
> However, D-TFBO (Algorithm 1) requires the sub-loop stopping criteria to be set as $\epsilon_y = \mathcal{O}(\frac{1}{T})$, $\epsilon_v = \mathcal{O}(\frac{1}{T})$, which depend on prior knowledge of $T. Thus, D-TFBO may not be feasible. We would like to explore this in greater detail in our future work.
>
> We have added this discussion in our revision (please see Appendix A.2).
>
> [1] Parameter-free accelerated gradient descent for nonconvex minimization. Marumo, Naoki, and Akiko Takeda. SIAM Journal on Optimization.
>
> **Q3:
> I would like to observe the progression of the loss function over time in the experimental results.**
>
> **A:**
> For regularization selection and data hyper-cleaning, the results of loss progress over time have already been presented in Appendix B. For coreset selection, we adopt the default settings of initial values, such as the constant learning rates in BCSR and $\alpha_0$, $\beta_0$, $\gamma_0$ in S-TFBO and D-TFBO, all set to 5. We re-ran the methods on Split-CIFAR100 under the balanced scenarios and recorded the loss and running time. The results of loss progress over iteration and time for these methods are shown in Appendix B.

---

> > ### Comment · Reviewer_ZFyM · 2024-11-21
> >
> > I sincerely thank the authors for their clear and comprehensive responses to my problems and concerns. I think this paper will interest the ICLR community, so I have decided to raise my score.

---

> > > ### Author Response · Authors · 2024-11-21
> > >
> > > Dear Reviewer,
> > >
> > > Thank you very much for your updates and for raising your score. We are glad that our responses were able to address your questions and provide clarification.
> > >
> > > Best, Authors

---

### Official Review · Reviewer_wFAr · 2024-11-03

**Soundness:** 3
**Presentation:** 3
**Contribution:** 2
**Rating:** 6
**Confidence:** 2

**Summary:**

The paper introduces tuning-free bilevel optimization algorithms to eliminate the need for prior knowledge of problem-specific parameters. Theoretical convergence is derived for methods and experiments are provided.

**Strengths:**

The paper provides a detailed convergence analysis for the algorithms.

**Weaknesses:**

While I understand the importance of addressing tuning-free bilevel optimization, the main technical novelty of this work is unclear.

**Questions:**

1. How would the methods extend to more general bilevel problems?
2. Could you update citations from arXiv preprints to published versions where available?
3. How does the performance of these tuning-free methods compare directly to well-tuned bilevel optimization algorithms in terms of convergence rate?

---

> ### Author Response · Authors · 2024-11-21
>
> We thank the reviewer wFAr for the time and valuable feedback!
>
> **W: While I understand the importance of addressing tuning-free bilevel optimization, the main technical novelty of this work is unclear.**
>
> **A:** In terms of technical novelty, our work introduces the following innovative designs motivated by the challenges we met:
>
> Since the error bound of $v$ depends on the error bound of $y$, and the error bound of $x$ depends on the error bounds of both $y$ and $v$, we need to address the intertwined dependency. Existing methods focus on single-level problems, where only a single sequence needs to be updated, so such considerations are unnecessary. However, solving bilevel problems requires handling three sequences with the error dependencies mentioned above. Consequently, this necessitates $2^3$ stages analysis, making it significantly more complex than the two-stage analysis used in single-level problems.
>
> To address this, we explore tuning-free methods within both double-loop and single-loop structures. Our D-TFBO algorithm introduces cold-start adaptive stepsizes that accumulate gradients exclusively within the sub-loops. Additionally, S-TFBO adopts a joint design of adaptive stepsizes for $y$, $v$, and $x$, corresponding to solving the inner problem, the linear system, and the outer problem, respectively. For example, S-TFBO use $\frac{1}{\max\\{\beta_t, \gamma_t\\}}$ as the stepsize to update $v_{t+1}$ and $\frac{1}{\alpha_t\max\\{\beta_t, \gamma_t\\}}$ as the stepsize to update $x_{t+1}$. Moreover, we need to provide more precise analysis on accumulated stepsizes to ensure our algorithms achieve matching convergence rates with existing well-tuned bilevel methods. This is more challenging than the analysis in the single-level cases.
>
> Note the proposed methods are the first to achieve completely tuning-free bilevel optimization. Moreover, the convergence rates of the proposed methods (nearly) match those of well-tuned algorithms.
>
> **Q1: How would the methods extend to more general bilevel problems?**
>
> **A:**
> This is a great question. Our proposed algorithms have significant potential for extension to more general cases. For instance, we could consider the PL condition for the lower-level problem by incorporating the analysis from [1]. Additionally, we can explore our algorithms in stochastic and distributed settings, where existing work [2,3] may provide insights to help overcome these challenges.
> Since this is the first work exploring adaptive and tuning-free stepsizes in bilevel settings, we aim to leave these challenges in future work.
>
> [1] A Generalized Alternating Method for Bilevel Optimization under the Polyak-Łojasiewicz Condition. Q. Xiao, S. Lu, and T. Chen. NeurIPS 2023.
>
> [2] On the Convergence of AdaGrad(Norm) on $\mathbb{R}^{d}$: Beyond Convexity, Non-Asymptotic Rate and Acceleration. Z. Liu, T. Nguyen, A. Ene, H. Nguyen. ICLR 2023.
>
> [3] SimFBO: Towards Simple, Flexible and Communication-efficient Federated Bilevel Learning. Y. Yang, P. Xiao, K. Ji. NeurIPS 2023.
>
> **Q2. Could you update citations from arXiv preprints to published versions where available?**
>
> **A:** Sure, thanks for reminding us. We have revised this.
>
> **Q3. How does the performance of these tuning-free methods compare directly to well-tuned bilevel optimization algorithms in terms of convergence rate?**
>
> **A:** Here we attached a table illustrating the sub-loop number, total interaction number, gradient complexity and hyperparameter tuning requirement to find an $\epsilon$-stationart point. We have added this in our revision (please see Appendix A.1).
>
> | Algorithms | Sub-loop $K$ | Convergence Rate $T$ | Gradient Complexity | Hyperparameters to tune |
> |------------|------------|------------|------------|------------|
> | AID-BiO [4] | $\mathcal{O}(1)$ | $\mathcal{O}(\frac{1}{\epsilon})$ | $\mathcal{O}(\frac{1}{\epsilon})$ | 5 |
> | ITD-BiO [4] | $\mathcal{O}(\log(\frac{1}{\epsilon}))$ | $\mathcal{O}(\frac{1}{\epsilon})$ | $\mathcal{O}(\frac{1}{\epsilon}\log(\frac{1}{\epsilon}))$ | 3 |
> | SOBA [5] | $\mathcal{O}(1)$ | $\mathcal{O}(\frac{1}{\epsilon})$ | $\mathcal{O}(\frac{1}{\epsilon})$ | 3 |
> | D-TFBO (Ours) | $\mathcal{O}(\frac{1}{\epsilon})$ | $\mathcal{O}(\frac{1}{\epsilon})$ | $\mathcal{O}(\frac{1}{\epsilon^2})$ | 0 |
> | S-TFBO (Ours) | $\mathcal{O}(1)$ | $\mathcal{O}(\frac{1}{\epsilon}\log^4(\frac{1}{\epsilon}))$ | $\mathcal{O}(\frac{1}{\epsilon}\log^4(\frac{1}{\epsilon}))$ | 0 |
>
> [4] Bilevel Optimization: Convergence Analysis and Enhanced Design. Kaiyi Ji, Junjie Yang, Yingbin Liang. ICML 2021.
>
> [5] A framework for bilevel optimization that enables stochastic and global variance reduction algorithms. Mathieu Dagréou, Pierre Ablin, Samuel Vaiter, Thomas Moreau. NeurIPS 2022.

---

### Official Review · Reviewer_uEX3 · 2024-11-04

**Soundness:** 3
**Presentation:** 3
**Contribution:** 3
**Rating:** 6
**Confidence:** 4

**Summary:**

This paper has proposed two tuning-free algorithms for stochastic bilevel optimization, D-TFBO and S-TFBO, which eliminates the need for stepsize tuning that depends on problem-specific parameters. D-TFBO follows a double-loop structure, while S-TFBO utilizes a fully single-loop approach with a joint design of adaptive stepsizes. Convergence rates for both methods are established, and numerical results on data hyper-cleaning, regularization selection, and coreset selection validate the effectiveness of the proposed algorithms.

**Strengths:**

Parameter tuning is challenging in bilevel optimization, as bilevel problems typically involve more hyperparameters than single-level learning. This paper presents two versions of tuning-free methods with both theoretical guarantees and experimental validation, making it a valuable contribution to the bilevel optimization community.

Theoretical guarantees for the proposed methods are solid, and the numerical performance demonstrates a clear advantage, showcasing the effectiveness of these approaches.

**Weaknesses:**

This paper addresses only deterministic bilevel optimization, leaving it unclear whether the proposed technique is robust in stochastic settings.

**Questions:**

This paper proposes two versions of tuning-free bilevel algorithms. It appears that the single-loop algorithm offers better gradient complexity, while the double-loop algorithm demonstrates superior empirical performance. Could the authors comment on the reasons for this discrepancy—such as whether it stems from a less tight convergence rate for the double-loop method or something else? Additionally, guidance on when to choose the double-loop versus single-loop version would be helpful for practioner.

---

> ### Author Response · Authors · 2024-11-21
>
> We thank the reviewer uEX3 for the time and valuable feedback!
>
> **W: This paper addresses only deterministic bilevel optimization, leaving it unclear whether the proposed technique is robust in stochastic settings.**
>
> **A:** However, extending the proposed methods to the stochastic setting is not straightforward and there are still some unresolved theoretical challenges. For instance, addressing the bias in the "inverse of cumulative gradient norms" stepsizes is not trivial; the variance in first- and second-order gradient estimates may affect stepsizes' bounds and algorithm convergence; and the two-stage analysis and coupled stepsize structure may require special conditions to function effectively. Some existing work [1,2] may offer insights to help address these challenges.
>
> As the first work exploring fully tuning-free bilevel optimization problems, this paper primarily focuses on the fundamental challenges and has already made substantial progress in both development and analysis. We are willing to address the aforementioned challenges in future work.
>
> [1] AdaGrad Stepsizes: Sharp Convergence Over Nonconvex Landscapes. R. Ward, X. Wu, L. Bottou. JMLR.
>
> [2] On the Convergence of AdaGrad(Norm) on $\mathbb{R}^{d}$: Beyond Convexity, Non-Asymptotic Rate and Acceleration. Z. Liu, T. Nguyen, A. Ene, H. Nguyen. ICLR 2023.
>
> **Q:
> This paper proposes two versions of tuning-free bilevel algorithms. It appears that the single-loop algorithm offers better gradient complexity, while the double-loop algorithm demonstrates superior empirical performance. Could the authors comment on the reasons for this discrepancy—such as whether it stems from a less tight convergence rate for the double-loop method or something else? Additionally, guidance on when to choose the double-loop versus single-loop version would be helpful for practioner.**
>
> **A:** This is a great question!
> The worse complexity but superior empirical performance can be caused by:
> 1. In our analysis, we consider the worst-case complexity, which accounts for the maximum number of sub-loop iterations required to ensure convergence.
> 2.  In practice, the sub-loop can terminate earlier within the "while" loops, requiring fewer iterations than predicted by the analysis.
> 3.  Some evidence also observes that double-loop provides better generalization performance [3,4].
>
> As noted in [5], developing a tighter convergence analysis in the strongly convex setting is an intriguing topic, and we plan to address this in future research. In practice, D-TFBO ensures higher accuracy, as shown in most of our experiments but is harder to implement and the sub-loops cause the waiting time to update $x$; S-TFBO achieves slightly worse performance but it has advantages such as simple implementation and no waiting time for updating $x$.
>
> As practical guidance for practitioners, D-TFBO is well-suited for scenarios requiring high accuracy, while S-TFBO is preferable for its simpler implementation and no waiting time when updating the objective variable.
>
> We have noted this in our revision (Please see Appendix B.1).
>
> [3] Will Bilevel Optimizers Benefit from Loops. K. Ji, M. Liu, Y. Liang, L. Ying. NeurIPS 2022.
>
> [4] On Implicit Bias in Overparameterized Bilevel Optimization. P. Vicol, J. Lorraine, F. Pedregosa, D. Duvenaud, R. Grosse. ICML 2022.
>
> [5] Linear Convergence of Adaptive Stochastic Gradient Descent. Y. Xie, X. Wu, R. Ward. AISTATS 2020.

---

> > ### Comment · Reviewer_uEX3 · 2024-11-25
> >
> > I thank the authors for their detailed response. It solved all of my concerns and I'll keep my score.

---

> > > ### Author Response · Authors · 2024-11-25
> > >
> > > Dear Reviewer,
> > >
> > > Thank you once again for taking the time to review our paper. We are pleased that our responses were able to address your questions and provide the necessary clarification.
> > >
> > > Best, Authors

---

### Meta-Review · Area_Chair_WGDp · 2024-12-20

**Metareview:**

This paper studies a practically important problem on bilevel optimization with adaptive tuning of stepsizes, which will reduce significant efforts in tuning stepsizes when some of the problem parameters are unknown. The paper proposes two novel tuning-free algorithms, where one employs a double-loop structure with stepsizes and the other features a fully single-loop structure. All the reviewers are active researchers in the field, and have achieved consensus on the acceptance of this paper. I also read the paper and believe that the paper will add value to the bilevel optimization community.

**Additional Comments On Reviewer Discussion:**

The discussion was fruitful and engaging.

---

### Decision · Program_Chairs · 2025-01-22

Accept (Poster)